# Convergent causal mapping unravels distinct frontal networks for visuospatial selective attention

Guglielmo Puglisi [1,7], Luca Viganò [1,7] ✉, Antonella Leonetti[1], Marco Rossi [2], Tommaso Sciortino[3], Marco Conti Nibali[3], Lorenzo Gabriel Gay[3], Luca Mollica[4], Luca Fornia[5], Gabriella Cerri[1] & Lorenzo Bello[6]

Orienting visuospatial attention towards relevant stimuli is vital for effective environmental interactions. Current attentional control models rely on functional neuroimaging, which is correlational, and lesion studies in stroke patients, affected by localization bias. Studying patients undergoing awake neurosurgery for brain tumour resection offers a unique chance to overcome these limitations and possibly enhance current neurofunctional models. We combined Lesion-Symptom-Mapping (LSM) in 163 brain tumour patients and Direct Electrical Stimulation (DES) in 47 patients during awake neurosurgery to unveil the network causally associated with visuospatial exploratory/selective attention. LSM and DES convergently identified a right dorsomedial frontal region linked to visuospatial neglect, potentially functioning as a pre-oculomotor hub for contralateral attentional deployment. Moreover, stimulation of right ventrolateral white matter was associated with visuospatial errors in both hemifields. Finally, we provided a tool that effectively detects and preserves frontal connectivity for visuospatial exploratory/selective attention in neurosurgical settings.

The ability to orient visuospatial attention toward relevant external stimuli is essential for effective interaction with the environment[1]. This requires balancing goal-directed attention with the flexibility to shift focus to unexpected yet relevant stimuli, ensuring adaptability. When this dynamic balance breaks down—as in hemispatial neglect—patients can exhibit a marked ipsilesional bias, difficulty reorienting to contralesional cues, and frequent omissions in the neglected hemifield[2]. Classical models propose that hemispatial neglect primarily results from right-hemisphere lesions that disrupt attention balance between the hemispheres, leading the left hemisphere to dominate attention toward the contralateral visual field. In contrast, damage to the left hemisphere rarely causes true neglect because the right hemisphere has the ability to orient attention to both sides of space, compensating for the left hemisphere impairment[3,4]. This view was complemented by spatial-motor accounts, such as the premotor theory, which framed neglect as a failure to generate the oculomotor 'drive' necessary for contralesional exploration[5]. In parallel, alternative frameworks argued that the fundamental deficit in visuospatial neglect was non-spatial, stemming from impaired arousal/vigilance functions for which the right hemisphere is dominant[6,7]. In their dual-network model Corbetta et al.[8,9], reconciled these theoretical streams proposing that visuospatial neglect is explained by both spatial and non-spatial impairments.

[1]MoCA Laboratory, Department of Medical Biotechnology and Translational Medicine, University of Milan, IRCCS Galeazzi Sant'Ambrogio, Milan, Italy. [2]Neurosurgical Oncology Unit, Department of Medical Biotechnology and Translational Medicine, University of Milan, IRCCS Galeazzi Sant'Ambrogio, Milan, Italy. [3]Neurosurgical Oncology Unit, IRCCS Galeazzi Sant'Ambrogio, Milan, Italy. [4]Department of Medical Biotechnology and Translational Medicine, University of Milan, Milan, Italy. [5]MoCA Laboratory, Department of Medical Biotechnology and Translational Medicine, University of Milan, Milan, Italy. [6]Neurosurgical Oncology Unit, Department of Oncology and Haemato-Oncology, University of Milan, Milan, Italy. [7]These authors contributed equally: Guglielmo Puglisi, Luca Viganò. ✉e-mail: luca.vigano2@unimi.it

This model proposes that a right-lateralized ventral attention network (VAN; temporoparietal junction, ventral frontal cortex) mediates automatic reorienting and maintains arousal for target detection, while a bilateral dorsal attention network (DAN; frontal eye fields, intraparietal sulcus) handles top-down voluntary orienting. Damage to the VAN not only hinders stimulus-driven reorienting but also leads to an interhemispheric DAN imbalance, amplifying ipsilesional bias[8,9].

Functional neuroimaging (primarily fMRI) and lesion symptom mapping (LSM) in stroke patients have provided critical insights into the identification and functional characterization of this large-scale network, laying the foundation for all subsequent research[8–11]. Nevertheless, both methodologies have intrinsic shortcomings. Functional neuroimaging measures correlations between brain regions and behaviour, but falls short of establishing causality[12,13]. LSM in stroke models provides causal insights into brain regions underlying dysfunction but faces a major bias due to the prevalence of lesions within the middle cerebral artery territory[14–16]. This localization bias led to a substantial underexploration of dorsomedial parieto-frontal regions, weakening the causal explanatory power of LSM. In fact, unlike stroke models, LSM studies conducted in brain tumour patients revealed that the resection of medial structures (e.g. the supplementary and cingulate eye fields) is associated with visuospatial selective attention deficits and neglect syndrome[17,18]. To address these limitations, we conducted a study involving patients undergoing awake neurosurgery for brain tumour removal. Distinctly to previous investigations in this area[17,18], we aimed to conduct a *convergent causal mapping* by integrating multiple causal approaches, specifically lesion-symptom mapping (LSM) and direct electrical stimulation (DES)[19–22]. DES, despite its inherent limitations (e.g. the potential spread of current to neighbouring and/or remote regions[19,23]), provides a causal reversible probe to test anatomo-functional relationship with a unique invasive access to the human brain. The strategy of combining these two causal methods allows their respective limitations to be mitigated. By demonstrating concordance between permanent deficits from surgical lesions (LSM) and transient disruptions from DES at the same anatomical locations, a high degree of causal inference can be achieved[24]. Moreover, the study of glioma patients allows for a systematic exploration of dorso-medial territories, as the removal of brain tumors may require surgical exposure and/or resection of those areas and the related connectivity. Specifically, we aimed at providing convergent causal evidence on the cortico-subcortical networks involved in visuospatial selective attention. Drawing from prior studies involving glioma patients[17,18], we predict that resection of right frontal dorsomedial areas might produce visuospatial neglect. Coherently, if these regions play a pivotal role in contralateral attention allocation, we anticipate that their intraoperative stimulation during a visuospatial selective attention task will induce neglect-like errors. Furthermore, according to the concept of the Ventral Attention Network (VAN) as a system sustaining reorienting, arousal, and vigilance toward both hemifields[8,9], disruptions in the ventrolateral regions are expected to lead to errors along the entire visual field.

This study was performed in two cohorts of patients ($n = 210$, 109 right hemisphere):
1) A retrospective cohort of 163 patients (82 right hemisphere) was analysed using support vector regression lesion-symptom mapping (SVR-LSM) to identify brain areas linked to declines in visuospatial exploratory/selective attention. We then delineated the structural network associated with these regions by generating a quantitative, region-to-region structural connectome from 100 healthy Human Connectome Project (HCP) participants. Using graph theory, we identified the critical hubs and edges of a neglect-specific network, and subsequently validated the clinical relevance of these connections by correlating their surgical disconnection with postoperative deficits in 15 patients with preoperative diffusion MRI. The core anatomical substrate of the network was visualised using a population-average track-density imaging (TDI) map.
2) A prospective cohort of 47 patients undergoing awake neurosurgery (27 right hemisphere) performed an intraoperative visuospatial selective attentional task (iVSAT) during brain mapping guided by DES. DES-related errors and the anatomical distribution of iVSAT sites were analyzed using spatial probability methodology[25], and in a subset of 9 patients, we used TDI maps to visualise the specific white matter underlying distinct, stimulation-induced attentional interferences.

Finally, the spatial convergence between lesion-symptom results and intraoperative functional mapping was assessed to show a causally validated right frontal network subserving contralateral visuospatial exploratory/selective attention. The overall workflow of the study, detailing the integration of these distinct methodological stages, is illustrated in Fig. 1.

## Results

### Retrospective study

**Patient cohort.** Initially, 171 patients were considered in the retrospective cohort. Of these, 8 patients were excluded due to visual deficits (2 had hemianopia, and 6 had quadrantopia). Then, 163 (82 right) patients operated for glioma resection were included. Sociodemographic and clinical characteristics of the patients are detailed in Supplementary Table 1 (mean age 43.9 years, s.d. 13.4; 72 females, 9 left-handers; 69 diagnosed with a Lower-grade glioma).

**Neuropsychological Assessment.** The preoperative and postoperative (1-month) neuropsychological performances in the visuospatial exploratory/selective attention test, i.e. Bells cancellation task, of patients recruited for the retrospective analysis are detailed in Supplementary Table 1. Preoperatively, 10 patients (5 left hemisphere) had a pathological total score (total missed targets). Differently, no patient had a pathological asymmetry score (left minus right omissions; higher scores denote more severe visuospatial neglect) (see Vallar et al. 1994[26] for Italian Norms). At the 1-month evaluation, an increased total score was observed in both right and left hemisphere patients (Supplementary Fig. 1). A repeated-measures ANOVA revealed a significant increase in total score from pre-surgery (right hemisphere mean = 1.39($\pm$1.50); left hemisphere mean = 1.9($\pm$2.19)) to postsurgery (right hemisphere mean = 3.50($\pm$3.46); left hemisphere mean = 3.64($\pm$3.29)); $F(1,161) = 54.23$, $p < .001$, $\eta^2_p$=.252 95% CI [0.177, 0.331]) with no significant Time×Hemisphere interaction ($F(1,161) = 0.50$, $p = .481$, $\eta^2_p$=.003 95% CI [0.000, 0.041]). Due to significant deviation from normality (Shapiro−Wilk $p < .001$), we used Wilcoxon signed-rank tests, which confirmed a statistically significant increase in total score following surgery in both left-hemisphere ($n = 81$, $Z = −4.35$, $p < .001$, $r\_rb = 0.632$, 95% CI [0.420, 0.819]) and right-hemisphere patients ($n = 82$, $Z = −5.25$, $p < .001$ $r\_rb = 0.725$, 95% CI [0.549, 0.867]), corroborating the ANOVA. At the one-month follow-up, 23/82 (28%) right-hemisphere and 31/81 (38%) left-hemisphere patients showed a pathological total score.

For the asymmetry score (Supplementary Fig. 2), a repeated-measures ANOVA revealed a significant Time×Hemisphere interaction ($F(1,161) = 24.92$, $p < .001$, $\eta^2_p$=.134, 95% CI [−0.274, 0.177]). Post-hoc tests showed left-hemisphere patients had no significant postoperative change in asymmetry score (mean pre $0.06 \pm 0.98$, post $0.1 \pm 0.97$; two sided $t(80) = −0.395$, $p = 0.694$, Cohen's $d = −0.044$, 95% CI [−0.274, 0.177]), whereas patients with right lesions showed a significantly higher score (mean pre $0.49 \pm 1.08$, post $2.48 \pm 3.29$; two side $t(81) = −5.272$, $p < 0.001$, Cohen's $d^z = −0.582$, 95% CI [−0.765, −0.413]). Wilcoxon tests confirmed these findings: no change for left-hemisphere patients ($n = 81$; $Z = −0.37$, $p = .712$, $r\_rb = 0.070$, 95% CI [−0.303, 0.435]) but a significant postoperative increase for right-

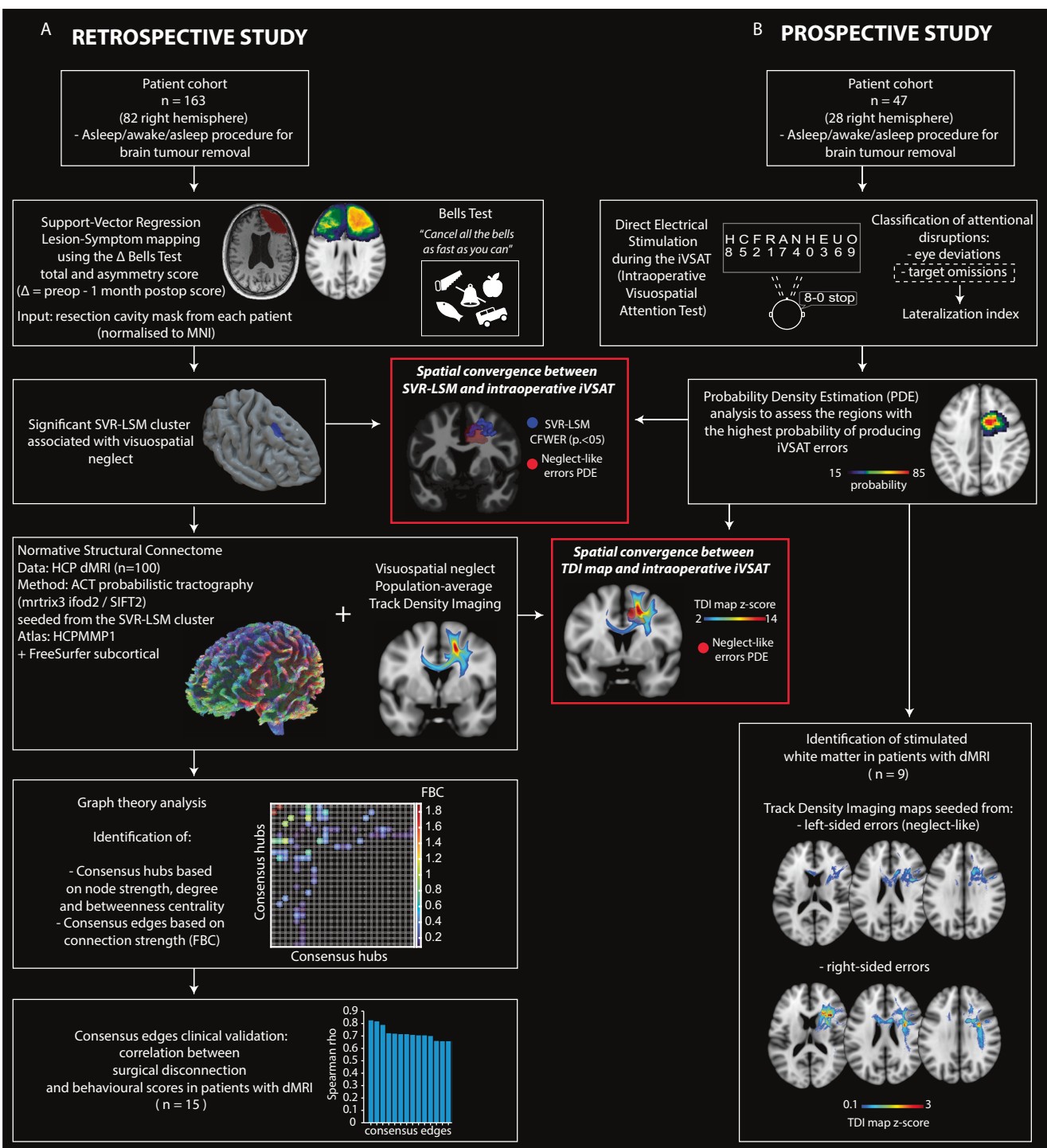

**Fig. 1 | Study flow chart.** The retrospective study (panel **A**, *n* = 163) utilised Support Vector Regression Lesion-Symptom Mapping (SVR-LSM) to identify a brain region causally associated with visuospatial exploratory/selective attention. This cluster was then used to define a normative structural connectome and its core connections. The prospective study (panel **B**, *n* = 47) employed intraoperative Direct Electrical Stimulation (DES) during a visuospatial selective attention task (iVSAT) to independently map eloquent brain structures. The red boxes highlight the analyses demonstrating the spatial convergence between the lesion-symptom mapping / connectome results and the intraoperative mapping.

hemisphere patients (*n* = 82; *Z* = −4.53, *p* < .001, *r_rb* = 0.604, 95% CI [0.399, 0.774]).

One month post-surgery, 21/82 (26%) right-hemisphere patients displayed pathological asymmetry score, indicating visuospatial neglect, while no left-hemisphere patients did. In summary, visuospatial selective attention declined bilaterally, but only right-hemisphere patients developed an ipsilesional attentional bias (neglect). In this group, total and asymmetry scores were strongly

correlated (*t*(80) = 10.79, *p* < 0.001, Spearman's *ρ* = 0.770, 95% CI [0.664, 0.846]), unlike in the left-hemisphere group (*t*(79) = − 0.48, *p* = 0.633, Spearman's *ρ* = −0.054, 95% CI [ − 0.269, 0.166]), indicating that in the right hemisphere group the overall deficit was driven by visuospatial neglect.

**Lesion-symptom mapping - SVR-LSM.** To identify brain regions whose surgical resection was associated with postoperative changes in

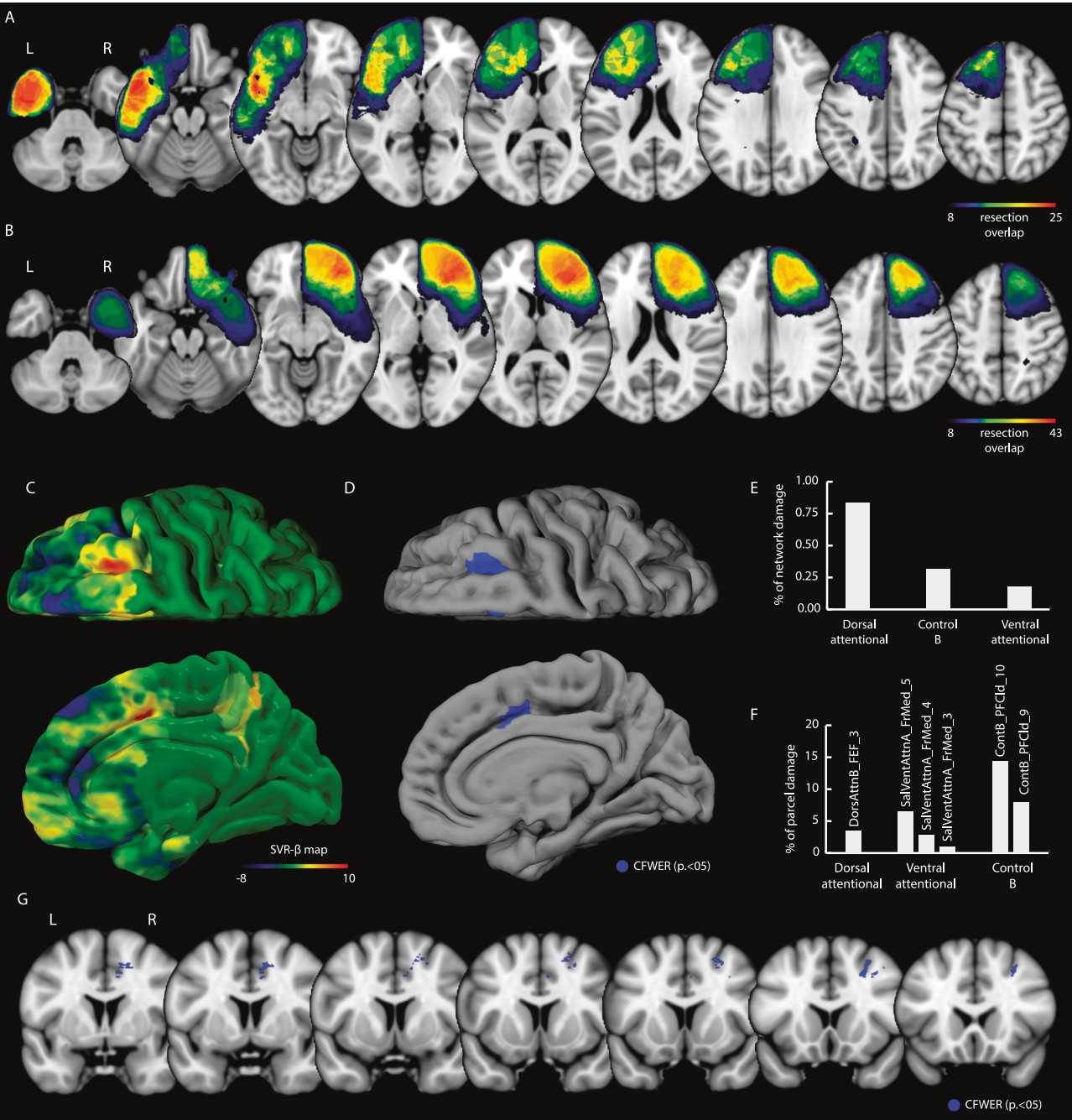

**Fig. 2 | Support Vector Regression Lesion-Symptom Mapping (SVR-LSM) results. A** Resection overlap maps for the left ($n = 81$) (**A**) and right ($n = 82$) (**B**) hemisphere patient cohorts. The colour bar indicates the number of patients with overlapping resections at each voxel. Lower threshold is set to display only voxels considered in the SVR-LSM statistical map (10% of patients). The raw β-map of the SVR-LSM is shown in (**C**). The cluster associated with the impairment in the asymmetry score of the Bell Test after correction for cluster-level family-wise error (CFWER, $p = 0.05$, v = 1) is shown in blue in a 3D FS average template (**D**) and on coronal slices (**G**). Percentage of network-wise (**E**) and parcel-wise damage (**F**) (resting-state functional networks, Schaefer et al. 2018) associated with the SVR-LSM cluster. Source data are provided as a Source Data file.

visuospatial exploratory/selective attention, we performed Support Vector Regression Lesion-Symptom Mapping (SVR-LSM) separately for patients with left and right hemisphere lesions, controlling for age, education, tumour grade, and resection volume (see Methods). The shared resected voxels in the two cohorts are illustrated in Fig. 2A, B. In left-hemisphere patients ($n = 81$), SVR-LSM revealed no significant associations between resection and postoperative changes in either the Bells Test total (Δ total score = difference between pre- and 1-month postoperative number of omitted targets) or asymmetry

score (Δ asymmetry score = difference between pre- and 1-month postoperative asymmetry score).

Since in the right hemisphere group, the bias toward one hemifield (i.e. visuospatial neglect) was the primary cause of the overall impairment in visuospatial exploratory/selective attention, only the Δ asymmetry score was taken into account as behavioural variable in the SVR-LSM. Results of SVR-LSM for both variables (total score and asymmetry score) are reported in the Supplementary Fig. 3, showing a considerable overlap of the significant clusters. Consistently, when

running SVR-LSM using the Δ total score as the dependent variable and the Δ asymmetry score as the covariate, no significant voxels survived. The unthresholded β-map associated with the Δ asymmetry score is shown in Fig. 2C. After cluster-level family-wise error correction, results showed the existence of two cortical clusters: 1) superior/middle frontal gyri (within the superior frontal sulcus, anterior to FEF); 2) a middle cingulate cluster (Fig. 2-D). Significant white matter voxels were found in the superior frontal gyrus (at the SMA/preSMA transition) and below the middle frontal gyrus (Fig. 2G). Hyperparameters: *Cost/BoxConstraint* = 79.78, *Sigma/KernelScale* = 0.97, *Epsilon* = 2.33, prediction accuracy = 0.30 ± 0.09, reproducibility index $r$ = 0.85.

To assess the coherence of this finding with a traditional clinical-categorical approach, we performed a supplementary analysis. A resection overlap map generated from only those patients with a clinically pathological asymmetry score ($n$ = 21) demonstrated substantial spatial overlap with the SVR-LSM cluster, supporting the region's association with clinically significant deficits (see Supplementary Fig. 4).

To assess damage to large-scale functional networks, we quantified the overlap between the SVR-LSM cluster and the 17-network Schaefer/Yeo atlas[27]. The Dorsal Attention (0.84% of its volume overlapped), Fronto-parietal Control B (0.32%), and Salience/Ventral Attention (0.18%) networks were affected (Fig. 2E). To identify the specific cortical sub-regions involved, a parcel-level analysis (Schaefer 1000-Parcel Atlas) highlighted the involvement of specific sub-regions, including the DorsAttnB_FEF_3, within the superior frontal sulcus anterior to the FEF proper, (3.5% overlap) in the Dorsal Attention Network, lateral dorsal prefrontal cortex parcels in the Fronto-parietal Control (e.g., ContB_PFCld_10, 14.5%; ContB_PFCld_9, 7.9%) and medial cingulate parcels in the Salience/Ventral Attention (e.g., Sal-VentAttnA_FrMed_5, 6.6%) networks (Fig. 2F).

**Normative visuospatial neglect structural connectome.** To define the core structural network associated with visuospatial neglect, we constructed connectomes for 100 healthy Human Connectome Project (HCP) participants[28]. Using MRtrix3, we performed multi-shell multi-tissue CSD[29], anatomically-constrained probabilistic tractography (10 million streamlines/subject)[30], and SIFT2 filtering to create quantitative connectomes[31]. We retained only streamlines traversing the SVR-LSM cluster associated with visuospatial neglect and generated a 379 × 379 connectome matrix for each participant. Each node is an HCP-MMP1.0 cortical parcel or a FreeSurfer subcortical parcel, and each edge is weighted by its Fibre Bundle Capacity (FBC, mm²) (See Methods). FBC represents a white matter connectivity measure in the form of the total intra-axonal cross-sectional area of a given connection[32].

**Graph theory analysis.** Graph-theory analysis of these individual FBC matrices was performed to identify key topological features of this neglect-specific network (see Methods). This analysis identified 10 consensus hubs – regions consistently demonstrating high centrality across participants, defined by a composite hub z-score (integrating within-subject normalized node strength, degree, and betweenness centrality) exceeding 1.5 in at least 50% of subjects. These were: R-6ma (SMA; $Z$ = 7.36), R-i6-8 (posterior MFG; $Z$ = 7.21), R-8Av (posterior MFG; $Z$ = 6.96), R-SCEF (Supplementary and Cingulate Eye Field; $Z$ = 5.88), R-Thalamus ($Z$ = 4.75), R-SFL (Superior Frontal Language area; $Z$ = 3.72), R-6a (posterior superior frontal sulcus; $Z$ = 3.61), R-s6-8 (posterior superior frontal gyrus; $Z$ = 2.26), R-8 C (posterior middle frontal gyrus; $Z$ = 2.25), and R-p24pr (middle cingulate gyrus; $Z$ = 1.87). The spatial distribution of hubs and their composite hub score is illustrated in Fig. 3A, B; detailed metrics are in Supplementary Table 2.

The analysis further delineated consensus top edges, representing the most consistently important structural connections of these hubs. These edges were identified based on their strength (Fibre

Bundle Capacity having a z-score > 1.5 relative to the hub's other connections in at least 50% of subjects). We identified 33 consensus top edges. Based on their anatomical profile, they can be subdivided in three groups: 1) 35 cortico-cortical connections linking the consensus hubs to other right-neighbour frontal regions; 2) 12 projection bundles connecting frontal hubs to the thalamus, caudate, pallidum, putamen, and brain-stem; and 3) 5 transcallosal links connecting right midline frontal parcels with left-hemisphere regions. All the consensus edges and their relative FBC are reported in Fig. 3C.

**Population-average TDI map.** Finally, to visualise the integrated anatomical structure of this visuospatial neglect network, we generated a population-averaged, SIFT2-weighted track-density image (TDI)[33,34] from the SVR-LSM filtered tractograms (see Methods section), showing the average apparent fibre density across the cohort. The white matter with the highest density across the population ($Z$ > 2) was localised below the SFG and MFG, comprising intra-frontal connections and white matter extending towards the corpus callosum and the internal capsule (Fig. 3D, E).

**Clinical validation of consensus edges via patient-specific disconnection modelling.** To validate the clinical relevance of the 'consensus edges' identified from the normative connectome, we investigated whether their surgical disconnection correlated with postoperative deficits in 15 patients with preoperative dMRI. Leveraging patient-specific tractography to account for pathology-induced distortions, we quantified disconnection for each edge as the percentage reduction in its Fibre Bundle Capacity (FBC). This was calculated by comparing the preoperative FBC to a postoperatively simulated state, which was generated by removing streamlines that transected the resection cavity. A Spearman rank correlation revealed that this percentage of disconnection was significantly associated with neglect severity (Δ Bells Test asymmetry score) for three specific pathways after Bonferroni correction: the right SCEF-thalamus ($rho$ = 0.83, $p$ = 0.006), right SCEF-brainstem ($rho$ = 0.82, $p$ = 0.008), and the transcallosal right SFL-left SCEF connection ($rho$=0.79, $p$ = 0.018) (Fig. 4). These results therefore provide clinical validation for these core pathways, isolating connections whose disruption most strongly associate with postoperative contralateral visuospatial exploratory/selective attention deficits.

## Prospective study

**Patient cohort.** Of the 50 patients considered for enrolment, 47 patients (28 right hemisphere) undergoing brain tumour resection in awake anaesthesia met the inclusion criteria. One patient was excluded due to hemianopia, and two were excluded for visuospatial attention deficits. Sociodemographic and clinical features as well as attentional scores are reported in Supplementary Table 3.

**Intraoperative stimulation.** An intraoperative adapted version of a classical visuospatial selective attention test[35] was used to assess and preserve visuospatial exploratory/selective attention during awake procedures involving areas highlighted by previous lesion-symptom correlation (Fig. 5). All the 47 patients (28 right, see Supplementary Table 3) were able to intraoperatively perform the iVSAT properly during surgery. The Low-Frequency Direct Electrical Stimulation (LF-DES) applied for iVSAT mapping had a mean intensity of 3.82 ± 1.19 mA (range: 2.0 – 6.0 mA) in right hemisphere patients and 3.42 ± 1.19 mA (range: 1.5 – 6.0 mA) in left hemisphere patients. During the initial phase of mapping on the exposed cortical surface, prior to resection, no interferences in iVSAT task performance were evoked by DES in either right or left hemisphere patients. After corticectomy, the tested area covered the regions below the frontal, middle, and inferior frontal gyri (SFG, MFG, and IFG, respectively), anterior to the white matter region recently described as belonging to the frontal circuitry

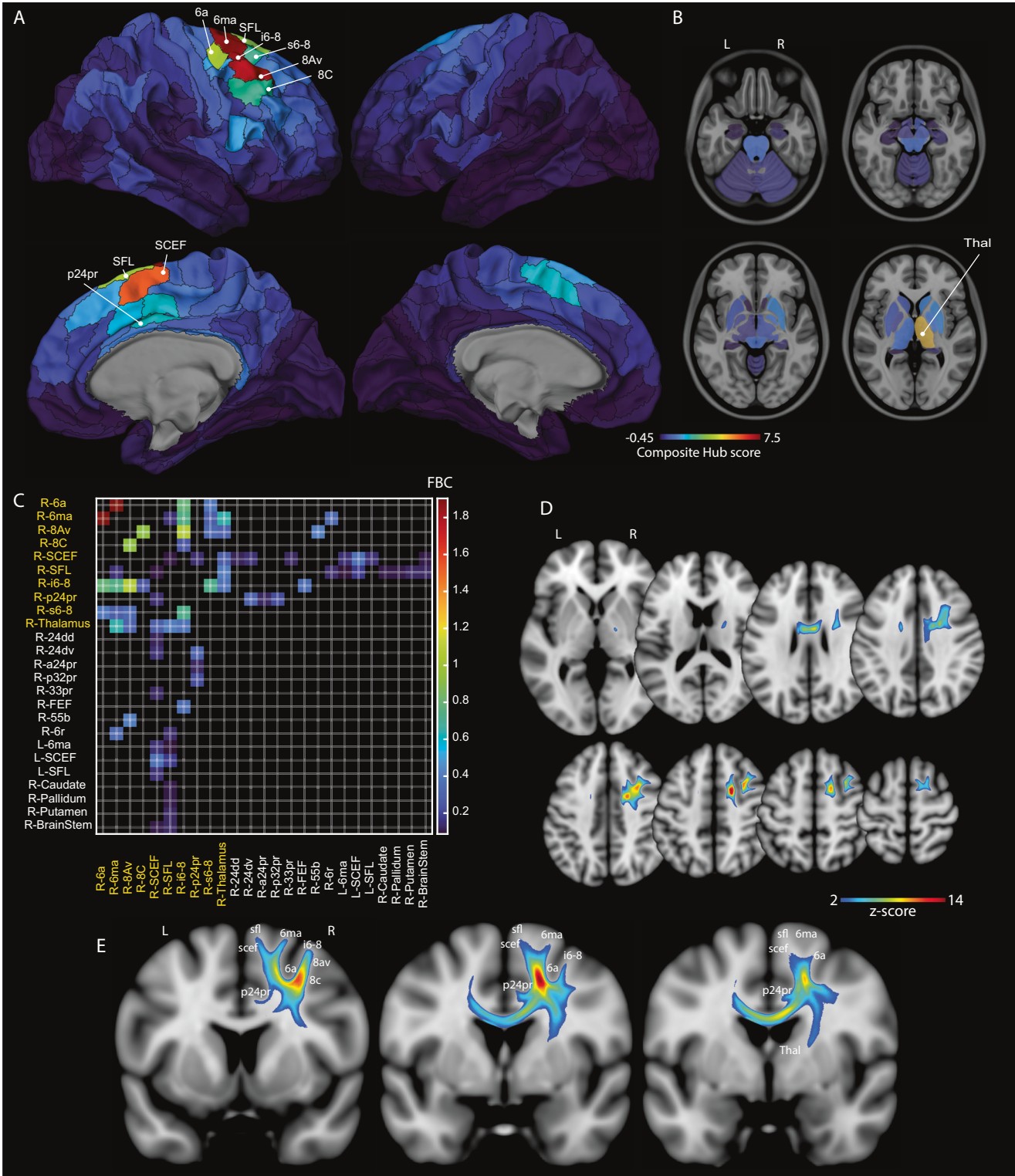

**Fig. 3 | Normative Structural Connectome of the Visuospatial Neglect Network (*n* = 100 HCP subjects).** All cortical (**A**) and subcortical (**B**) nodes are represented. Consensus hubs identified via graph theory analysis are labelled. The colour bar indicates the strength of the composite hub Z-score. **C** Connectivity matrix showing the relative Fibre Bundle Capacity (FBC) for the 33 consensus top edges (consensus hubs in yellow). **D, E** Population-averaged track-density image (TDI) visualising the core white matter pathways of the neglect network. The pathways with the highest fibre density (*Z* > 2) are localised in the white matter below the superior and middle frontal gyri, with extensions towards the corpus callosum and internal capsule. The network is shown in axial (**D**) and coronal (**E**) views. Source data are provided as a Source Data file.

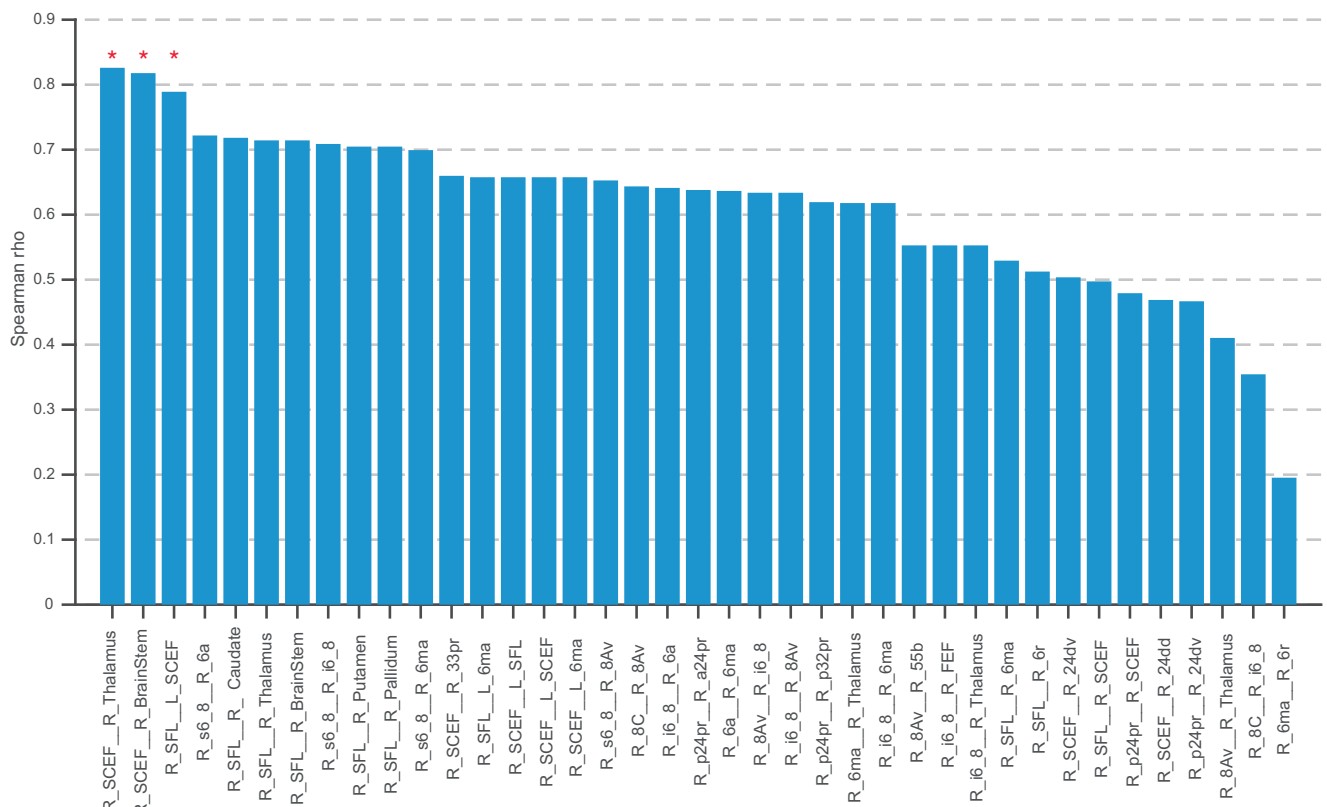

**Fig. 4 | Correlation between Consensus Edge Disconnection and Postoperative Neglect Severity.** The bar chart displays the Spearman's rank correlation (*rho*) between the percentage of surgical disconnection for each consensus edge and the severity of postoperative neglect (Δ Bells Test asymmetry score) in $n = 15$ patients with dMRI. Statistical analysis was performed using a two-sided Spearman's rank correlation. P-values were adjusted for multiple comparisons across 39 edges using the Bonferroni correction, with an adjusted $p < 0.05$ considered statistically significant. The three connections that showed a significant positive correlation are indicated by red asterisks: the right Supplementary and Cingulate Eye Field (SCEF) to the right thalamus ($\rho = 0.826$, Bonferroni-adjusted $p = 0.0057724$), the right SCEF to the brainstem ($\rho = 0.8178$, Bonferroni-adjusted $p = 0.0076243$), and the transcallosal connection between the right Superior Frontal Language (SFL) and the left SCEF ($\rho = 0.78911$, Bonferroni-adjusted $p = 0.018318$). Source data are provided as a Source Data file.

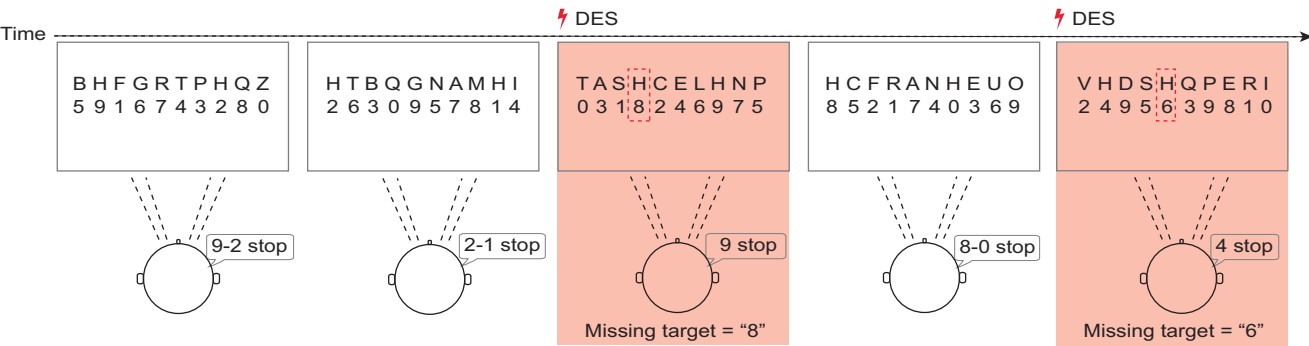

**Fig. 5 | Intraoperative visuospatial selective attention task (iVSAT).** Patients visually scanned letter arrays to identify two 'H' targets and read aloud the corresponding numbers below. The timeline illustrates correct performance on baseline trials (white boxes) and target omissions induced by Direct Electrical Stimulation (DES) (red boxes).

supporting hand-object interaction and used as the posterior boundary of the resection[22].

In the right hemisphere, 77 (mean per patient: 2.8 sites) iVSAT eloquent sites were found (Fig. 6A). Stimulation of 15 sites (13 patients) produced involuntary contralateral eye movements (Fig. 6A yellow sites). Stimulation of 62 sites produced target omission during iVSAT performance ($n$ of patients 28, mean per patient 2.2) (Fig. 6A red sites). 69% of iVSAT errors (missed target) were located in the contralesional (left) visual field. In Fig. 6B, C the distribution of all eloquent sites based on the error lateralisation index is depicted. Stimulations at the level of the SFG, below the SMA/preSMA

transition produced 93.1% of errors in the left hemifield and 6.9% in the right. When stimulation was applied below the MFG 66.7% of errors occurred in the left hemifield and 33.3% in the right hemifield, while stimulation below the IFG produced 53.3% of errors in the left hemifield and 46.7% in the right.

In the left hemisphere, DES during the iVSAT failed to produce selective attention errors (i.e., target omission). However, task-unrelated involuntary contralateral eye movements (11 eloquent sites, 8 patients) were elicited by DES.

The sites detected during iVSAT performance were also checked for language, motor, and non-visuospatial attention tasks errors, and

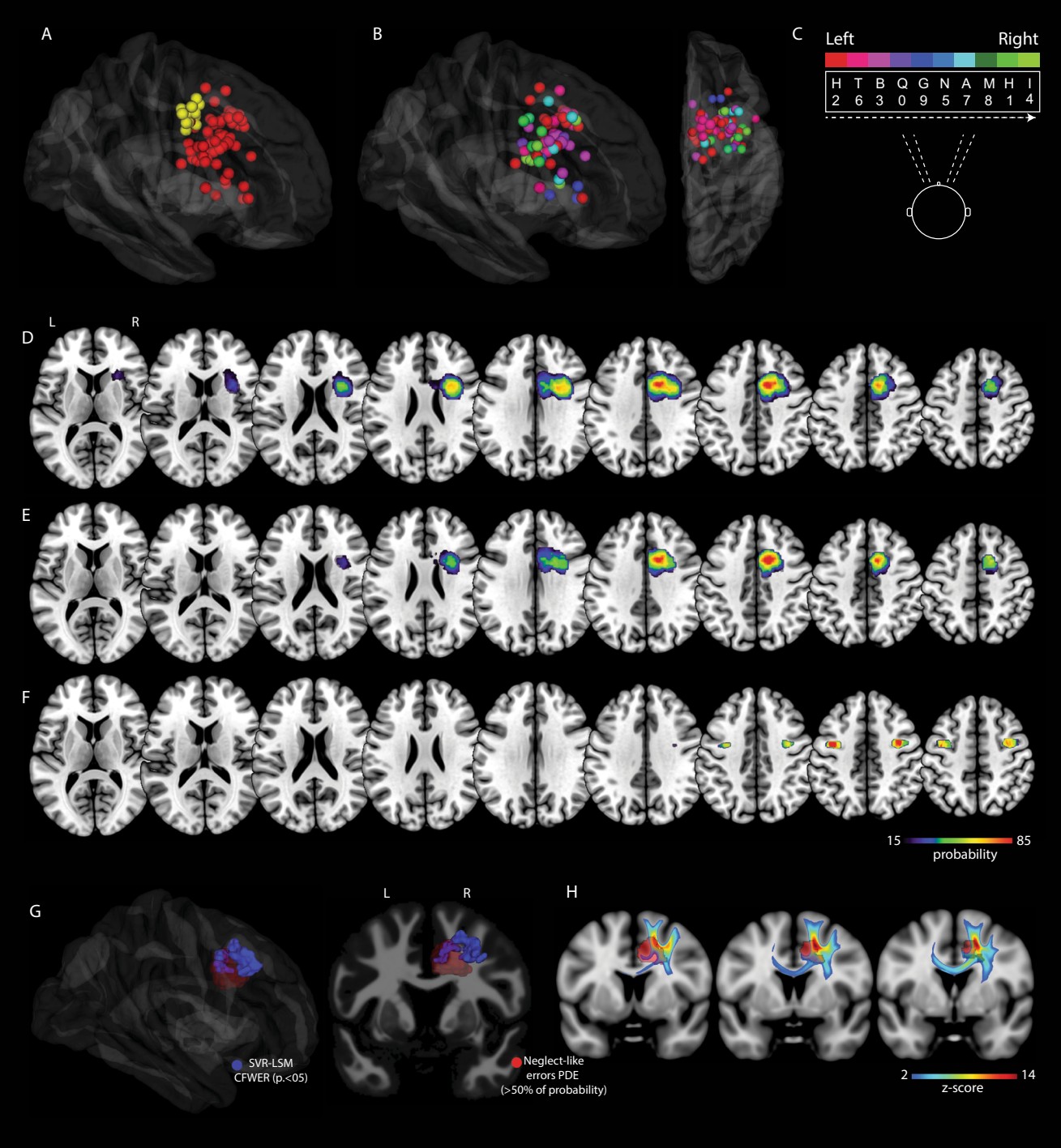

**Fig. 6 | Intraoperative mapping results from the prospective study.**
**A** Subcortical eloquent sites producing target omissions (red) or involuntary contralateral eye movements (yellow) during the Intraoperative Visuospatial Selective Attention Task (iVSAT) (left hemisphere $n = 19$; right hemisphere $n = 28$).
**B**, **C** Distribution and colour-coding of iVSAT errors based on a lateralisation index.
**D–F** Probability density estimation (PDE) maps reveal the spatial probability of sites causing any iVSAT error (**D**), left-sided neglect-like errors (**E**), and contralateral eye movements (**F**). Spatial overlap between the PDE map for neglect-like errors (red, >50% probability) and: - (**G**) the SVR-LSM cluster associated with visuospatial neglect (blue) (CFWER, $p = 0.05$, v = 1); - (**H**) = the core white matter pathways of the normative neglect network, visualised via track-density imaging (TDI) (Z > 2).

no overlap was observed. All the sites detected during iVSAT performance were spared from resection. When eloquent iVSAT sites were stimulated with high frequency DES (To5) up to 15 mA, no motor evoked potentials were evoked in any of the recorded effectors (face, upper- and lower-limbs), excluding the occurrence of motor interferences during the iVSAT errors.

**Probability density estimation.** Anatomical localization of eloquent iVSAT sites was computed by means of probability density estimation[25]. Independently from the lateralisation index, iVSAT errors occurred in a portion of white matter below the right SFG, MFG, and IFG (Fig. 6D). The highest probability of evoking neglect-like error was associated with the white matter below the transition between the

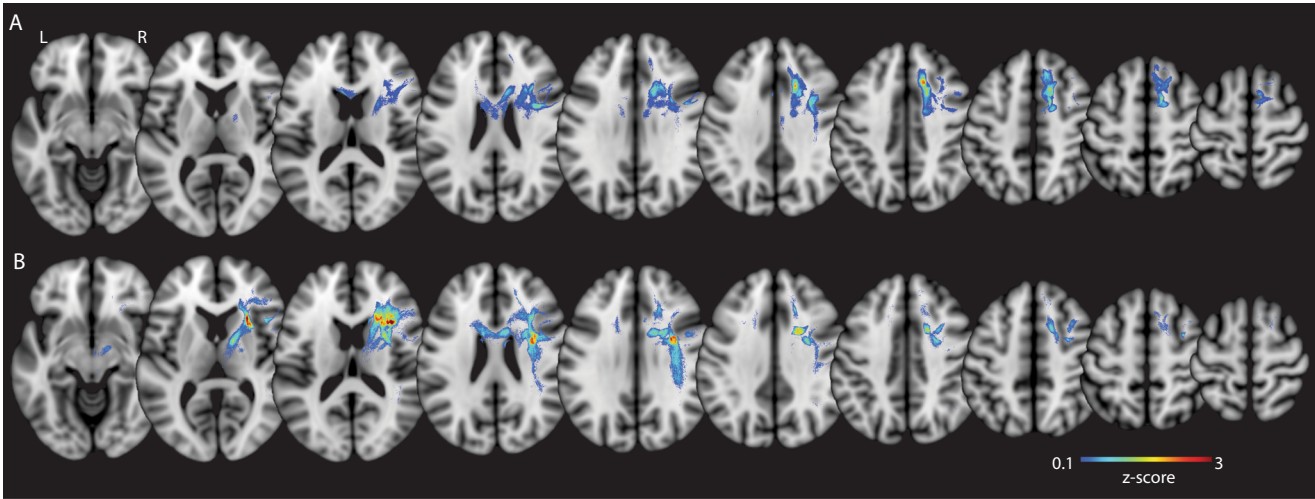

**Fig. 7 | Average Track density imaging (TDI) maps of Intraoperative visuospatial selective attention task (iVSAT) eloquent sites (prospective study; *n* = 9 patients with dMRI).** The images display the average apparent fibre density (z-score) of tracts seeded from eloquent iVSAT sites in nine patients from the prospective study. **A** The average TDI map of pathways associated with stimulation sites that induced neglect-like (contralesional) errors. **B** The average TDI map of pathways associated with stimulation sites that induced right-sided (ipsilesional) errors.

right SMA-proper and the pre-SMA and the grey matter of the mid-cingulate gyrus. Moreover, neglect-like errors were also associated with the white matter below the right MFG and IFG (Fig. 6E). Finally, stimulations producing contralateral eye movements (the only response induced in both hemispheres) clustered in a region corresponding to the left and right frontal eye fields, more posterior/lateral compared to neglect-like errors (Fig. 6F). Regarding the convergence between intraoperative findings and lesion-symptom mapping, the right hemisphere PDE of sites eliciting 'neglect-like' errors (highest probability, >50%) showed substantial spatial overlap with the significant SVR-LSM cluster associated with visuospatial neglect. This overlap was particularly prominent in the grey matter of the mid-cingulate gyrus and in the white matter below the SMA / preSMA transition (Fig. 6G). Moreover, within the different white matter regions highlighted by the population-averaged TDI map derived from HCP data generated using the SVR-LSM cluster as seed, its spatial convergence with the PDE of neglect like errors confirmed the crucial role of the pathways specifically below the SFG at the SMA / preSMA boundary (overlap visualised in Fig. 6H).

**Identification of stimulated white matter tracts.** In the subset of patients with preoperative dMRI (n. 9), 18 eloquent iVSAT sites were found. In 13 sites DES induced neglect-like errors, while in 5 sites the omissions were localised in the right hemifield.

To identify the structural connections underpinning these distinct DES-related errors, we generated SIFT2-weighted track-density imaging (TDI) maps seeded from each eloquent site and computed average maps for each error type (neglect-like vs right-sided errors). The average TDI map for neglect-like errors revealed a high fibre density in the white matter below the SFG and MFG. These pathways showed robust medial projections into the corpus callosum and inferiorly toward the internal capsule, a pattern highly consistent with the population-average TDI map derived from the normative HCP data associated with visuospatial neglect (Fig. 7A).

In contrast, the average TDI map for right-sided errors demonstrated a peak fibre density in more lateral white matter regions, primarily situated deep to the MFG and IFG, with the highest density located adjacent to the head of the caudate nucleus and below the IFG. Although this network also contained projections towards the corpus callosum and internal capsule, it was uniquely distinguished by a significant lateral fronto-parietal component (Fig. 7B).

**Effect of Intraoperative Visuospatial Selective Attention Task (iVSAT) on Postoperative Visuospatial Neuropsychological Outcomes.** To evaluate whether iVSAT mapping influences postoperative visuospatial performance, we compared pre- and postoperative total scores and asymmetry scores in the prospective cohort. For the total score the main effect of Time was not significant, $F(1, 45) = 0.10$, $p = 0.761$, $\eta^2_p = 0.002$, 95% CI [0.000, 0.102], indicating that mean performance was not statistically different at the two assessments. The Time × Hemisphere interaction was likewise non-significant, $F(1, 45) = 0.003$, $p = 0.950$, $\eta^2_p = 0.000$, 95% CI [0.000, 0.107], showing that patients with left- and right-hemisphere lesions exhibited the same trend (Supplementary Fig. 5). Similarly, for the asymmetry score, the main effect of Time was null, $F(1, 45) = 0.002$, $p = 0.967$, $\eta^2_p = 0.000$, 95% CI [0.000, 0.110], indicating that average performance was identical at both measurements. Likewise, the Time × Hemisphere interaction was non-significant, $F(1, 45) = 0.046$, $p = 0.831$, $\eta^2_p = 0.001$, 95% CI [0.000, 0.110], showing that left- and right-hemisphere groups followed the same trajectory (Supplementary Fig. 6). Due to significant deviation from normality (Shapiro–Wilk $p < .001$), Wilcoxon signed-rank tests were used to assess within-group pre–post changes. In left-hemisphere patients, there was no significant change in total score (Z = −0.09, $p = 0.929$, r_rb = 0.015, 95% CI [−0.618, 0.667]) or asymmetry (Z = −0.28, $p = 0.779$, r_rb = −0.083, 95% CI [−0.821, 0.733]). Similarly, right-hemisphere patients showed no significant pre–post differences in total score (Z = −0.14, $p = 0.887$, r_rb = 0.026, 95% CI [−0.571, 0.600]) or asymmetry (Z = −0.12, $p = 0.906$, r_rb = −0.026, 95% CI [−0.719, 0.600]) (see Supplementary Fig. 5/6). Importantly, none of the patients with right hemisphere damage exhibited postoperative neglect one month after surgery.

## Discussion

In this study, we present a convergent causal mapping approach that integrates direct electrical stimulation with machine learning-based multivariate lesion-symptom mapping on brain tumour patients, to identify brain regions involved in visuospatial exploratory/selective attention. This approach indicated that the lesion of the grey matter within the right posterior superior frontal sulcus / mid-cingulate gyrus and of the associated white matter are linked to the onset of visuospatial neglect. Crucially, DES applied on the same frontomedial regions during awake surgery in the right hemisphere produced transient visuospatial exploratory/selective attention interferences in the

contralateral hemifield (neglect-like errors). Instead, stimulation of more ventrolateral white matter below the MFG and IFG was associated with errors in both hemifields. Interestingly, no visuospatial impairments were associated with the resection of any specific left hemisphere region; moreover, DES applied to the left frontal white matter did not disrupt visuospatial exploratory/selective attention.

While prior DES studies established the causal role of a right-lateralized network in neglect, they predominantly focused on posterior parietal regions using the line bisection task[36–38], designed to probe the perceptual component of neglect[17,39]. By requiring patients to actively search for targets among distractors, our iVSAT enabled us to directly test the selective and exploratory components of attention. We found a critical spatial colocalization between the SVR-LSM cluster associated with visuospatial neglect and the intraoperative DES sites that elicited neglect-like errors. This convergence is anatomically substantiated by the strong spatial correspondence between the probability density map of neglect-like error sites and both the mid-cingulate gyrus LSM cluster at the cortical level, and the core pathways of the normative TDI map within the white matter beneath the SFG.

Our first set of evidence relates to the analysis of the role of right frontal dorsomedial circuitry in visuospatial exploratory/selective attention. Following surgery, while a substantial decline in the global attentional performance was observed in patients with either left or right hemisphere lesions, only right hemisphere resections were associated with a significant number of contralateral omissions (i.e., neglect-related errors). Consistently, the SVR-LSM analysis revealed significant voxels associated with visuospatial neglect exclusively in the right hemisphere. The clusters highlighted by the SVR-LSM involved, at the cortical level, areas lying along the superior frontal sulcus and the middle cingulate cortex. Subcortically, this cluster extended to white matter regions below the MFG and the SFG at the level of the SMA/pre-SMA transition. These findings align with recent lesion studies in glioma patients, demonstrating that lesions of frontal dorsomedial tracts lead to deficits in visuospatial exploratory/selective attention and contribute to the manifestation of visuomotor neglect syndrome[17,18]. With the present study, we moved beyond the identification of a priori white matter tracts, providing a quantitative, region-to-region connectomic blueprint of the network causally linked to visuospatial neglect, isolating the involvement of specific cortical and subcortical parcels. Using a population-averaged, SIFT2-weighted fibre density template derived from HCP normative data, we showed that the anatomical core of this network resides in the white matter beneath the superior and middle frontal gyri. This high-density core comprises dense local intra-frontal connections and gives rise to projections extending medially into the corpus callosum and inferiorly toward the internal capsule. A graph-theory analysis of this network architecture further revealed its topological organization, identifying ten right-hemisphere hubs with the highest centrality. These hubs are grouped into three distinct anatomical clusters: (1) a medial SFG–cingulate complex including SMA sub-regions and the supplementary motor and cingulate eye fields; (2) a more lateral premotor–dorsolateral frontal strip along the posterior SFG/MFG; and (3) a subcortical hub in the thalamus. The network's most robust connections ('consensus edges') stemming from the consensus hubs further structured this system, comprising extensive intra-frontal cortico-cortical, cortico-subcortical loops to the thalamus, basal ganglia, and the brainstem, and transcallosal fibres. This connectomic fingerprint delineates a multi-component system, providing a granular, network-based description of the white matter associated with the emergence of visuospatial neglect after neurosurgical frontal resections.

Having defined the network's normative architecture, we next sought to validate the clinical relevance of its core connections. To achieve this, we leveraged patient-specific preoperative dMRI data ($n = 15$) to test whether the surgical disconnection of the network's 'consensus edges' correlated with postoperative neglect severity. This approach allowed us to pinpoint critical connections while inherently accounting for inter-subject anatomical variability and potential tumour-induced distortions. This analysis revealed that visuospatial neglect was most significantly associated with the disconnection of three specific pathways: a projection between the right Supplementary and Cingulate Eye Field (SCEF) and the right thalamus, another from the right SCEF to the brainstem, and a transcallosal connection between the right Superior Frontal Language area (SFL) and the left SCEF.

The disruption of the right SCEF-thalamus pathway aligns with established models where the thalamus is a central node in cortico-basal ganglia-thalamic loops supporting attentional orienting. Our results are in agreement with non-human primates' electrophysiological studies, showing that these loops support saccade generation and the orienting of visuospatial attention[40,41]. Within this framework, the basal ganglia circuitry regulates the initiation and suppression of saccades via inhibitory and disinhibitory pathways to the superior colliculus[41]. The thalamus, in turn, receives input from the basal ganglia and projects back to frontal areas like the SCEF, closing the loop and allowing for the integration of signals necessary for voluntary, goal-directed saccades[42,43]. Disconnecting the thalamic link in this circuit would therefore impair the top-down modulation of oculomotor control. Consistently, human lesion studies show that hemineglect is associated with unilateral basal ganglia and thalamic damage[44,45]. Furthermore, the association between neglect and the disconnection of the right SCEF-brainstem pathway points to the disruption of a direct command signal for oculomotor control. The brainstem contains the final common pathways for generating saccades and orienting movements[46]. The correlation we observed suggests that damage to this descending frontal projection compromises the executive control over these fundamental brainstem mechanisms, a finding consistent with recent evidence linking pontine lesions to neglect syndromes[47].

Finally, the critical role of the transcallosal SFL-SCEF connections highlights the role of interhemispheric integration. However, if disrupting these fibres directly caused the syndrome, we would expect resection of this bilateral connection from either hemisphere to produce a similar deficit. However, our SVR-LSM analysis found a strong correlation between disconnection and neglect in right-hemisphere patients, but no significant association in left-hemisphere patients. This finding, combined with evidence that callosotomy rarely causes neglect in isolation[48], argues against the hypothesis that transcallosal disconnection alone is sufficient to explain the onset of neglect in our patients. An alternative explanation for our findings posits that when right-sided attentional systems are compromised, the concurrent disruption of callosal fibres prevents left-hemispheric compensation, thereby worsening the severity of neglect[49].

Interestingly, our findings provide critical insights into the different roles in visuospatial exploratory/selective attention played by the frontal eye fields, the frontal dorsomedial regions (SMA / pre-SMA boundary) and the frontal ventrolateral regions (MFG and IFG). First, neglect-like errors related to the DES of the dorsomedial white matter occurred without evoking any observable eye movements. The latter were clearly evoked by stimulation of a sector corresponding to the frontal eye fields (FEFs). The different outcomes produced by DES when applied to dorsomedial white matter (i.e., neglect-like errors) and to the FEF area (i.e., contralateral eye deviations) might suggest that the two sectors have a different but complementary involvement in visuospatial selective attention. In fact, while the FEF have been described as specialised in the transformation of visual stimulus location into saccadic commands[50], in virtue of specific direct and indirect descending connections to the SC[51], the dorsomedial iVSAT cluster might represent a pre-oculomotor region supporting

programming of goal-related oculomotor movements required by the intraoperative task execution.

Moreover, LSM results highlighted two cortical clusters including the posterior superior frontal sulcus and the SFG medial wall/middle cingulate gyrus. Our lesion-load analysis suggested that the SVR-LSM cluster is embedded within key large-scale functional systems. Although the Dorsal Attention Network (DAN) was proportionally the most affected, the cluster also impacted the Fronto-Parietal Control (FPCN) and Salience/Ventral Attention (VAN) networks. We speculate that this region might function as an integrative pre-oculomotor hub. Here, the DAN's machinery for voluntary attention deployment is guided by top-down goals from the FPCN's DAN-linked subsystem, which ensures focus on task-relevant information[52]. Simultaneously, this hub might integrate vigilance and arousal signals from the VAN, creating a balanced system for effective spatial exploration.

When DES was applied to a more ventrolateral subcortical sector below the MFG and IFG, again, no involuntary eye movements were evoked. However, while neglect-like errors were evoked by stimulation of the dorsomedial white matter, DES applied on this sector was indeed associated with target omissions in both left and right hemifields. This evidence is in agreement with the 'standard' theory of neglect[4], which postulates that the right hemisphere controls shift of attention to both sides of the visual field, while the left hemisphere controls attention to the right side. According to this framework, the inefficacy of DES during the iVSAT in the left hemisphere might be explained by the compensatory activity played by the right one. Consistent with the DAN/VAN model[8,9], our results support the role of the right dorsomedial connectivity in contralateral attention deployment. Moreover, the occurrence of attentional errors in both hemifields during DES of ventrolateral white matter might be coherent with VAN's non-spatial role in phasic alerting and rapid target detection[9].

However, an apparent discrepancy emerged between this last intraoperative finding and postoperative outcome: resection of these same ventrolateral regions was not associated with a specific deficit on the Bells Test. This mismatch does not necessarily imply functional resilience but might reflect the differential sensitivity of the two tasks to network disruption. The iVSAT, which demands rapid detection of infrequent targets among distractors, might engage more the stimulus-driven functions of the VAN compared to the Bells Test, which is untimed, self-paced and barely requires fast target detection. Therefore, the absence of a postoperative deficit on the Bells Test should not be overinterpreted as evidence against the role of these ventrolateral pathways in exploratory/selective attention. Instead, it might indicate that our standard assessment was not sensitive to capture the specific VAN-related functions—such as alerting and vigilance—that were transiently possibly disrupted by DES during the iVSAT. While future studies should ideally employ a more unified testing approach, implementing a comprehensive battery of tasks is challenging given the clinical constraints of the intraoperative neurosurgical setting. The development of tasks that can efficiently and systematically probe distinct attentional components (e.g., exploration, vigilance, reorienting) under these demanding conditions remains a critical next step. Such an approach would enable a more direct correlation between intraoperative and postoperative findings, helping to delineate more granular functional dissociations within the frontal attention networks.

A final issue to be discussed regards the LSM studies in strokes classically reporting the disconnection of the VAN associated with the occurrence of neglect-syndrome[53]. The fronto-parietal VAN/DAN model[8] suggests that a deficit in the ventral system may lead to hypo-activation of the dorsal one, possibly resulting in a relative over-activation of the left DAN, leading to an attentional bias toward the right hemifield (i.e visuospatial neglect). Differently, in our study, resection of VAN connectivity was not associated with visuospatial

neglect, an apparent conflicting evidence related to data in stroke studies. However, it was observed that damage of the parietal component of SLF II and III, compared to their frontal terminations, is the most important factor predicting the severity of the impairment[49,54]. This may explain why, in this study, which involved predominantly frontal lobe resections, no association was observed between lateral fronto-parietal pathways and visuospatial neglect. An alternative explanation may be related to the plasticity occurring in brain tumours compared to strokes, possibly leading to an altered balance between the VAN and the DAN. This hypothesis draws upon the considerable rewiring potential shown by large-scale fronto-parietal hetero-modal networks[55–57]. However, the discrepancy between our results and the stroke literature cannot be conclusively addressed here due to the different aetiology preventing a direct comparison[58].

A significant limitation of our study is the relatively small number of patients with parietal lobe involvement due to the lower incidence of gliomas compared to more frequently affected regions, such as the insular and frontal lobes. In fact, while our SVR beta-map highlights the possible involvement of medial superior parietal regions in the onset of visuospatial neglect, this association did not achieve statistical significance. This limitation, in our study, could produce an underestimation of the role of parietal regions. As a consequence, our results should be confined to the role of the frontal lobe connections in visuospatial exploratory/selective attention. Moreover, while the absence of intraoperative iVSAT errors during mapping of the left hemisphere could be justified in light of the right hemisphere's compensatory role, it should be acknowledged that, in the left hemisphere, functional borders defined with language mapping do not always allow for extensive visuospatial mapping as in the right hemisphere. A further limitation lies in the relatively early timing of the post-surgical assessment, which may not have captured long-term outcomes. However, this choice was deliberate to ensure a more accurate measure of the direct effects of surgery on visuo-spatial abilities. At later time points, neuroplasticity mechanisms may lead to functional compensation, making it difficult to attribute deficits solely to the resection[59,60]. Additionally, many patients undergo adjuvant treatments such as radiotherapy and chemotherapy after the first month post-surgery, which have well-documented cognitive effects[61]. These factors could confound the interpretation of long-term cognitive outcomes. Future studies could explore longer follow-ups while carefully controlling for these confounding variables.

To conclude, our study provides convergent causal evidence that the integrity of a right dorsomedial frontal network, centred on the posterior SFG and middle cingulate gyrus, is critical for visuospatial exploratory/selective attention. We demonstrated that resection of this region is associated with neglect, and crucially, that intraoperative stimulation of its underlying white matter elicits contralateral neglect-like errors. Based on this convergent causal evidence, future research should aim to evaluate whether this convergent substrate offers an effective target for neuromodulation therapies for visuospatial neglect. Additionally, stimulation of lateral frontal white matter produced errors in both left and right hemifields. This data aligns with the proposed role of the right hemisphere in controlling attention towards both the left and right hemifields[4]. Consistently with the DAN/VAN model, our data support the role of the right dorsomedial connectivity in contralateral attention deployment and the notion of a non-spatial function of the VAN[8,9]. Moreover, our data extends the classical DAN/VAN model to the posterior superior frontal sulcus and the middle cingulate gyrus, possibly favouring the integration of VAN-related computations (e.g., vigilance and arousal) with DAN-related functions (voluntary deployment of attention and saccadic control). Finally, from a clinical standpoint, we provided an intraoperative visuospatial attention task (iVSAT) to detect and help preserve frontal connectivity supporting visuospatial exploratory/selective attention.

## Methods

### Retrospective study

**Patient cohort.** 171 patients who underwent a neurosurgical resection for brain tumour removal were screened for inclusion in the retrospective study. Patients provided formal consent for the procedures, and the study was approved by the local ethics committee (L2093), in accordance with the principles outlined in the Declaration of Helsinki. The study was performed with strict adherence to the clinical procedure for tumour removal. Written informed consent was obtained from all subjects involved in the study for the surgical procedure and the subsequent use of their fully anonymized data for research and publication purposes. Sex of patients is reported in the Supplementary Information. Analyses were not performed separately for this variable, as the study is not designed to investigate sex-based differences. Patients with pre- and post-operative visual deficits (hemianopia or quadrantopia), motor or comprehension deficits, previous neurosurgical treatment, or adjuvant radiotherapy before surgery were excluded from the study. All patients underwent a comprehensive neurosurgical assessment, a magnetic resonance imaging (MRI), and a neuropsychological assessment both before the surgery and one month postoperatively.

**Intraoperative brain mapping and monitoring.** All surgeries were performed in asleep-awake-asleep anaesthesia with the aid of brain mapping and monitoring techniques[62]. The craniotomy exposed the tumour area and a limited portion of the surrounding cortex. For cortical and subcortical motor mapping, high-frequency DES (HF-DES) was used to identify and preserve the corticospinal system[62,63]. To identify and preserve sites producing interference on language, motor/praxis[20,62–65], visual[66], and executive functions[21,67], Low-Frequency DES was used during the awake phase of the procedure. For further details on motor mapping and monitoring, see the Supplementary Materials.

**Neuropsychological assessment and statistical analysis.** In each patient, neuropsychological performance in the visuospatial attentive domain was assessed one week before and at one month after surgery. The Bells test (BT) was used to assess visuospatial exploratory/selective attention performance: it requires the patient to find and cancel 35 targets (bells) embedded among 280 distractors (houses, horses, etc.). BT allows for a quantitative and qualitative assessment of visual neglect[68]. Specifically, the total score (i.e., number of omitted targets) is a measure of global visuospatial exploratory/selective attention performance, while the asymmetry score (i.e., the difference between the number of omitted bells on the left side and the number of omitted bells on the right side) is considered a measure of visuospatial neglect. Scores were compared with Normative Italian data to classify pathological patients[26]. The task performance at the different time points (preoperative and postoperative) was compared (see Statistical analysis).

**MR data.** As part of the clinical routine, pre- and postoperative MRI were performed on a 3 T scanner and acquired for lesion morphological characterization and volumetric assessment. A post-contrast gadolinium T1-MPRAGE sequence was performed using the following parameters: echo time: 2.75 ms, repetition time: 1600 ms, flip angle 9°, inversion time 900 ms; 176 slices; isotropic voxel size of 1 mm. A spin-echo, single-shot echo-planar imaging diffusion sequence was acquired in a subset of subjects (15 patients in the retrospective study, 9 patients in the prospective one). 73 volumes were collected: 64 diffusion-weighted volumes at a $b$-value of 2000 s/mm² and 9 interleaved non-diffusion-weighted ($b = 0$ s/mm²) volumes in the AP phase encoding direction. Repetition time = 16.9 s; echo time = 96 ms. A GRAPPA acceleration (parallel reduction factor = 2) was employed.

Data were acquired with an isotropic voxel size of 2 mm³ (matrix = 128 × 128, 64 slices, slice thickness = 2 mm) and a flip angle of 90°.

Resection cavities were delineated on the postoperative MR image by the co-first author (L.V.) using ITK-SNAP, and both registered to a common template by means of lesion masking approach using the Clinical Toolbox in SPM (enantiomorphic normalization). Reliability of the normalization process was visually inspected case-by-case by using SPM checkReg function.

**Lesion symptom mapping.** To assess the association between postoperative attentional deficits and the resection of specific brain regions, multivariate lesion-symptom mapping (LSM) was performed. Separate analyses were conducted for patients with left hemisphere lesions ($n = 81$) and right hemisphere lesions ($n = 82$) to maximise statistical power, as analyzing unilateral lesions together can otherwise weaken the observed associations within each hemisphere[69–71]. The dependent variables for these analyses were the Δ total score (difference between pre- and 1-month postoperative total number of omitted targets) and the Δ asymmetry score (difference between pre- and 1-month postoperative asymmetry score) of the Bells Test. A support vector regression lesion-symptom mapping (SVR-LSM) approach was employed using a MATLAB toolbox[72], leveraging the MATLAB Statistics and Machine Learning Toolbox; a non-linear radial basis function (Gaussian) kernel was utilised for the SVR. To account for potential influences of sociodemographic factors and tumour characteristics, patient age, years of education, and tumour grade were included as covariates in the SVR-LSM models. Optimisation of hyperparameters was performed via resubstitution loss and Bayesian optimisation with 200 iterations and 5-fold cross-validation. In addition, the range for the optimised parameters was set following the range of $C$ and *Gamma* suggested by ref. 73 and more recently adopted in different studies[20,74]. $C$ range = 1–80, *Gamma* equivalent *Sigma* range = 0.1–30. A default *Epsilon* range was set. For each analysis and combination of parameters selected after the optimisation procedure, both prediction accuracy and reproducibility were evaluated. Based on studies using a similar procedure[20,75,76], the LSM results were considered reliable when showing accuracy ≥0.25 and reproducibility ≥ 0.85. For each hemispheric analysis, only voxels resected in at least 10% of the respective patient group were included. Resection volume was controlled using the direct total lesion volume control (dTLVC)[73]. A voxelwise thresholding was applied after generating SVR β-maps with 5000 permutations ($P < 0.005$). To correct for multiple comparisons, a cluster-level family-wise error correction was applied (CFWER, $P = 0.05$, $\upsilon = 1$).

SVR-LSM was used with continuous behavioural scores as dependent variables to preserve the full variance in the data and to maximise statistical power[72,73]. To examine the concordance between our SVR-LSM findings and a traditional clinical-categorical approach, we also generated a statistical resection overlap map using only the subgroup of patients with a postoperative pathological score and assessed its spatial overlap with the multivariate results.

To characterize the large-scale functional network affiliations of the brain regions identified by our SVR-LSM, we performed a parcel-level lesion load analysis. The statistically significant SVR-LSM cluster (a binary mask after family-wise error correction) was used as input for the Lesion Quantification Toolkit[77]. We calculated the overlap of this ROI with the resting-state functional MRI cortical parcellation (17 networks / 1000 parcels)[27]. We selected the 1000-parcel resolution to achieve a fine-grained analysis capable of precisely localising our focal SVR-LSM effects within the architecture of the 17 larger networks. This approach acknowledges the functional heterogeneity within broad cortical territories and allows for a more detailed anatomical characterization, balancing high resolution with interpretation at the level of established large-scale systems[27]. Furthermore, to provide a more direct measure of the proportional impact on each large-scale system,

we also calculated the percentage of each of the 17 entire networks' total volume that was overlapped by the SVR-LSM cluster.

**Normative Visuospatial Neglect Structural Connectome.** To define the normative structural connectome of the neglect-associated region, we generated region-to-region connectivity matrices for the white matter passing through the significant SVR-LSM cluster. The Human Connectome Project (HCP) 100 unrelated subjects diffusion-weighted MRI (dMRI) dataset was used[28]. These pre-processed multi-shell data, acquired on a Siemens 3 T scanner, had already undergone corrections for motion, eddy current, and EPI distortions, and co-registration to each participant's T1w scan. Subsequent processing used MRTrix3 (https://www.mrtrix.org/) after B1 bias field correction (ANTs N4[78]).

**Streamlines tractography analysis.** Tractography analysis followed a protocol for structural connectome construction using constrained spherical deconvolution in multi-shell diffusion-weighted imaging recently described by Tahedl et al.[79]. From the pre-processed HCP dMRI acquisitions, we estimated response functions (RFs) for white matter (WM), grey matter (GM), and cerebrospinal fluid (CSF) using the Dhollander unsupervised algorithm[80]. To enhance group-level comparability, these subject-specific RFs were averaged, and the mean RFs were used to calculate fibre orientation distributions (FODs) via multi-shell multi-tissue constrained spherical deconvolution (MSMT-CSD)[29]. The resulting FOD images underwent bias-field correction and global intensity normalization[81].

A five-tissue-type (5TT) segmentation image (comprising cortical GM, subcortical GM, WM, CSF) was generated from T1w images using Hybrid Surface and Volume Segmentation (HSVS)[82], providing anatomical priors for Anatomically-Constrained Tractography (ACT)[30]. Whole-brain ACT probabilistic tractography was performed using the iFOD2 algorithm[83] with back-tracking[30] and dynamic seeding[31] to generate 10 million streamlines per subject (step size=0.5×voxel size; max angle=45°; length=2.5–250 mm). The whole-brain tractograms were then filtered using Spherical-deconvolution Informed Filtering of Tractograms (SIFT2), which assigns a cross-sectional area multiplier to each streamline to ensure streamline density quantitatively matches the apparent fibre density from the FODs[30,31]. These SIFT2-weighted tractograms were filtered to retain only streamlines traversing the significant SVR-LSM cluster associated with visuospatial neglect.

Using FreeSurfer, the Human Connectome Project multimodal cortical parcellation (HCP-MMP1.0; 360 regions)[28] and the FreeSurfer ASEG subcortical atlas (19 regions, resulting in 379 total regions)[84] were mapped onto each subject's T1w image. These cortical and subcortical regions were combined to define a subject-specific atlas of 379 nodes[79]. Structural connectomes were then generated from the SIFT2-weighted tractograms, including only those connections traversing the SVR-LSM cluster. Connectome edges were quantified by summing the SIFT2 weights of streamlines connecting each pair of nodes, yielding the Fibre Bundle Capacity (FBC)–an estimate of the total intra-axonal cross-sectional area for each connection[32,79]. Resulting FBC values were then multiplied by the subject-specific SIFT2 proportionality coefficient ($\mu$), transforming them into an absolute measure (mm²) for quantitative inter-subject comparisons[32]. This yielded a 379×379 FBC matrix for each HCP subject representing the structural connectivity of the neglect-implicated network.

**Graph theory analysis.** To characterize the topological organization[85] of the neglect-associated network, graph theory analyses were performed on each subject's 379×379 FBC matrix including those connections passing through the SVR-LSM cluster, using the Brain Connectivity Toolbox (BCT)[86] in MATLAB. The analysis was performed on unthresholded matrices, a decision informed by the current lack of consensus regarding optimal thresholding methods for connectomic data[87]. This approach is further supported by the use of SIFT2 streamline filtering, as evidence indicates that for SIFT2-filtered connectomes, removing even the weakest ~70%–90% of connections does not significantly alter subsequent graph theory metrics[88], and that SIFT2 may be sufficient for robust connectome construction without additional graph thresholding[89]. For each node within individual subject networks, three key measures were computed: 1) node strength: the sum of FBC weights of all connections to a node, quantifying its overall connectivity capacity; 2) node degree: the number of connected nodes, indicating its direct interaction level with others nodes; 3) betweenness centrality (BC), the fraction of all shortest paths passing through a node, highlighting nodes crucial for efficient information flow[90–92]. To identify hub regions, these metrics were Z-scored within each subject, and a composite hub score was calculated by averaging the three Z-scores for each node[93]. 'Consensus hubs' were defined as nodes with a composite Z-score exceeding 1.5 in at least 50% of subjects.

To complement the connectivity analysis, the most important edges of each consensus hub ('consensus top edges') were identified. For each subject and for each node identified as a consensus hub, the FBC values of its connections were Z-scored (excluding self-loops). An edge was deemed consensus top edge if its FBC Z-score exceeded 1.5 in at least 50% of the subjects. Finally, this procedure allowed for the unsupervised/data-driven identification of most strategic regions (nodes) disconnected by the SVR-LSM cluster and their individual main connectivity (edge) with specific regions, overall quantitatively assessing for the main networks specifically associated with visuospatial neglect.

**Track density imaging.** To visualise the network's anatomical substrate, we generated track-density images (TDI)[33,34] for each subject. After filtering the whole-brain tractography by the SVR-LSM cluster, streamlines were non-linearly registered to the MNI152 template using an ANTs-based workflow[94] available through brainlife.io (app-ants-mni[95]). We then generated 0.2 mm isotropic maps, incorporating SIFT2 weights so that voxel intensity reflects apparent fibre density. Each map was scaled by the subject-specific proportionality coefficient $\mu$ to ensure absolute units across participants[32]. The individual maps were then z-scored and averaged to produce a population-average TDI map of the network's spatial distribution and density[96].

**Diffusion tractography in single patients.** Tractography in individual patients ($n = 15$) was performed on preoperative dMRI data using a pipeline adapted from our HCP workflow to accommodate single-shell clinical acquisitions and tumour signal (see also[97]). Raw images were first denoised with Marchenko–Pastur PCA[98,99] and corrected for Gibbs ringing[100]. Susceptibility and eddy current distortions were removed with the Synb0-DisCo synthetic b0 approach (due to single AP phase-encode)[101], followed by B1 bias-field correction using ANTs N4[78]. Response functions for white matter, grey matter, and CSF were estimated with the Dhollander algorithm[102], and their group averages were used in single-shell three-tissue CSD to compute fibre orientation distributions (FODs) from the $b = 2000$ s/mm² data[103] (instead of multi-shell multi-tissue CSD). FODs were then bias-field corrected and intensity-normalized[81]. Anatomically-constrained tractography[30] employed a five-tissue-type segmentation generated via HSVS[82] on each T1-weighted scan and registered to dMRI, augmented by a manually delineated tumour/oedema mask as a fifth tissue class to prevent premature streamline termination in lesioned regions[104,105]. Whole-brain probabilistic tractography (iFOD2; same parameters used for HCP data) generated 10 million streamlines per patient, which were subsequently filtered with SIFT2[31,32]. Finally, the same 379-node atlas used for the HCP cohort was mapped to each patient's T1w image after inpainting the pathological tissue with healthy-appearing tissue with SynthSR, to improve anatomical processing (segmentation and surface extraction)[106]. After registration of the 379-node atlas to dMRI, for

each patient, a preoperative 379×379 Fibre Bundle Capacity (FBC) matrix was generated using the SIFT2-weighted tractograms; these FBC values were then multiplied by the subject-specific SIFT2 proportionality coefficient ($\mu$), consistent with the HCP data processing, yielding FBC in absolute mm² units.

**Validation of consensus edge disconnection against postoperative deficits.** To validate the clinical relevance of the normative network's core pathways, we tested whether the surgical disconnection of its 'consensus edges' (see Graph theory analysis) correlated with postoperative neglect severity in our patient cohort ($n = 15$). This was achieved by quantifying the disconnection for each consensus edge using patient-specific preoperative dMRI and simulated postoperative tractography, an approach that inherently accounts for anatomical variability and pathology-induced tract distortions. The extent of white matter disconnection for these specific consensus edges was assessed via the following procedure: 1) Postoperative T1w images and resection cavities were registered to each patient's preoperative dMRI space via three-stage ANTs registration (Rigid, Affine, and SyN diffeomorphic)[94,107]. 2) The aligned resection cavities were used as exclusion ROIs to filter the preoperative SIFT2-weighted tractogram, creating a 'postoperatively simulated' tractogram. 3) A 'postoperative' FBC matrix was generated from this simulated tractogram. 4) The percentage of disconnection for each consensus edge was calculated as: (FBCpreop − FBCpostop)/FBCpreop * 100. 5) Finally, for each consensus edge, its percentage of disconnection was correlated across the 15 patients with their corresponding Bells test Δ asymmetry score using Spearman's rank correlation with Bonferroni corrections.

**Prospective study**

**Patients cohort.** In the prospective study, 50 patients undergoing brain tumour resection in awake anaesthesia were considered for inclusion. All participants gave written informed consent to the surgical mapping procedure (AIRC − 17482) and data analysis for research and publication purposes, following the principles outlined in the Declaration of Helsinki. Sex of patients is reported in the supplementary materials. Analyses were not performed separately for this variable, as the study is not designed to investigate sex-based differences. The study was conducted with strict adherence to the clinical procedure for tumour removal. The inclusion criteria were the same adopted in the retrospective study, except for one condition: patients presenting with preoperative visuospatial exploratory/selective attention deficits were excluded due to the potential impact of such deficits on the intraoperative administration and reliability of intraoperative attention assessment. Intraoperative brain mapping routine, MR acquisition, and neuropsychological assessment were identical to those applied in the retrospective study. In 9 patients, for clinical purposes, a preoperative dMRI acquisition was performed (see above MR Data and Diffusion tractography in single patients for further details on acquisition parameters and tractography reconstruction).

**Intraoperative assessment of visuospatial exploratory/selective attention.** To assess and preserve visuospatial exploratory/selective attention during awake procedures an adapted version of a classical visuospatial attention test[35] was used in a prospective series of 47 patients (28 right hemisphere) undergoing resection for a frontal lobe glioma in awake anaesthesia (Fig. 5). Considering the constraints related to the position required for the surgical procedure on the operating bed, visually guided upper limb movements cannot be reliably assessed during surgery. For this reason, the intraoperative version of the visuospatial selective attention test (iVSAT) consisted in a series of slides presenting 10 consecutive random letters containing 2 target letters ("H") among distractors ("B","C","N".). Beneath each letter were also presented 10 random single digits as a reference. Patients (pre-operatively trained) were asked to naturally explore the superior raw (letters) as fast as they could and to read aloud only the digits below the target letter (Fig. 5). To ensure that patients had completed the visual scanning, they were instructed to say 'stop' when they felt they had found all the targets. By using this approach, we ensured that the test maintained high specificity for assessing visuospatial exploratory/selective attention without interference due to motor control issues, as it relies on visual scanning and attention rather than visually guided hand movement. Intraoperative iVSAT mapping was performed using Low-Frequency Direct Electrical Stimulation (LF-DES). Stimulation consisted of biphasic square-wave pulses with a pulse width of 0.5 ms, delivered at 60 Hz in trains of 1–4 seconds using a bipolar stimulator probe with a 5 mm inter-electrode tip distance. While patients were performing the iVSAT task, LF-DES was applied at both the cortical and subcortical levels to identify sites where stimulation induced task interference. The current intensity for iVSAT mapping was individually titrated for each patient. Consistent with standard clinical practice for eloquent area mapping[63], this was set at the lowest current amplitude initially tested on the ventral premotor cortex that consistently elicited errors during a concurrent language task. This same current intensity was then utilised for all subsequent cortical and subcortical iVSAT mapping in that patient.

A stimulation site was considered positive for interference when an error, i.e., a missed target, occurred in three non-consecutive stimulation trials. To exclude language or motor interferences, the same sites were also tested with naming and praxis tasks. During the iVSAT, the patients' verbal responses were reported in real time to the neurosurgeon by the neuropsychologist, fully blinded to the neuroanatomical location of the stimulation site. The occurrence, during stimulation, of involuntary eye movements was behaviourally monitored and recorded. The location of sites of interference was recorded using neuronavigation (Curve, Brainlab AG) at the end of the subcortical mapping procedure before tumour debulking, to avoid possible shift. Subsequently, effective sites were drawn to the preoperative T1w as spherical rois with a 5 mm diameter, considering the resolution of the bipolar probe[108], based on image and coordinates acquired with the neuronavigation system. Videos of the intraoperative flaps during mapping were used to confirm the site localisation based on gyro-sulcal anatomy of the structure surrounding the probe. Finally, the stimulation sites, the preoperative T1w and the postoperative T1w (1-month follow-up) were coregistered to the MNI space by means of lesion masking approach (enantiomorphic normalization) using the Clinical Toolbox in SPM12.

**Intraoperative lateralisation score.** The analysis of participants' performance during the intraoperative visuospatial selective attention test (iVSAT) was focused on the spatial localization of errors. To assess the extent of lateralisation and any neglect-like tendencies, a specific scoring system was implemented. The score assigned to each missed target letter ("H") was based on its horizontal position within the visual field, with a score of 1 indicating an error at the far left (i.e., when the H was the first letter) and a score of 10 indicating an error at the far right (i.e., when the H was the last letter). Consequently, a lower score on the error scale indicated a more left-lateralized error pattern associated with a specific stimulation site. This method enables a precise assessment of whether DES applied on a specific positive site results in a more left- or right-lateralized pattern of errors.

**PDE of subcortical eloquent sites.** The region of highest probability of producing an iVSAT error was computed using a modified in-house version of probability kernel density estimation (PDE analysis) implemented in MATLAB[25], already used in previous studies[20,22,109]. Independently from the intraoperative lateralisation score, a PDE based on concentration of eloquent stimulation sites was used to disclose the region with the highest probability of finding any type of iVSAT error. Moreover, a second PDE was independently computed to highlight the

region with the highest probability of producing an error in the left visual field ("neglect-like errors). The intraoperative lateralisation score was used to select only the coordinates of the sites associated with a neglect-like error. These were defined as all omissions of a target letter among the first 5 items making up the trial. Finally, a third PDE was computed to study the localization of sites where DES elicited involuntary eye movements.

**Identification of stimulated white matter pathways.** Diffusion MRI (dMRI) processing for the prospective cohort mirrored that of the retrospective study (see Diffusion tractography in single patients). Each eloquent stimulation site, represented as a 5 mm spherical ROI, was used to filter the patient's preoperative SIFT2-weighted 10 million streamlines. From these filtered streamlines, track-density imaging (TDI) maps were generated using SIFT2 weights and multiplied by the subject-specific proportionality coefficient ($\mu$). TDI maps were z-scored. Two average TDI maps were then created, one for pathways associated with neglect-like errors and one for pathways associated with right-sided errors.

**Analysis of postoperative neuropsychological outcomes.** To assess the impact of iVSAT-guided resection on postoperative visuospatial exploratory/selective attention, we compared preoperative and one-month postoperative total and asymmetry scores from the Bells Test within the prospective cohort (see Statistical analysis).

**Statistical analysis.** All statistical analyses were performed using IBM SPSS 24.0 software. The assessment of behavioural scores at the different time points (preoperative and postoperative) was compared by a two-way repeated measure ANOVA using *hemisphere* as a between-factor and *time-point* as a within-factor. Post-hoc pairwise multiple comparisons analyses were conducted with paired t-tests. Correlational analysis was performed using the Spearman correlation coefficient. In case of deviation from normal distribution, the Wilcoxon signed-rank test was used to confirm the statistical significance.

### Reporting summary
Further information on research design is available in the Nature Portfolio Reporting Summary linked to this article.

## Data availability
The group-level neuroimaging results generated in this study (SVR-LSM significant cluster; population-average TDI map; PDEs of intraoperative iVSAT interferences; average TDI maps associated with iVSAT errors) have been deposited in the Zenodo repository under accession code [https://doi.org/10.5281/zenodo.17544485]. The de-identified minimum dataset necessary to reproduce the study's findings, including individual-level demographic, clinical, processed intraoperative and neuroimaging data, is available under restricted access to protect patient confidentiality, as stipulated by the ethical approval for this study. Access is subject to approval by the IRCCS Ospedale Galeazzi-Sant'Ambrogio and will require the signing of a Data Use Agreement (DUA). Requests for access should be directed to the corresponding author. The raw patient neuroimaging data are protected and are not available due to data privacy regulations. The source data underlying the figures, including fully anonymized behavioural scores and disconnection metrics, are provided as a Source Data file with this paper. The Human Connectome Project (HCP) '100 unrelated subjects' dataset used in this study is publicly available from the ConnectomeDB database (db.humanconnectome.org). Source data are provided with this paper.

## Code availability
SVR-LSM analyses and hyperparameters optimisations were performed using the SVR-LSM toolbox for MATLAB (https://github.com/atdemarco/svrlsmgui). Diffusion MRI processing, tractography, and connectome construction were performed using tools within MRTrix3 (https://www.mrtrix.org/), Freesurfer (https://surfer.nmr.mgh.harvard.edu/) and ANTs (https://github.com/ANTsX/ANTs). Graph theory analyses were conducted using the Brain Connectivity Toolbox (https://sites.google.com/site/bctnet/) in MATLAB.

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

## Acknowledgements

This work has been supported by a grant from Associazione Italiana per la Ricerca sul Cancro (AIRC – 17482) to L.B.

## Author contributions

Conceptualization (G.P., L.V., L.B.); Methodology (G.P., L.V., L.F., L.B.); Connectomic analysis (L.V., L.M.); Neuropsychological assessment (G.P., A.L.); Intraoperative data acquisition (G.P., L.V., A.L., M.R., T.S., M.C.N., L.G.G., L.B.); Visualisation (G.P., L.V.); Statistical analysis (G.P., L.V., L.F.); Writing– original draft (G.P., L.V.); Writing– review & editing (G.P., L.V., L.F., L.B., G.C., A.L.); Funding acquisition (L.B.).

## Competing interests

The authors declare no competing interests.
