## [Transparent Peer Review file · Nature Communications]

Convergent causal mapping unravels distinct frontal networks for visuospatial selective attention

Corresponding Author: Dr Luca Viganò

Version 0:

Reviewer comments:

Reviewer #1

(Remarks to the Author)

Thank you very much for the opportunity to review this interesting manuscript, which investigates the contribution of the dorsal frontal networks to visuo-spatial attention using behavioral data from neurosurgical patients.

First of all, I fully agree with the authors that the combination of direct electrostimulation mapping and lesion-symptom mapping in the same patient population is a robust methodological approach, that indeed allow for the generation of convergent and causal anatomo-functional findings. Overall, I am therefore satisfied with the design of this study.

I also agree that the study of glioma patients often provides original insights, as their preferred locations differ from those in classical neurological populations commonly used in lesion-symptom studies, particularly stroke injuries. As a result, traditionally understudied structures, especially those situated on the dorsal and mesial surfaces of the brain, can be adequately investigated

Having said that, the findings reported appear to be largely confirmatory compared to those recently published (cited by the authors), although they are admittedly presented with a superior level of evidence due to the dual methodology employed. The contribution of the supplementary and cingulate eye fields to visuo-spatial attention and visuomotor neglect has indeed been demonstrated in a recent work, along with the potential role of white matter connections associated with these mediodorsal areas—particularly the -striatal, -thalamic, and fronto-parietal pathways.

In this context, I had expected the current work to provide superior insights into the white matter connections that may subserve “selective” attention. However, the findings largely replicate previously obtained results, with no clear identification of which specific tracts may contribute most significantly (or their distinct functional contributions). To the authors' credit, this is not surprising given the complexity of white matter connectivity that shapes/projects into this region of the brain (preSMA, SMA, FEF).

I also methodological concerns about some aspects of the study.

I have compiled a list of comments that I hope will be useful in refining the work or broadening its scope.

#1 The authors specify several criteria for excluding patients: “Patients with preoperative visuospatial attentive deficits, pre- and postoperative visual deficits, motor or comprehension deficits, previous neurosurgical treatment, or adjuvant radiotherapy prior to surgery were excluded from the study.” I have several points to address:

- Why not include patients with visuo-spatial attention deficits, given that the statistical modeling is based on a delta measure? I can conceive that this is an exclusion criterion for performing intraoperative mapping, but it is less relevant for conducting LSM analyses. Right?
- In line with this, why not perform LSM analyses on preoperative data to determine whether some glioma-induced pathological variance can be modeled?
- Are the authors referring to quadrantanopia or hemianopia when discussing visual deficits?
- The authors may want to specify what proportion of patients was excluded before surgery based on the exclusion criteria (especially regarding cognitive/functional criteria).

#2 The authors wrote: “All the procedures aimed at achieving a supratotal resection independent of clinical or imaging features”. Is really the case? I can understand that surgeries are guided by electrostimulation mapping and are, in this way,

performed independently of clinical or imaging features. However, given the diffuse properties of gliomas, it is expected that in the vast majority of cases, surgical resection will not be "supratotal." I find the phrasing here to be inaccurate.

#3 A limitation of this study relates to the timing of the assessment. While the use of longitudinal data (pre- vs. post-surgery) is a strong point for evaluating the specific consequences of surgery on visuo-spatial abilities, the post-surgery assessment is conducted relatively early. This timing does not allow for an evaluation of the chronic, long-term impact of the procedure on cognitive abilities. The authors need to incorporate this point into their discussion.

#4 Regarding the stimulation parameters, I understand that more detailed information is provided in the Supplementary Materials. However, for the sake of completeness, it might be relevant to include some descriptive statistics on the stimulation parameters used to induce visuo-spatial deficits, such as the mean intensity, standard deviation, etc.

#5 For the LSM study, only one task was designed to assess visuo-spatial abilities: the Bell Test, a cancellation task. However, since visuo-spatial neglect is a polymorphic neuropsychological deficit, it is generally advisable to use a variety of tasks to fully characterize the type of neglect being assessed and its correlation with anatomical structure damage. This represents a significant limitation of the study, as it restricts its scope. This would indeed have helped to dissociate the distinct contributions of individual tracts to visuo-spatial attention and cognition.

#6 For the SVR-LSM procedure, while the method appears to be adequately described overall, it is important to provide more specific details about the type of SVR kernel used. Different kernels, such as linear, nonlinear, or radial basis function (RBF), can significantly influence the model's performance and interpretation, and their selection depends on the data structure.

Others points related to SVR-LSM:

- If I'm not mistaken, the authors did not account for sociodemographic variables in their statistical models, despite these being known to potentially impact visuo-spatial abilities and their recovery, particularly age and education. Why not include them in a nuisance model?

- The authors may want to justify their selection of "DTLVC" for volume correction, as other correction methods are available in the toolbox and may be more conservative.

- If I understand correctly, the authors used Schaefer's RS-fMRI-based parcellation to determine which functional networks the significant voxel-level outputs overlap with. Do the authors binarize the statistical map and use it as a lesion mask in the LQT toolbox? Additionally, it is unclear why the authors opted for a 1000-area parcellation if the goal is to identify the functional networks that may be disrupted, as this level of granularity might not be necessary for such an analysis. Could the authors clarify this choice?. Wouldn't the network-symptom mapping method described by Boes's team (e.g., DOI: 10.1038/s41591-022-01834-y) be more suitable for achieving this goal? This approach might provide a more direct and robust way to identify functional networks associated with specific symptoms.

#7 The authors wrote: "Given that the SVR-LSM revealed that a postoperative worsening in visuospatial selective attention was associated only with a right frontal cluster (see Results), all the subsequent analyses were performed on right hemisphere patients." Is this really a valid argument? The absence of significant results in SVR-LSM based on topographies (resection locations) does not necessarily imply that SVR-LSM based on disconnections would also yield non-significant results. In my opinion, it would be more relevant to perform the analysis and explicitly demonstrate that no significant results are found.

#8 Regarding "the disconnection lesion-symptom mapping" part. Do the authors ensure that the analyses performed using the De Marco's toolbox were configured to handle continuous lesion data appropriately?

#9 Regarding this part: "Identification of disconnected white matter pathways. To label the white matter tracts associated with the deficit, i.e. the subcortical region associated with the impairment in selective visuospatial attention, the significant SVR-LSM and SVR-DSM cluster were merged and compared with the Human Connectome Project tractography atlas (Yeh et al., 2021) by means of tractotron as part of BCBToolKit (Foulon et al., 2018). At each voxel of the SVR clusters, the probability of finding a white matter tract of the HCP atlas was computed. Only tracts reliably disconnected (>50% of probability) by the SVR clusters were considered.

I'm not entirely clear on what has been done here. From my understanding, Tractotron provides a probability that a given tract is damaged based on a lesion map (i.e., if one voxel of a lesion map intersects a fiber tract with 100% overlap in the normative data, the tract is considered 100% disconnected). Do the authors mean that the analyses were performed iteratively on each voxel within the significant clusters identified by SVR-LSM and then averaged? In the table provided in the supplementary file, no measure of variability is provided.

#10 The authors combined data from patients with lower-grade and high-grade gliomas. As the authors are aware, the mechanisms leading to cognitive deficits in these two types of tumors are distinct. Why not control for this difference in the statistical models? This adjustment is easily feasible, especially given the capabilities of the toolbox to perform GLM/nuisance models.

#11 In connection with the previous comment, high-grade gliomas, more so than lower-grade gliomas, cause significant distortions of anatomical structures, particularly white matter tracts. This inevitably introduces some bias into the results of SVR-LSM analyses, whether focused on resection topographies or disconnection maps. How confident are the authors that these distortions do not unduly influence their findings?

#12 I wonder whether it would be possible to perform an analysis that estimates the pathological variance explained by resection topographies versus the variance explained by structural disconnections. This issue is increasingly being discussed in the lesion-symptom mapping literature. I mention this because the SVR-LSM results highlighted a relatively circumscribed white matter area, whereas it would be somewhat expected for this area to be more widespread.

#13 For the ANOVAs, could the authors provide measures of effect size?

#14 For a comprehensive visualization of the results, it would be beneficial for the readers if all individual data points (e.g., pre- vs. post-surgery; left vs. right hemisphere) were plotted. This would provide greater clarity and transparency in illustrating the variability and patterns within the dataset.

#15 In their analyses, the authors did not provide any data to evaluate whether patients, on average, differed from neurologically intact participants before surgery or the proportion of deficits based on established normative data. Including such comparisons may provide a clearer context for assessing the LSM results.

#16 In the figures, the resection density maps range from 1 to the maximum. It would be more effective to threshold the maps to display only the voxels that were included in the analyses.

#17 In my opinion, it would have been interesting to perform an analysis examining the association between the FEF (frontal eye field) identified through electrostimulation and visuo-spatial performance. For example, is there a relationship between the distance from the resection to the FEF and the severity of visuo-spatial deficits? This could provide additional insights into the functional relevance of the FEF in visuo-spatial attention.

Reviewer #2

(Remarks to the Author)

This manuscript contributes significantly to understanding the frontal connectivity supporting visuospatial attention, particularly through the novel integration of DSM and DES. The findings about the dorsomedial and ventrolateral white matter contributions are interesting and backed by interesting analyses. However, several methodological choices (e.g., filtering DSM to right-hemisphere patients, voxel inclusion thresholds) need clearer justification. Additionally, some limitations should be discussed more thoroughly so future research can investigate them.

If these concerns are addressed, the manuscript has the potential to make an impactful contribution to the field.

Comments:

Introduction

"This strategy, grounded in recent guidelines (Siddiqi et al., 2022), provides the highest degree of causal inference in human brain mapping studies." Considering the shortcomings of LSM mentioned above, I believe this is an overstatement.

The first time the "asymmetry score" is mentioned, there is no explanation of that score. I know it's standard in visual neglect, but I think the concept is simple enough to be very briefly explained in one sentence or even a sentence fragment to avoid unfamiliar readers having to look it up. A few sentences later, it is mentioned "(i.e. visuospatial neglect)," but while clarifying a bit, it doesn't say whether a low or high score is good.

Results

Am I correct in understanding that "total missed target" becomes "total score" and then "total missed target score" and goes back to "total score"? I think it would be easier to clarify/harmonise that.

"parcel-level lesion load analysis" the reported values make it hard to evaluate the results. Are all the parcels the same volume? If so, it would seem that the control network has the most cumulated volume of damage from SVR-LSM. From a quick look at the discussion, the other two networks are the stars of the show, not this one.

"Disconnectome symptom mapping" I'm a bit sceptical about filtering the DSM analysis to only right-hemisphere patients based on the LSM results. The two analyses are independent, so I don't see why LSM findings justify excluding left-hemisphere patients. This approach could miss potential contributions from commissural tracts, like the corpus callosum, which might play a role in visuospatial attention. It also reduces the sample size and might explain why the DSM cluster is surprisingly small and similar to the LSM cluster—something I wouldn't expect given the different inputs. Including left-hemisphere patients could have provided a more complete picture, even if they didn't show significant effects.

Figures 1, 2 and 6 have inconsistent display conventions and no orientation indications. This might confuse a lot of readers and needs to be fixed.

Why is the full name of the Bells test mentioned only at this point, after "asymmetry score" has been mentioned multiple times?

I have no idea what I am supposed to see in Figure 3. Why is the Bells Test in the middle? Why is it a correlation matrix between a set of tracts and only one variable? What are the values between the tracts? Correlations between pairs of disconnected tracts?

"White matter disconnection: tractography in single patients" this paragraph needs to be rewritten. There is not enough information at this point of the article to understand where these results come from. "tracts correlated with the Δ asymmetry score Bells Test": It is the percentage of disconnections (number of streamlines, which I am not sure is a reliable measure of disconnections or connection for that matter) correlated to the score, not "the tracts". And this is not explained here. Initially, I thought the disconnection measure was a binary variable (disconnected or not).

Prospective study:

This part of the results is very confusing. The left and right hemispheres are not always consistently mentioned.

Discussion

"atlas-based disconnectome" I believe disconnectome is not "atlas-based".

Filtering DSM to right hemisphere patients: The discussion does not adequately address the decision to exclude left hemisphere patients from the DSM analysis. While the right hemisphere's dominance in visuospatial attention is well-

documented, left hemisphere lesions or disconnections (e.g., commissural tracts) could still play a role. For instance, the DES findings suggest that left hemisphere stimulation occasionally induces neglect-like errors. This omission limits the scope of the DSM analysis and deserves critical discussion here.

Differences between the Bells Test (used for LSM and DSM) and the iVSAT (used during DES) could explain why DES findings suggest bilateral and ventrolateral contributions that are not captured in LSM or DSM. The discussion does not consider how task demands might influence which regions and tracts are identified as critical.

Methods

"Only voxels resected in at least 5% of the patients were included." What does that mean? Does that mean the lesion masks were modified before running the disconnectome to only contain the voxels present in at least 5% of the patients? This is hardly conventional. There doesn't seem to be a justification for that choice. Added to the selection of right hemisphere lesions, this is a lot of filtering.

Figure 6: Why are the colour scales changing between panels D, E and F? They seem to be on the same scales of values.

Supplementary Table 2: the caption is unclear and incomplete.

Reviewer #3

(Remarks to the Author)

In the current study, Puglisi and collaborators combined Lesion-Symptom-Mapping (LSM) in 153 brain tumour patients and Direct Electrical Stimulation (DES) in 47 patients during awake neurosurgery to unveil the network causally associated with visuospatial attention. LSM and DES identified different grey and white matter regions in a right dorsomedial frontal network linked to asymmetry score.

I appreciate the effort and complexity in conducting this study, which aims to contribute to our understanding of the neural correlates of visuo-spatial attention through LSM and DES. The topic is relevant and of interest to the field. However, several concerns regarding the methodological and conceptual aspects of the study should be addressed to strengthen its contribution and improve its clarity and validity.

Major Concerns

1. The authors rely on specific test scores to define visuo-spatial attention performance. However, these do not align with established diagnostic criteria for visuo-spatial neglect, limiting the study's ability to accurately infer crucial knowledge from pathological visuo-spatial attention. Incorporating clinical evidence of visuo-spatial neglect in the tested patients using the standard diagnostic frameworks would significantly enhance the validity of the findings.
2. The anatomical correlates identified in the study appear to reflect task-specific processes rather than generalizable visuo-spatial attention mechanisms, providing only a limited theoretical framework. Previous studies (e.g., Verdon et al., 2010; Committeri et al., 2007) have demonstrated neural correlates based on task type variability. The tasks employed in this study visuo-motor cancellation (Bells Test) and visual exploration (adapted Diller)—tap into distinct cognitive functions, which should be explicitly acknowledged and discussed. Additionally, the results largely confirm previous findings and their novelty could be better contextualized as it does not emerge in the actual text the authors provide.
3. The broad age range of participants (16–77 years) introduces potential variability due to developmental and age-related neurobiological changes, such as ongoing myelination processes in younger participants, especially considering that the sample investigated mainly concerns patients with frontal lobe lesions. Including age as a covariate in the analysis would control for these effects and enhance the robustness of the conclusions.
4. The LSM method includes only voxels resected in at least 5% of participants. While there are no strict guidelines for this threshold, a higher value (e.g., 10% of the sample) is generally recommended to enhance reliability and reduce noise. It would be helpful for the authors to either adjust this threshold or provide a clear justification for their choice to ensure the robustness of the findings.
5. While the paper emphasizes convergent results, the lack of task performance interference during DES applied at the cortical level rather suggests discrepancies in the findings. This evident contradiction should be addressed to provide a more nuanced interpretation of the results and their implications for the hypothesized frontoparietal network.
6. The association between DES and lesion location with visuo-spatial attention deficits has been demonstrated in prior studies. For example, the involvement of the right-lateralized frontoparietal network in cancellation tasks is well-documented. The manuscript could better highlight how its findings advance existing knowledge or test specific theoretical models of attention.
7. I would suggest toning down statements such as "Our results confirm a right-lateralized dorsomedial circuit for contralateral visuospatial attention". A crucial limitation, as noted by the authors, is the lesion distribution bias toward the

frontal lobe. This bias restricts the scope of exploration, both in terms of the lesion basis and the effects of electrical stimulation on other brain areas potentially involved in visuo-spatial attention, as already well-known literature on this topic. A more cautious interpretation of the results would better reflect these constraints.

Minor Concerns

1. The introduction provides a narrow view of attention models. Expanding this section to include alternative theoretical frameworks would situate the study within a broader context. Additionally, the authors could elaborate on how their results challenge or support these models.
2. While the authors critique the limitations of LSM, they also rely on this methodology. Balancing this critique with an acknowledgement of DES limitations, such as diffusion to neighbouring regions, would provide a fairer assessment of both approaches.
3. Reducing the use of complex acronyms (e.g., DorsAttnB_FEF_3) and providing consistent explanations for terms would improve readability and accessibility for a broader audience.
4. The rationale for using the 17/1000 Schaefer et al. (2018) parcellation should be clarified. Why was this specific framework chosen, and how does it align with the study's aims?
5. The section titled "Impact of iVSAT use on neuropsychological outcomes" could be revised for clarity. Additionally, clearer explanations of terms and concepts throughout the manuscript would enhance understanding.
6. Another issue is that the results are presented using a neuroradiological convention rather than a neurological convention. I suggest that the authors align the presentation with the neurological convention, ensuring that right hemisphere results are displayed on the right side. This adjustment would improve clarity and consistency for readers in the field.
7. The authors used ANOVA to analyze the behavioural data and included the test asymmetry score in the LSM analysis. Were the data assessed for normality to ensure that the assumptions of ANOVA were met? If not, it would be important to evaluate whether the data distribution is suitable for this analysis or consider alternative statistical approaches. Additionally, the LSM analysis may also be impacted by non-normal data distribution, which could affect the robustness and interpretability of the results. Addressing these issues would help strengthen the study's methodological rigour.
8. When performing the SVR-LSM analysis, the authors chose to control for lesion volume using the dTLVC approach described by Zhang et al. (2014). However, the SVR-LSM software (as released by De Marco and collaborators) includes the capability to control lesion volume directly, accounting for both the lesion map and behavioural data. Why did the authors opt not to use this implementation suggested by De Marco et al.? Clarifying this choice would provide valuable insight into the methodological decisions and their potential impact on the results.

Version 1:

Reviewer comments:

Reviewer #1

(Remarks to the Author)

I have now read the revised manuscript in detail. I would like to thank the authors for taking my comments and suggestions very seriously and for their impressive work. I believe the revised anatomical analysis workflow is much more robust, and the subsequent results go well beyond what I was expecting. In particular, I am very pleased to see that the authors now clearly succeed in delineating the relevant white matter systems underlying visuospatial neglect, especially the implication of the projections of the supplementary cingulate eye field. From my perspective, this finding is highly original and carries important implications both fundamentally and clinically.

My final suggestion would be to provide a methodological figure, placed at the end of the introduction, to help readers easily navigate the methodological workflow, given the large number of analyses presented.

Congratulations on this excellent work.

Reviewer #2

(Remarks to the Author)

The authors addressed all major points. Either they implemented the changes I requested, or they replaced the criticised analyses with more robust alternatives and explained their rationale.

I would just note that the rebuttal could have been clearer if the authors had quoted their revised text directly. As it is, checking the changes requires going back and forth with the manuscript, which makes the process a bit tedious.

Reviewer #3

(Remarks to the Author)

The authors have significantly improved the manuscript, and I acknowledge the imaging methodological innovation and the considerable effort involved in testing such a large cohort of patients. I congratulate the authors on their work. However, several points still require clarification. While the results are of interest and contribute to the field, they substantially overlap with findings from previous classical neuropsychological studies, thus somewhat modulating the overall novelty of the present work. There are also specific methodological aspects, particularly concerning the neuropsychological testing, that need to be more explicitly addressed.

Page 6 lines 158-175

A central issue in the manuscript concerns the use of a continuous "total score" in the analyses rather than a threshold-based pathological score for the Bell's test. While this approach may allow for more nuanced statistical analysis, it compromises the clinical interpretability of the findings. In the absence of a comparison to normative data or cut-off thresholds, it remains unclear whether the observed deficits reflect actual neuropsychological impairments (e.g., pathological score). Clarifying this point would strengthen the conclusions regarding lesion-symptom mapping and its clinical relevance.

Page 7 lines 189-193

A similar consideration applies to the LSM analysis. Although the use of a continuous behavioral score is methodologically sound, it does not allow for a clear distinction between patients with and without attentional impairments. In contrast, classical neuropsychological assessments typically compare a patient's performance to a normative dataset and classify outcomes dichotomously (i.e., pathological vs. non-pathological). Incorporating a clinical categorical variable, indicating whether each patient shows a clinically significant deficit, could provide additional insights into the neural correlates of attention deficits. At the very least, the manuscript should acknowledge this limitation and justify the methodological choice more explicitly.

Page 10 lines 301-315

The same limitation mentioned above applies to the iVSAT scores. It would be informative to assess whether stimulation induced a clinically meaningful neuropsychological deficit in individual patients. This would ideally require a comparison with normative data, which could clarify whether the observed behavioral changes reflect actual deficits rather than subclinical modulations. Without such a reference point, it is more accurate to interpret the results as evidence of attentional modulation rather than as a demonstration of the neural bases of pathological attentional impairments. For instance, in the discussion, at around line 382, the authors discuss "visuospatial selective attention impairment" due to the stimulation.

Page 17, line 515

The use of two different tasks to assess attentional functions introduces a methodological limitation that should be explicitly acknowledged. Differences in task demands, sensitivity, and specificity may contribute to discrepancies between the patients' clinical profiles and the areas identified through stimulation (e.g., as discussed around line 515). A more detailed discussion of this point would help contextualize the results and avoid potential overinterpretations. It may also guide future studies toward a more unified and systematic testing approach.

Retracing the steps of my previous comment, the manuscript refers broadly to "visuospatial attention," but it is important to clarify that the study addresses only a specific component of this multifaceted construct. Although the authors' findings are in line with previous literature, the interpretation should be appropriately restrained, given the methodological differences and the specificity of the tasks employed. I recommend adjusting the tone of the manuscript to reflect that the study investigates only a subcomponent of visuo-spatial attention and does not offer a comprehensive exploration of the neural underpinnings of visuo-spatial attention.

Version 2:

Reviewer comments:

Reviewer #3

(Remarks to the Author)

REVIEWER COMMENTS

Reviewer #1 (Remarks to the Author):

Thank you very much for the opportunity to review this interesting manuscript, which investigates the contribution of the dorsal frontal networks to visuo-spatial attention using behavioral data from neurosurgical patients.

First of all, I fully agree with the authors that the combination of direct electrostimulation mapping and lesion-symptom mapping in the same patient population is a robust methodological approach, that indeed allow for the generation of convergent and causal anatomo-functional findings. Overall, I am therefore satisfied with the design of this study.

I also agree that the study of glioma patients often provides original insights, as their preferred locations differ from those in classical neurological populations commonly used in lesion-symptom studies, particularly stroke injuries. As a result, traditionally understudied structures, especially those situated on the dorsal and mesial surfaces of the brain, can be adequately investigated

Having said that, the findings reported appear to be largely confirmatory compared to those recently published (cited by the authors), although they are admittedly presented with a superior level of evidence due to the dual methodology employed. The contribution of the supplementary and cingulate eye fields to visuo-spatial attention and visuomotor neglect has indeed been demonstrated in a recent work, along with the potential role of white matter connections associated with these mediadorsal areas—particularly the -striatal, -thalamic, and fronto-parietal pathways.

In this context, I had expected the current work to provide superior insights into the white matter connections that may subserve “selective” attention. However, the findings largely replicate previously obtained results, with no clear identification of which specific tracts may contribute most significantly (or their distinct functional contributions). To the authors' credit, this is not surprising given the complexity of white matter connectivity that shapes/projects into this region of the brain (preSMA, SMA, FEF).

We thank the Reviewer for the positive evaluation of our manuscript, for appreciating the strength of our dual-methodology approach, and for this very insightful comment.

We agree with the Reviewer's core point. While our initial submission provided strong causal evidence for the involvement of dorsomedial frontal regions, its tractography analysis was partially confirmatory. The previous method—which involved performing a disconnectome-symptom mapping (SVR-DSM) and then projecting the resulting focal cluster onto a probabilistic tract atlas using Tractotron—could only suggest a list of candidate tracts without capturing with more anatomical precision the contributions of the specific cortical or subcortical areas they connect. This level of granularity is missing in the

recent investigations which linked dorsomedial white matter tracts with visuospatial neglect (Herbet et al. 2022; Nakajima et al. 2021). These studies identified entire, a-priori defined white matter tracts. Each of them connect multiple cortical and subcortical sub-regions. For instance, an atlas-defined fronto-striatal tract is an extended pathway connecting the striatum to multiple areas within the SFG and the MFG. Consequently, if a lesion analysis implicates this entire tract, it remains undefined whether the observed deficit is due to the disconnection of the MFG, the SFG, or only a specific sub-region within them. The previous investigations lacked the resolution to distinguish between these possibilities.

To address this, and with the explicit aim of providing a more granular and novel understanding of the relevant white matter network associated with visuospatial neglect, we have undertaken a complete revision of our tractography pipeline. We have replaced the previous analysis with a state-of-the-art, multi-stage connectomic approach (Tahedl et al. 2025).

First, instead of relying on a pre-defined tract atlas, we now use the significant SVR-LSM cluster as a seed to perform a data-driven exploration of its structural connectome. Using a large cohort of 100 healthy HCP subjects (100 unrelated HCP data set), we built a region-to-region connectome quantitatively weighted by Fibre Bundle Capacity (FBC), a measure of total intra-axonal cross-sectional area derived from SIFT2-filtered tractograms (Smith et al. 2015; 2022). This method does not depend on prior anatomical knowledge and is thus ideally suited to characterizing and quantifying the region-to-region connectivity of complex networks (see Methods, Section 1.6 lines 674-719 ; Results, Section 1.4 lines 219-231).

This connectomic approach allowed us to move beyond the method employed in the first submission. By applying graph theory measures (strength, degree, betweenness centrality), we identified the most critical ‘consensus hubs’ of the network (including key regions in the medial frontal cortex like the Supplementary Motor Area and Cingulate Eye Field, more lateral premotor areas surrounding the superior frontal sulcus, and the thalamus) and, crucially, their most important connections (‘consensus edges’) (Rubinov and Sporns 2010; Bullmore and Sporns 2009). This provides a far more specific, region-to-region structural network associated with visuospatial neglect deficits (see Methods, Section 1.6.2 lines 720-747; Results, Section 1.4.1 lines 233-255).

We then validated the clinical relevance of these specific network edges in our own patients with preoperative dMRI. The previous version of our paper relied on manual virtual dissection of atlas-defined tracts and correlating deficits with the percentage of resected streamlines. We now show a significant correlation between the postoperative deficit and the percentage of disconnection of the FBC for each consensus edge. This analysis revealed that damage to three connections in particular—a right SCEF-thalamus connection, a right SCEF-brainstem connection, and a transcallosal SFL-SCEF connection—was most significantly associated with visuo-spatial neglect deficits (see Methods, Section 1.7 lines 760-803; Results, Section 1.5 lines 266-279). This directly answers the Reviewer’s call for identifying which connections contribute most significantly.

To complement this edge-based analysis, we have now included two track-density imaging (TDI, Calamante et al. 2010,2011) approaches:

- 1) A population-average TDI map from the HCP data that visualizes the anatomical substrate of the normative neglect-related network (see Methods, Retrospective study Section 1.6.3 lines 749-759; Results, Retrospective study Section 1.4.2 257-264).
- 2) In our prospective cohort, we generated average TDI maps from the DES-eloquent sites. This analysis revealed a clear anatomical dissociation: neglect-like errors were associated with medial frontal pathways consistent with our normative network, whereas errors confined to the right visuo-spatial hemifield were linked to more lateral fronto-parietal connections. This suggests a structural dissociation between two functionally distinct frontal attention networks (see Methods - Prospective study section 2.5 lines 885-896 and Results - Prospective study , Section 2.4 lines 339-354).

We hope that this completely revised connectomic pipeline provides quantitative and granular insights into the white matter substrates of visuospatial attention that the Reviewer called for.

Calamante, F., Tournier, J.-D., Heidemann, R. M., Anwender, A., Jackson, G. D., & Connelly, A. (2011). Track density imaging (TDI): Validation of super resolution property. *NeuroImage*, 56(3), 1259–1266. <https://doi.org/10.1016/j.neuroimage.2011.02.059>

Calamante, F., Tournier, J.-D., Jackson, G. D., & Connelly, A. (2010). Track-density imaging (TDI): Super-resolution white matter imaging using whole-brain track-density mapping. *NeuroImage*, 53(4), 1233–1243. <https://doi.org/10.1016/j.neuroimage.2010.07.024>

Herbet, G., & Duffau, H. (2022). Contribution of the medial eye field network to the voluntary deployment of visuospatial attention. *Nature Communications*, 13(1), Articolo 1. <https://doi.org/10.1038/s41467-022-28030-3>

Nakajima, R., Kinoshita, M., & Nakada, M. (2021). Simultaneous Damage of the Cingulate Cortex Zone II and Fronto-Striatal Circuit Causes Prolonged Selective Attentional Deficits. *Frontiers in Human Neuroscience*, 15, 762578. <https://doi.org/10.3389/fnhum.2021.762578>

Tahedi, M., Tournier, J.D. & Smith, R.E. Structural connectome construction using constrained spherical deconvolution in multi-shell diffusion-weighted magnetic resonance imaging. *Nat Protoc* (2025). <https://doi.org/10.1038/s41596-024-01129-1>

Smith RE, Tournier J-D, Calamante F, Connelly A. SIFT2: Enabling dense quantitative assessment of brain white matter connectivity using streamlines tractography. *NeuroImage* 2015;119:338–51. <https://doi.org/10.1016/j.neuroimage.2015.06.092>.

Smith, Robert E., et al. “Quantitative Streamlines Tractography: Methods and Inter-Subject Normalisation.” *Aperture Neuro*, vol. 2, Apr. 2022, pp. 1–25, <https://doi.org/10.52294/ApertureNeuro.2022.2.NEOD9565>.

Rubinov M, Sporns O. Complex network measures of brain connectivity: uses and interpretations. *Neuroimage*. 2010 Sep;52(3):1059-69. doi: 10.1016/j.neuroimage.2009.10.003.

I also methodological concerns about some aspects of the study.

I have compiled a list of comments that I hope will be useful in refining the work or broadening its scope.

#1.1 The authors specify several criteria for excluding patients: "Patients with preoperative visuospatial attentive deficits, pre- and postoperative visual deficits, motor or comprehension deficits, previous neurosurgical treatment, or adjuvant radiotherapy prior to surgery were excluded from the study." I have several points to address:
- Why not include patients with visuo-spatial attention deficits, given that the statistical modeling is based on a delta measure? I can conceive that this is an exclusion criterion for performing intraoperative mapping, but it is less relevant for conducting LSM analyses. Right?

We thank the Reviewer for this insightful question regarding patient inclusion for the SVR-LSM analysis. Our initial decision to exclude patients with any significant preoperative visuospatial attention deficits was aimed at ensuring a homogeneous baseline for assessing postoperative changes and minimizing potential confounds in our primary SVR-LSM analysis of delta scores. We agree with the Reviewer's premise that using delta scores (the change between preoperative and postoperative performance) in the SVR-LSM should mitigate this specific confound.

To comprehensively address this point, we have revised our SVR-LSM analysis. We have now incorporated patients who presented with preoperative impairments specifically on the Bell's Test total score (n=10; 5 left, 5 right hemisphere lesions; defined as pathological based on Italian normative data by Vallar et al. 1994; see Supplementary Table 1). It is important to note that no patients in our cohort exhibited preoperative deficits on the Bell's Test asymmetry score. The inclusion criteria in the Methods (Retrospective study - 1.1 Patient cohort, lines 585-594) has been updated to include those patients. Coherently, Results have been changed adding information on these 10 patients (Results - Retrospective study - 1.1/1.2 lines 144-182).

The inclusion of these 10 participants with preoperative total score deficits did not alter the principal SVR-LSM findings; the significant lesion-deficit clusters identified remained consistent with our original results (Figure 1D/G).

We fully agree with the Reviewer's distinction regarding the prospective cohort. For the intraoperative mapping study, the exclusion of patients with preoperative visuospatial attention deficits remains essential. Reliable intraoperative functional assessment is critical for safe and effective mapping, and pre-existing impairments would significantly compromise the interpretability and utility of these real-time evaluations, potentially impacting surgical decisions and patient safety.

Vallar, G., Rusconi, M. L., Fontana, S., & Musicco, M. (1994). Tre test di esplorazione visuo-spaziale: taratura su 212 soggetti normali [Three clinical tests for the assessment of visuo-spatial exploration. Norms from 212 normal subjects]. *Archivio di Psicologia, Neurologia e Psichiatria*, 55(4), 827–841.

- In line with this, why not perform LSM analyses on preoperative data to determine whether some glioma-induced pathological variance can be modeled?

We appreciate the Reviewer's suggestion to explore potential glioma-induced pathological variance using preoperative SVR-LSM. In response, we conducted two exploratory SVR-LSM analyses on the preoperative data, using the Bell's Test total score and the Bell's Test asymmetry score, respectively, as dependent variables. These analyses employed the same parameters as our main postoperative SVR-LSM.

Neither of these preoperative SVR-LSM analyses yielded any significant lesion-symptom clusters in either hemisphere. These null findings are not surprising, primarily because of the limited variance and low incidence of impairment in the preoperative behavioral scores within our cohort. Specifically, for the Bell's Test total score, the mean (\pm SD) number of omissions was 1.9 (\pm 2.2) for left hemisphere patients and 1.6 (\pm 1.5) for right hemisphere patients, with a low overall prevalence of pathological scores (6.2% and 6.1%, respectively) (Results - Retrospective study - 1.2 Neuropsychological Assessment). For the Bell's Test asymmetry score, the preoperative scores were even more homogeneous, with a mean (\pm SD) of 0.06 (\pm 0.9) for left hemisphere patients and 0.42 (\pm 1.1) for right hemisphere patients (no pathological scores).

Given these consistently null preoperative findings, which are likely a consequence of the limited behavioral variability at baseline, and considering the manuscript's primary focus on elucidating the anatomical correlates of postoperative visuospatial deficits, we have not included these specific exploratory analyses in the main text. We believe their inclusion would not substantially alter the paper's core contributions. However, we would be happy to add a brief mention of these preoperative analyses if the Reviewer or Editor considers it beneficial.

1.2. Are the authors referring to quadrantanopia or hemianopia when discussing visual deficits?

In our study, the term “visual deficits” encompasses both hemianopia and quadrantanopia. We have clarified this in the Methods: “*Patients with pre- and post-operative visual deficits (hemianopia or quadrantanopia) [...]*” (Methods - Retrospective study - 1.1 Patient cohort lines 585-592). We also updated the Results section: 1) “*8 patients were excluded due to visual deficits (2 had hemianopia, and 6 had quadrantanopia)*” (Results - Retrospective study - 1.1 Patient cohort, lines 144-149); 2) Prospective study: “*One patient was excluded due to hemianopia [...]*” (Results - Prospective study - 2.1 Patient cohort, lines 281-285; Methods - Prospective study - 2.1 Patient cohort - lines 805-818).

1.3 The authors may want to specify what proportion of patients was excluded before surgery based on the exclusion criteria (especially regarding cognitive/functional criteria).

We agree that this information should be added. We have modified the text accordingly:

Retrospective study: 171 patients were considered in the retrospective cohort (Methods - 1.1 Patient cohort lines 585-592). Of these, 8 patients were excluded due to visual deficits: 2 had hemianopia, and 6 had quadrantanopia. Regarding cognitive deficits, we initially planned to exclude 10 patients with preoperative visuospatial attention impairments (see response to comment 1.1). However, following the Reviewer's suggestion, we retained these patients in the analysis. The final number of patients is 163 (Results - 1.1 Patient cohort lines 144-149)

Prospective study: "Of the 50 patients considered for enrolment, 47 patients (27 right hemisphere) undergoing brain tumour resection in awake anaesthesia met the inclusion criteria. One patient was excluded due to hemianopia, and two were excluded for visuospatial attention deficits." (Results - 2.1 Patient cohort lines 281-285)

#2 The authors wrote: "All the procedures aimed at achieving a supratotal resection independent of clinical or imaging features". Is really the case? I can understand that surgeries are guided by electrostimulation mapping and are, in this way, performed independently of clinical or imaging features. However, given the diffuse properties of gliomas, it is expected that in the vast majority of cases, surgical resection will not be "supratotal." I find the phrasing here to be inaccurate.

Thank you for raising this point regarding our statement on surgical goals. We agree that the original wording could be misinterpreted. Given that the primary focus of the current manuscript is not oncological, we concur that this information is not strictly necessary. Therefore, to improve the clarity and focus of the paper, we have removed the sentence.

However, we appreciate the opportunity to clarify our surgical strategy. Our approach is guided by maximizing resection within functional boundaries, as determined by intraoperative brain mapping. As the Reviewer correctly noted, SpTR (supratotal resection) is not achievable in all cases and depends on several clinical factors (Gallotti et al. 2025). As detailed in our group's previous large-scale studies on lower-grade gliomas (Rossi et al. 2021; Gallotti et al. 2025), resection is pursued until eloquent sites are encountered, irrespective of MRI-defined tumor margins. Whenever possible, resections aim at achieving a SpTR as growing evidence links SpTR with progression-free survival (Molinaro et al., 2020; Rossi et al., 2021; Yordanova & Duffau, 2017, Gallotti et al. 2025).

Rossi M, Gay L, Ambrogi F, Conti Nibali M, Sciortino T, Puglisi G, Leonetti A, Mocellini C, Caroli M, Cordera S, Simonelli M, Pessina F, Navarria P, Pace A, Soffietti R, Rudà R, Riva M, Bello L. Association of supratotal resection with progression-free survival, malignant transformation, and overall survival in lower-grade gliomas. *Neuro Oncol.* 2021 May 5;23(5):812-826. doi: 10.1093/neuonc/noaa225.

Gallotti AL, Rossi M, Conti Nibali M, Sciortino T, Gay LG, Puglisi G, Leonetti A, Bruno F, Rudà R, Soffietti R, Cerri G, Bello L. Neuro-oncological superiority of supratotal resection in lower-grade gliomas. *Neuro Oncol.* 2025 Jun 21;27(5):1270-1284. doi: 10.1093/neuonc/noae264.

Yordanova YN, Duffau H. Supratotal resection of diffuse gliomas - an overview of its multifaceted implications. *Neurochirurgie*. 2017 Jun;63(3):243-249. doi: 10.1016/j.neuchi.2016.09.006.

Molinaro AM, Hervey-Jumper S, Morshed RA, Young J, Han SJ, Chunduru P, Zhang Y, Phillips JJ, Shai A, Lafontaine M, Crane J, Chandra A, Flanigan P, Jahangiri A, Cioffi G, Ostrom Q, Anderson JE, Badve C, Barnholtz-Sloan J, Sloan AE, Erickson BJ, Decker PA, Kosel ML, LaChance D, Eckel-Passow J, Jenkins R, Villanueva-Meyer J, Rice T, Wrench M, Wiencke JK, Oberheim Bush NA, Taylor J, Butowski N, Prados M, Clarke J, Chang S, Chang E, Aghi M, Theodosopoulos P, McDermott M, Berger MS. Association of Maximal Extent of Resection of Contrast-Enhanced and Non-Contrast-Enhanced Tumor With Survival Within Molecular Subgroups of Patients With Newly Diagnosed Glioblastoma. *JAMA Oncol*. 2020 Apr 1;6(4):495-503. doi: 10.1001/jamaoncol.2019.6143. Erratum in: *JAMA Oncol*. 2020 Mar 1;6(3):444. doi: 10.1001/jamaoncol.2020.0360.

#3 A limitation of this study relates to the timing of the assessment. While the use of longitudinal data (pre- vs. post-surgery) is a strong point for evaluating the specific consequences of surgery on visuo-spatial abilities, the post-surgery assessment is conducted relatively early. This timing does not allow for an evaluation of the chronic, long-term impact of the procedure on cognitive abilities. The authors need to incorporate this point into their discussion.

While it is true that a longer follow-up could assess the chronic impact of surgery on cognitive abilities, we deliberately chose to perform the post-surgical assessment at one month for two key reasons.

First, the assessment at one month provides a more accurate and reliable measure of the direct impact of surgery on visuo-spatial abilities. Over longer periods, neuroplasticity mechanisms may lead to compensatory processes that confound the attribution of deficits to the resection itself. Evaluating patients at an early stage allows us to capture deficits that are more likely to be directly related to surgical intervention rather than to subsequent adaptive processes (Cargnelutti et al. 2020; Duffau et al.2021) .

Second, in many cases, patients undergo adjuvant therapies such as radiotherapy and chemotherapy after the 1-month follow up, which are known to influence cognitive performance (Leonetti et al., 2021). These treatments introduce additional variability, making it challenging to isolate the specific contribution of surgical resection to long-term cognitive outcomes. By assessing patients at an early stage (one month), is one of the best trade-off to minimize the confounding effects of these therapeutic interventions and ensure that our findings primarily reflect the consequences of surgery. We have expanded the discussion to clarify the rationale for conducting post-surgical assessments at one month and to acknowledge that while longer-term follow-ups could provide valuable insights, they would introduce confounding factors related to neuroplasticity and adjuvant treatments (see Discussion lines 555-564).

Cargnelutti E, Ius T, Skrap M, Tomasino B. What do we know about pre- and postoperative plasticity in patients with glioma? A review of neuroimaging and intraoperative mapping studies. *Neuroimage Clin*. 2020;28:102435. doi: 10.1016/j.nicl.2020.102435.

Duffau H. Introducing the concept of brain metaplasticity in glioma: how to reorient the pattern of neural reconfiguration to optimize the therapeutic strategy. *J Neurosurg*. 2021 Oct 8;136(2):613-617. doi: 10.3171/2021.5.JNS211214.

Leonetti A, Puglisi G, Rossi M, Viganò L, Conti Nibali M, Gay L, Sciortino T, Howells H, Fornia L, Riva M, Cerri G, Bello L. Factors Influencing Mood Disorders and Health Related Quality of Life in Adults With Glioma: A Longitudinal Study. *Front Oncol*. 2021 May 20;11:662039. doi: 10.3389/fonc.2021.662039.

#4 Regarding the stimulation parameters, I understand that more detailed information is provided in the Supplementary Materials. However, for the sake of completeness, it might be relevant to include some descriptive statistics on the stimulation parameters used to induce visuo-spatial deficits, such as the mean intensity, standard deviation, etc.

We agree with the Reviewer that reporting the descriptive statistics of the stimulation parameters used during the iVSAT mapping is relevant for interpretation of the results and to allow reproducibility. Therefore, we have made the following revisions:

- In the Methods section (Prospective study - 2.2 Intraoperative assessment of visuospatial attention lines 835-844), we have better explained the Low-Frequency Direct Electrical Stimulation (LF-DES) mapping parameters (e.g., biphasic pulses, 60 Hz, 0.5 ms pulse width, 1-4s trains, 5mm tip distance) and the strategy for determining stimulation intensity.
- In the Results section (Prospective study - 2.2 Intraoperative stimulation lines 292-297), we now report the descriptive statistics for the LF-DES intensity used for iVSAT mapping across the patient cohort (right hemisphere: mean intensity of 3.82 ± 1.19 mA [range: 2.0 – 6.0 mA]; left hemisphere: mean intensity 3.42 ± 1.19 mA [range: 1.5 – 6.0 mA]).

#5 For the LSM study, only one task was designed to assess visuo-spatial abilities: the Bell Test, a cancellation task. However, since visuo-spatial neglect is a polymorphic neuropsychological deficit, it is generally advisable to use a variety of tasks to fully characterize the type of neglect being assessed and its correlation with anatomical structure damage. This represents a significant limitation of the study, as it restricts its scope. This would indeed have helped to dissociate the distinct contributions of individual tracts to visuo-spatial attention and cognition.

Our choice to focus exclusively on a classical cancellation task (Bells test), was guided by two main considerations. First, clinical feasibility constraints played a key role in task selection. Given the time limitations of neuropsychological assessment in a neurosurgical setting, it was crucial to prioritize a task that was both practical for clinical use and highly sensitive to neglect-related impairments. Cancellation tasks, including the Bells Test, have been widely recognized as some of the most reliable and sensitive tools for detecting visuo-spatial deficits in both acute and chronic stages of neglect (Ferber & Karnath, 2001; Azouvi et al., 2002, Menon, 2004) . Second, there is strong evidence that cancellation tasks effectively capture key components of neglect, including those assessed by other paradigms such as line bisection. Previous studies (Azouvi et al., 2002) have demonstrated that neglect-related deficits in bisection tasks are often reflected in cancellation performance, indicating that these tasks share overlapping neural mechanisms. While a more comprehensive battery could have allowed for better differentiation of multiple components of neglect syndrome, we believe our results still offer valuable insights into key functional correlates of spatial attention impairments.

Azouvi P, Samuel C, Louis-Dreyfus A, Bernati T, Bartolomeo P, Beis JM, Chokron S, Leclercq M, Marchal F, Martin Y, De Montety G, Olivier S, Perennou D, Pradat-Diehl P, Prairial C, Rode G, Siéroff E, Wiart L, Rousseaux M; French Collaborative Study Group on Assessment of Unilateral Neglect (GEREN/GRECO). Sensitivity of clinical and behavioural tests of spatial neglect after right hemisphere stroke. *J Neurol Neurosurg Psychiatry*. 2002 Aug;73(2):160-6. doi: 10.1136/jnnp.73.2.160.

Menon A, Korner-Bitensky N. Evaluating unilateral spatial neglect post stroke: working your way through the maze of assessment choices. *Top Stroke Rehabil*. 2004 Summer;11(3):41-66. doi: 10.1310/KQWL-3HQL-4KNM-5F4U.

#6 For the SVR-LSM procedure, while the method appears to be adequately described overall, it is important to provide more specific details about the type of SVR kernel used. Different kernels, such as linear, nonlinear, or radial basis function(RBF), can significantly influence the model's performance and interpretation, and their selection depends on the data structure.

We thank the Reviewer for highlighting the need to specify our SVR kernel and implementation. In our analyses we used the SVR-LSM toolbox using the MATLAB Statistics and Machine Learning Toolbox, employing a a nonlinear radial basis function (Gaussian) kernel. This information is now reported in the Methods (Retrospective study - 1.5 Lesion symptom mapping lines 642-645).

- If I'm not mistaken, the authors did not account for sociodemographic variables in their statistical models, despite these being known to potentially impact visuo-spatial abilities and their recovery, particularly age and education. Why not include them in a nuisance model?

Following the Reviewer's suggestion, we have incorporated: 1) age, 2) education and 3) tumour grade as covariates in the lesion-symptom mapping (LSM) analysis to assess whether these variables influenced the results. The inclusion of these factors did not alter the main findings, confirming the robustness of our results (Figure 1 D/G). This suggests that the observed effects are primarily driven by lesion location rather than demographic or tumour grade. The methods and results sections were modified to include this information (Retrospective study - Methods: 1.5 Lesion symptom mapping lines 645-647; Results: 1.3 Lesion-symptom mapping - SVR-LSM lines 184-207).

- The authors may want to justify their selection of "DTLVC" for volume correction, as other correction methods are available in the toolbox and may be more conservative.

Thank you for this suggestion. We now implemented the "regress-out-of-both" approach (i.e. regressing lesion volume from both the behavioural scores and the lesion maps) and compared it directly to our dTLVC pipeline. As shown in Revision Figure 1, the regress-both correction yields a substantially larger suprathreshold cluster compared to dTLVC and drives the average prediction accuracy to -0.20 (SD = 0.08), compared with the optimal value obtained with dTLVC (0.30, SD = 0.09).

At present, there is still no consensus on the gold standard for volume correction (Zhang et al. 2014; DeMarco and Turkeltaub 2018; Wiesen et al.2019). Similarly to our approach,

in a recent paper, Wiesen et al. (2019) directly compared the two methods, finding that dTLVC yielded a better prediction accuracy for their dataset and finally opting for this solution for presenting their main results.

Zhang Y, Kimberg DY, Coslett HB, Schwartz MF, Wang Z. 2014. Multivariate lesion-symptom mapping using support vector regression. *Human Brain Mapping* 35(12): 5861–5876. <https://doi.org/10.1002/hbm.22590>

DeMarco AT, Turkeltaub PE. 2018. A multivariate lesion symptom mapping toolbox and examination of lesion-volume biases and correction methods in lesion-symptom mapping. *Human Brain Mapping* 39(11): 4169–4182. <https://doi.org/10.1002/hbm.24289>

Wiesen D, Sperber C, Yourganov G, Rorden C, Karnath H-O. 2019. Using machine learning-based lesion behavior mapping to identify anatomical networks of cognitive dysfunction: Spatial neglect and attention. *NeuroImage* 201: 116000. <https://doi.org/10.1016/j.neuroimage.2019.07.013>

Supplementary Figure 1: Significant SVR-LSM cluster associated with postoperative Bell's Test asymmetry score changes, corrected for cluster-level family-wise error. The figure compares clusters derived using two different lesion volume control methods: (Red) Controlling for lesion volume via regression in the SVR model (prediction accuracy: -0.20 ± 0.08 SD). (Green) Using direct total lesion volume control (dTLVC; prediction accuracy: 0.30 ± 0.09 SD). The dTLVC method was adopted for the main analyses presented in the manuscript.

- If I understand correctly, the authors used Schaefer's RS-fMRI-based parcellation to determine which functional networks the significant voxel-level outputs overlap with. Do the authors binarize the statistical map and use it as a lesion mask in the LQT toolbox? Additionally, it is unclear why the authors opted for a 1000-area parcellation if the goal is to identify the functional networks that may be disrupted, as this level of granularity might not be necessary for such an analysis. Could the authors clarify this choice? Wouldn't the network-symptom mapping method described by Boes's team (e.g., DOI: [10.1038/s41591-022-01834-y](https://doi.org/10.1038/s41591-022-01834-y)) be more suitable for achieving this goal? This approach might provide a more direct and robust way to identify functional networks associated with specific symptoms.

1) The SVR-LSM significant cluster is indeed a binary mask retaining only significant voxels after CFWER. This is now explicit in the Methods (Retrospective study - 1.5 Lesion symptom mapping line 663)

2) Concerning the choice of the 1000 Schaefer atlas, our primary aim for this analysis was to assign the significant SVR-LSM cluster to established large-scale functional networks, while also retaining the ability to appreciate the contribution of potentially distinct sub-regions within those networks, especially given that our SVR-LSM cluster itself comprises different focal components.

We therefore utilized the Schaefer 1000-parcel atlas. This choice was guided by the understanding that there is likely no single "correct" or universally optimal resolution for brain parcellations due to the brain's hierarchical organization (Schaefer et al., 2018; Churchland and Sejnowski, 1988). Even well-defined cortical areas can exhibit significant functional heterogeneity and can often be meaningfully subdivided (e.g., different somatotopic representations within motor cortex, or eccentricity-based divisions within V1) Schaefer et al. (2018).

Given that our SVR-LSM approach can identify focal effects, a higher-resolution parcellation like the Schaefer 1000 allows for a more precise localization of lesion overlap when the significant SVR-LSM cluster (or its sub-components) might align with finer functional distinctions within broader network regions compared with less granular parcellations. This helps in understanding which specific parts of larger networks are implicated.

While our primary reporting focuses on the 17 large-scale networks (by aggregating contributions from the 1000 parcels), the finer-grained data remains available and supports a more detailed anatomical interpretation if specific parcels show particularly high lesion overlap within our ROI.

Schaefer et al. (2018) themselves note that "*different resolution parcellations might be useful for different applications,*" and suggest that "*if the effect of interest is highly focal... then a higher... resolution parcellation might be more effective.*"

We have justified this choice in the Methods section (lines 666-670).

3) We thank the Reviewer for suggesting the network-symptom mapping (NSM) approach developed by Boes and colleagues. We agree that NSM offers a powerful method to identify functional resting-state networks whose disruption by lesion is associated with specific symptoms.

Our current study's primary SVR-LSM analysis employed a voxel-based exploratory approach to identify brain regions whose direct damage is associated with the postoperative visuospatial deficit. A key novelty of our paper lies in demonstrating the convergence (spatial co-localization) between this SVR-LSM derived cluster and the intraoperative functional mapping results (specifically, the probability density estimation of sites eliciting neglect-like errors during DES).

Within this framework, the subsequent LQT analysis, using the significant SVR-LSM cluster as input, serves as a post-hoc characterization. Its purpose is to describe the large-scale functional network affiliations of this empirically identified critical region, thereby providing richer functional context than a simple anatomical labeling of the SVR-LSM cluster alone.

Furthermore, to provide a highly detailed anatomical characterization of the white matter pathways constituting this critical SVR-LSM region – information that is also directly relevant for neurosurgical planning aimed at preserving these connections – we implemented an extensive structural connectome analysis (detailed in sections Methods-retrospective study 1.6-1.7 lines 674-803). This involved using the SVR-LSM cluster to define a normative connectome from HCP data (identifying consensus hubs and edges) and then assessing the clinical impact of disconnecting these specific edges in our patient cohort. This approach provides a granular, pathway-specific understanding of the structural network implicated by the SVR-LSM findings.

Implementing a full NSM pipeline, as described by Boes and colleagues, would represent a substantial new analytical undertaking for this revision, requiring different input data and modeling approaches. While this is undoubtedly a valuable research direction, we believe it extends beyond the primary scope of the current paper, which focuses on the aforementioned convergence of lesion-based SVR-LSM and intraoperative DES findings. We plan to explore NSM approaches in future work; indeed, we are currently acquiring preoperative and postoperative patient-specific resting-state fMRI data, which will allow us to perform functional NSM using both normative and patient-specific connectomes to account for potential glioma-induced network modifications.

Schaefer A, Kong R, Gordon EM, Laumann TO, Zuo XN, Holmes AJ, Eickhoff SB, Yeo BTT. Local-Global Parcellation of the Human Cerebral Cortex from Intrinsic Functional Connectivity MRI. *Cereb Cortex*. 2018 Sep 1;28(9):3095-3114. doi: 10.1093/cercor/bhx179.

Churchland PS, Sejnowski TJ. Perspectives on cognitive neuroscience. *Science*. 1988 Nov 4;242(4879):741-5. doi: 10.1126/science.3055294.

#7 The authors wrote: “Given that the SVR-LSM revealed that a postoperative worsening in visuospatial selective attention was associated only with a right frontal cluster (see Results), all the subsequent analyses were performed on right hemisphere patients.” Is this really a valid argument? The absence of significant results in SVR-LSM based on topographies (resection locations) does not necessarily imply that SVR-LSM based on disconnections would also yield non-significant results. In my opinion, it would be more relevant to perform the analysis and explicitly demonstrate that no significant results are found.

We thank the Reviewer for this important question regarding the analysis of disconnection effects in the left hemisphere.

First, we would like to clarify a methodological point that may not have been entirely clear in our initial submission (an aspect also noted by Reviewer 2). Our SVR-LSM analyses

(using resection cavities as input) were performed separately for patients with left hemisphere lesions and right hemisphere lesions. This separation is consistent with standard practice in lesion-symptom mapping aimed at maximizing statistical power, as analyzing unilateral lesions together can otherwise weaken observed associations within each respective hemisphere (Röhrig et al., 2023; Frenkel-Toledo et al. 2019, 2021). Please note that this is now better clarified in both the Methods and the Results (Retrospective study - Methods: 1.5 Lesion symptom mapping lines 633-639; Results: 1.3 Lesion-symptom mapping - SVR-LSM lines 184-207).

The lesion-based SVR-LSM for the left hemisphere group did not yield any significant associations with either the Δ total or Δ asymmetry scores of the Bell's Test. For this reason, we decided to not perform SVR-DSM in this cohort of patients. However, we agree with the Reviewer that the absence of significant results in a resection-based SVR-LSM does not automatically preclude finding significant results with a disconnection-based SVR-LSM (SVR-DSM), as the inputs are indeed different.

For this revision, we have therefore performed exploratory SVR-DSM (input = disconnection maps) with the same parameter used in the main SVR-LSM in the left cohort using the Δ Bell's Test total score and Δ asymmetry score as dependent variables.

Consistent with the resection-based SVR-LSM, this exploratory SVR-DSM for the left hemisphere lesion group also yielded no significant voxel clusters associated with either behavioral outcome.

For the right hemisphere, the disconnection-based SVR-DSM produced a focal cluster largely overlapping with the resection-based SVR-LSM findings (see first submission).

Ultimately, as we detail further in our response to point #12, we found the SVR-DSM approach less informative than anticipated for elucidating specific pathways, particularly given the focal nature of its findings even in the right hemisphere. To achieve greater granularity and better insight into the white matter connections, we therefore replaced the SVR-DSM analysis entirely with a region-to-region connectome-based approach (normative HCP connectome definition followed by patient-specific edge disconnection analysis). (For a full explanation of the reasoning behind this methodological shift, please see our response to your introductory remarks on this revision and our reply to point #12).

Therefore, while we have confirmed through this additional exploratory SVR-DSM that the left hemisphere did not show significant disconnection-symptom mapping results with that particular method, these SVR-DSM analyses are no longer part of the revised manuscript. Our current manuscript focuses on the SVR-LSM (lesion-based) to define the initial critical region (which was right-hemisphere specific) and the subsequent detailed connectome and edge-disconnection analyses.

Frenkel-Toledo S, Fridberg G, Ofir S, Bartur G, Lowenthal-Raz J, Granot O, Handelzalts S, Soroker N. Lesion location impact on functional recovery of the hemiparetic upper limb. *PLoS One*. 2019 Jul 19;14(7):e0219738. doi: 10.1371/journal.pone.0219738.

Frenkel-Toledo S, Ofir-Geva S, Mansano L, Granot O, Soroker N. Stroke Lesion Impact on Lower Limb Function. *Front Hum Neurosci*. 2021 Feb 1;15:592975. doi: 10.3389/fnhum.2021.592975. PMID: 33597852; PMCID: PMC7882502.

#8 Regarding “the disconnection lesion-symptom mapping” part. Do the authors ensure that the analyses performed using the De Marco’s toolbox were configured to handle continuous lesion data appropriately?

We agree that this needs clarification. The probabilistic disconnection maps were indeed binarized prior to being input into the SVR-LSM toolbox. Consistent with common practice (e.g., Billot et al., 2022; Foulon et al., 2018), a voxel was considered 'disconnected' (value of 1 in the binary mask) if having a $\geq 50\%$ probability of being disconnected. As detailed in our response to Point #12 and in our introductory comments to this revision, the SVR-DSM approach is no longer included in the current manuscript, having been replaced by a connectome-based edge disconnection analysis for greater anatomical specificity.

Billot A, Thiebaut de Schotten M, Parrish TB, Thompson CK, Rapp B, Caplan D, Kiran S. Structural disconnections associated with language impairments in chronic post-stroke aphasia using disconnectome maps. *Cortex*. 2022 Oct;155:90-106. doi: 10.1016/j.cortex.2022.06.016.

Foulon C, Cerliani L, Kinkingnéhun S, Levy R, Rosso C, Urbanski M, Volle E, Thiebaut de Schotten M. Advanced lesion symptom mapping analyses and implementation as BCBToolKit. *Gigascience*. 2018 Mar 1;7(3):1-17. doi: 10.1093/gigascience/giy004.

#9 Regarding this part: “Identification of disconnected white matter pathways. To label the white matter tracts associated with the deficit, i.e. the subcortical region associated with the impairment in selective visuospatial attention, the significant SVR-LSM and SVR-DSM cluster were merged and compared with the Human Connectome Project tractography atlas (Yeh et al., 2021) by means of tractotron as part of BCBToolKit (Foulon et al., 2018). At each voxel of the SVR clusters, the probability of finding a white matter tract of the HCP atlas was computed. Only tracts reliably disconnected (>50% of probability) by the SVR clusters were considered.

I’m not entirely clear on what has been done here. From my understanding, Tractotron provides a probability that a given tract is damaged based on a lesion map (i.e., if one voxel of a lesion map intersects a fiber tract with 100% overlap in the normative data, the tract is considered 100% disconnected). Do the authors mean that the analyses were performed iteratively on each voxel within the significant clusters identified by SVR-LSM and then averaged? In the table provided in the supplementary file, no measure of variability is provided.

We thank the Reviewer for requesting clarification on our use of Tractotron (BCBToolKit; Foulon et al., 2018) for identifying white matter tracts associated with our significant SVR-LSM cluster. We apologize if our initial description was unclear.

The procedure was as follows:

- 1) The significant SVR-LSM cluster was used as the single input 'lesion' mask to Tractotron.
- 2) This SVR-LSM ROI was compared against the Human Connectome Project probabilistic atlas (Yeh et al., 2021). In this atlas, each voxel has an associated probability of containing streamlines belonging to various predefined white matter tracts.
- 3) By means of Tractotron we determined, for each tract in the HCP atlas, whether it was 'implicated' or 'disconnected' by our SVR-LSM ROI. Consistent with the methodology described by Foulon et al. (2018) for Tractotron, a tract was considered 'reliably disconnected' if our SVR-LSM ROI overlapped with voxels where that specific tract had a probability of being present greater than 50%.

The analysis was not performed iteratively on each voxel within the SVR-LSM cluster and then averaged. Instead, Tractotron assesses the intersection of the entire SVR-LSM ROI with each probabilistic tract definition from the atlas. The table previously included in our supplementary file listed the white matter tracts from the HCP atlas that met this >50% probability of intersection criterion with our SVR-LSM cluster.

However, upon significant revision of our pipeline to achieve a more quantitative and granular understanding of specific pathway involvement, this Tractotron-based analysis is no longer included in the current manuscript. We found that our newly implemented connectome-based approach (detailed in Methods - Retrospective study - sections 1.6-1.7 lines 674-803), which involves defining a normative structural connectome filtered by the SVR-LSM cluster and then assessing patient-specific disconnections of consensus edges, provides a far more informative characterization.

We have revised the Methods section to reflect this updated approach, and the previous Tractotron analysis and its supplementary table have been removed.

#10 The authors combined data from patients with lower-grade and high-grade gliomas. As the authors are aware, the mechanisms leading to cognitive deficits in these two types of tumors are distinct. Why not control for this difference in the statistical models? This adjustment is easily feasible, especially given the capabilities of the toolbox to perform GLM/nuisance models.

Following the Reviewer's suggestion, we have incorporated age, education and tumour grade as covariates in the lesion-symptom mapping (LSM) analysis to assess whether these variables influenced the results. The inclusion of these factors did not alter the main findings, confirming the robustness of our results. This suggests that the observed effects are primarily driven by lesion location rather than demographic or tumour grade. The use of covariates is now reported in the manuscript (Methods section - Retrospective study - 1.5 Lesion symptom mapping lines 633-647).

#11 In connection with the previous comment, high-grade gliomas, more so than lower-grade gliomas, cause significant distortions of anatomical structures, particularly white matter tracts. This inevitably introduces some bias into the results of SVR-LSM analyses, whether focused on resection topographies or disconnection maps. How confident are the authors that these distortions do not unduly influence their findings?

We agree with the Reviewer that this is an inherent limitation of LSM studies involving glioma patients.

Based on different observations, we exclude that our main results are significantly biased by anatomical dislocation of normal anatomy:

1) In our primary SVR-LSM analysis, we explicitly addressed the heterogeneity of our patient cohort by including tumor grade as a covariate in the regression model, alongside age, education, and resection volume. By statistically controlling for tumor grade, our model accounts for variance attributable to differences between lower- and high-grade gliomas. This ensures that the identified brain regions are associated with the behavioral deficit independent of the specific histopathological factors that contribute to structural distortion.

2) To validate our SVR-LSM findings, we implemented a multi-step approach that bridges normative connectomics with patient-specific clinical data. We first used the SVR-LSM cluster, identified from our retrospective cohort, as a seed to define a "visuospatial neglect network" in a large dataset of 100 healthy Human Connectome Project (HCP) participants. On this normative dataset, we applied graph theory analysis to identify the network's most critical hubs and connections, termed "consensus edges". By leveraging patient-specific tractography, we then aimed to directly account for pathology-induced distortions and to increase the specificity of the results (Methods - Retrospective study - 1.7 lines 760-803). This validation was performed in each patient's native diffusion space using tractography methods specifically adapted to mitigate tumor-induced distortions. We found that disconnection across the consensus edges generally showed a positive correlation trend with postoperative neglect severity (Figure 3), confirming that the anatomical pathways highlighted in our normative analysis are anatomically plausible in individual patients. This analysis was further refined by identifying the most critical connections. Specifically, the disconnection of three pathways—the right SCEF-thalamus, right SCEF-brainstem, and the transcallosal right SFL-left SCEF—was most strongly and significantly associated with the deficit after Bonferroni correction (Results - Retrospective study section 1.5 lines 266-279). This result validates the clinical relevance of the overall network while also pinpointing the core connections whose disruption most severely impacts visuospatial neglect.

3) Finally, the most important evidence is the spatial convergence between the SVR-LSM cluster associated with visuospatial neglect and the intraoperative DES sites that elicited transient neglect-like errors. This provides powerful validation that the SVR-LSM results are not a methodological artifact of anatomical distortion (Results lines 330-338, Figure 5GH). It is highly improbable that a systematic bias in our lesion analysis (retrospective cohort) would lead to identify a cluster that also matches the location of functionally

eloquent sites identified through an entirely different method (DES) in a separate patient cohort (prospective cohort).

#12 I wonder whether it would be possible to perform an analysis that estimates the pathological variance explained by resection topographies versus the variance explained by structural disconnections. This issue is increasingly being discussed in the lesion-symptom mapping literature. I mention this because the SVR-LSM results highlighted a relatively circumscribed white matter area, whereas it would be somewhat expected for this area to be more widespread.

We thank the Reviewer for this question.

In our revised manuscript, we aimed to address the Reviewer's earlier comments about achieving greater granularity in identifying specific white matter connections. To this end, we transitioned from our initial approach to a comprehensive connectome-based analysis. It is pertinent to note that our first submission included an SVR-DSM based on whole-brain disconnection maps generated using BCBtoolkit "Disconnection map" tool. For these maps, a voxel was considered 'disconnected' in a patient if the patient's resection cavity overlapped with streamlines passing through that voxel in at least 50% of the healthy control subjects within the normative atlas (a commonly used threshold, e.g., Billot et al., 2022; Foulon et al., 2018). This SVR-DSM yielded a surprisingly focal cluster, not pointing to the involvement of a more diffuse white matter pattern.

We believe the primary reason for this focal SVR-DSM result relates to the high degree of similarity in the overall disconnection profiles across a large majority of patients. Our analyses (e.g., Fig. 2A of the first submission) indicated that over 90% of patients shared a very similar pattern of widespread disconnection. In such a scenario the input features (disconnection maps) were highly overlapping across all subjects (independently by the behavioral score). Therefore, this low variance prevented the disclosure of significant voxels in other white matter regions.

Moreover, the SVR-DSM cluster was highly consistent/overlapping with the one highlighted by SVR-LSM using resection cavities, therefore not helping in describing a large-scale disconnection profile possibly associated with the deficit nor adding additional information.

Therefore, to gain deeper and more specific insights into the *large-scale disconnection profiles* possibly associated with the deficit, we replaced the SVR-DSM analysis with a region-to-region approach aimed at describing, with high granularity, the white matter "disconnected" by the SVR-LSM cluster. To this aim, we characterized the normative structural connectome (using HCP data and graph theory) of all white matter pathways passing through this region, identifying key 'consensus hubs' and 'consensus edges' that constitute this large-scale network (Methods - Retrospective study - 1.6 Normative Visuospatial Neglect Structural Connectome lines 674-747). Finally, for our patient cohort (Methods - Retrospective study - 1.7.1 Validation of consensus edge disconnection against postoperative deficits lines 787-803), we quantified the percentage of surgical

disconnection for each of these specific consensus edges using their patient-specific preoperative tractography and correlated this with the postoperative change in visuospatial attention.

This revised connectome-based approach allows us to move beyond a single focal SVR-DSM finding by identifying multiple critical edges within a broader network.

We believe our revised connectome-based methodology, by focusing on the disconnection of specific, functionally relevant pathways (quantified from normative and patient-specific data), offers a significant improvement compared to the SVR-DSM analysis presented in the first submission.

Billot A, Thiebaut de Schotten M, Parrish TB, Thompson CK, Rapp B, Caplan D, Kiran S. Structural disconnections associated with language impairments in chronic post-stroke aphasia using disconnectome maps. *Cortex*. 2022 Oct;155:90-106. doi: 10.1016/j.cortex.2022.06.016.

Foulon C, Cerliani L, Kinkingnéhun S, Levy R, Rosso C, Urbanski M, Volle E, Thiebaut de Schotten M. Advanced lesion symptom mapping analyses and implementation as BCBtoolkit. *Gigascience*. 2018 Mar 1;7(3):1-17. doi: 10.1093/gigascience/giy004.

#13 For the ANOVAs, could the authors provide measures of effect size?

We incorporated in the results section the effect size estimates obtained from the partial eta squared and Cohen's d for all significant effects (Results - Retrospective study - 1.2 Neuropsychological Assessment lines 150-182).

#14 For a comprehensive visualization of the results, it would be beneficial for the readers if all individual data points (e.g., pre- vs. post-surgery; left vs. right hemisphere) were plotted. This would provide greater clarity and transparency in illustrating the variability and patterns within the dataset.

Following the reviewer's recommendation, we have added violin plots displaying individual data points for pre- vs. post-surgery performance, as well as comparisons between left and right hemisphere lesions. These visualizations provide a clearer representation of intra-individual variability and overall data distribution (Supplementary Fig.1,2,4,5,).

#15 In their analyses, the authors did not provide any data to evaluate whether patients, on average, differed from neurologically intact participants before surgery or the proportion of deficits based on established normative data. Including such comparisons may provide a clearer context for assessing the LSM results.

Thank you for raising this relevant point. In response, we have added a paragraph in the Results/Methods sections that describes the number of patients with pathological scores, including relevant references to normative data (Methods - Retrospective study - 1.3 Neuropsychological assessment and statistical analysis lines 615-616) / (Results - Retrospective study - 1.2 Neuropsychological Assessment lines 150-182) / (Results - Prospective study - 2.5 Effect of Intraoperative Visuospatial Selective Attention Task

(iVSAT) on Postoperative Visuospatial Neuropsychological Outcomes lines 320-338). Moreover, pathological scores are highlighted in bold in Supplementary table 1 and 3.

#16 In the figures, the resection density maps range from 1 to the maximum. It would be more effective to threshold the maps to display only the voxels that were included in the analyses.

In Figure 1, the maps showing overlap between patients' resection cavities now display only those voxels included in the analysis. Please note that, in accordance with suggestions from Reviewers 2 and 3, the SVR-LSM now includes only voxels resected in at least 10% of participants (i.e., eight patients).

#17 In my opinion, it would have been interesting to perform an analysis examining the association between the FEF (frontal eye field) identified through electrostimulation and visuo-spatial performance. For example, is there a relationship between the distance from the resection to the FEF and the severity of visuo-spatial deficits? This could provide additional insights into the functional relevance of the FEF in visuo-spatial attention.

We thank the Reviewer for this interesting suggestion to examine the relationship between the distance from the resection cavity to the functionally-defined Frontal Eye Field (FEF) and the severity of visuospatial deficits. While exploring such spatial relationships can be informative, we have several considerations based on our current findings that lead us to believe this specific analysis might not yield substantial additional insights beyond our current approach:

1) Our SVR-LSM analysis identified a significant cluster in the right frontal lobe, immediately anterior to the intraoperatively mapped FEF, as being associated with postoperative visuospatial deficits (Δ Bell's Test asymmetry). Given this, any lesion impacting this SVR-LSM cluster will inherently be in close proximity to the FEF. Conversely, lesions distant from this critical frontal zone (e.g., temporal) would naturally have a greater distance to the FEF and are also less likely to cause this specific frontal-type deficit. Therefore, a simple correlation between 'distance to FEF' and deficit severity would likely be confounded by the broader anatomical location of the lesions (i.e., frontal vs. non-frontal).

2) Relying on a single 'distance to FEF' metric also presents several inherent limitations compared to comprehensive voxel-based approaches like SVR-LSM. Such distance measures reduce complex three-dimensional lesion information to a single, potentially noisy value, thereby losing spatial specificity and statistical power (Bates et al., 2003; Rorden & Karnath, 2004; Medina & Fischer-Baum, 2017)

3) An examination of our retrospective cohort's resection overlap map (Fig. 1A-B) reveals that the region corresponding to the functionally-defined FEF (Fig. 5F) was largely spared from direct resection in most patients (see Revision Figure 2). This demonstrates that the visuospatial neglect deficits captured by our analysis emerged even when the FEF core remained intact. While the role of the FEF in visuospatial attention is established, our findings suggest its direct resection is not a necessary condition to produce neglect. This reinforces our primary SVR-LSM finding: the critical causal region lies immediately anterior to the FEF, and

the visuospatial deficits likely arise from damage to this specific area and its associated pathways.

4) To further explore the network context of the SVR-LSM findings and the potential role of regions like the FEF, we have implemented a new structural connectome analysis in this revised manuscript. This analysis characterized the white matter network passing through the significant SVR-LSM cluster. Interestingly, this connectome mapping did not identify the FEF (as defined by the HCP-MMP1.0 atlas parcel, which is highly overlapping with the FEF intraoperative PDE) as a primary consensus hub, nor were its direct connections among the most prominent 'consensus top edges' within this specific neglect-associated network. While the FEF is undeniably crucial for oculomotor control, our findings suggest that the critical network components underlying the observed visuospatial neglect deficits (as captured by the SVR-LSM) are centered on other nearby frontal regions (e.g., R-SCEF, R-6ma, R-8Av, R-i6-8; see Results sections 1.4/1.5 and Figs. 2 and 3 lines 219-279) and their specific connectivity profiles. The SVR-LSM cluster itself may represent a distinct, critical modulatory area or pathway convergence point whose specific connectivity, rather than its mere distance to the FEF, is key for understanding the observed deficits.

Given these considerations, we believe our current methodology—employing SVR-LSM to identify critical regions and then characterizing their network connectivity via structural connectome analysis—offers pathway-specific insights with greater neurobiological specificity than would be achievable with a 'distance to FEF' analysis alone.

For the Reviewer's interest, we report here the Spearman correlations between the minimum Euclidean distance—measured between the intraoperatively defined FEF probability density estimate (PDE) and the resection cavities—and both the Δ Bell's Test total score (the postoperative change in total omissions) and the Δ Bell's Test asymmetry score (the postoperative change in the asymmetry score).

Left hemisphere:

Δ Bell's Test total score: $\rho = 0.27935$, $p = 0.01155$.

Δ Bell's Test asymmetry score: $\rho = -0.0174$, $p = 0.87748$.

Right hemisphere:

Δ Bell's Test total score: $\rho = 0.22217$, $p = 0.04485$.

Δ Bell's Test asymmetry score: $\rho = 0.37086$, $p = 0.0006$.

These results were predictable considering the location of the SVR-LSM cluster, but in light of the above arguments, we do not believe they should be included in the main text. If the reviewers deem it important to include this data, we will be happy to do so.

Reviewer #2 (Remarks to the Author):

This manuscript contributes significantly to understanding the frontal connectivity supporting visuospatial attention, particularly through the novel integration of DSM and DES. The findings about the dorsomedial and ventrolateral white matter contributions are interesting and backed by interesting analyses. However, several methodological choices (e.g., filtering DSM to right-hemisphere patients, voxel inclusion thresholds) need clearer justification. Additionally, some limitations should be discussed more thoroughly so future research can investigate them.

If these concerns are addressed, the manuscript has the potential to make an impactful contribution to the field.

Comments:

Introduction

"This strategy, grounded in recent guidelines (Siddiqi et al., 2022), provides the highest degree of causal inference in human brain mapping studies." Considering the shortcomings of LSM mentioned above, I believe this is an overstatement.

We agree that the term 'highest degree of causal inference' is a strong statement that requires justification.

The premise of the Reviewer's concern relates to the shortcomings of LSM we mentioned. We would like to clarify that our statement in the introduction was related to the well-documented localization bias inherent to stroke-based LSM. A strength of our study design was to use a brain tumour cohort, which overcomes this specific vascular territory bias and allows for a more uniform sampling of lesion locations.

More importantly, our claim of a high degree of causal inference is based not on LSM alone, but specifically on our convergent causal mapping strategy, which integrates both LSM and DES. This approach directly aligns with the framework proposed by Siddiqi et al. (2022) in their "Causal mapping of human brain function" review. In their proposed 'causality continuum' (see Fig. 2b - Siddiqi et al. 2022), the authors explicitly create a top tier for 'Convergent causal mapping', assigning it the highest possible score (6.5/6.5). They state that demonstrating such cross-modal coherence (e.g., between lesions and stimulation) enables stronger causal inference than any single modality alone. Our study, by demonstrating that permanent deficits from surgical lesions (LSM) and transient deficits from stimulation (DES) converge on the same anatomical substrate, is a direct

implementation of the methodology that these guidelines identify as providing the most robust causal evidence.

Nevertheless, in light of the Reviewer's valid point and to ensure our claim is not perceived as an overstatement, we have moderated the language in the introduction. Please note that, following suggestion from Reviewer 3 (minor point #2), the introduction has also been updated to include a balanced acknowledgement of the limitations of DES. The revised text now reads: "*DES, despite its inherent limitations (e.g. the potential spread of current to neighbouring and/or remote regions, (Borchers et al., 2012; Duffau, 2015)), provides a causal reversible probe to test anatomo-functional relationship with a unique invasive access to the human brain. The strategy of combining these two causal methods allows their respective limitations to be mitigated. By demonstrating concordance between permanent deficits from surgical lesions (LSM) and transient deficits from DES at the same anatomical locations, a high degree of causal inference can be achieved lines 103-109 (Siddiqi et al. 2022)*".

Siddiqi, S. H., Kording, K. P., Parvizi, J., & Fox, M. D. (2022). Causal mapping of human brain function. *Nature Reviews Neuroscience*, 23(6), 361–375. <https://doi.org/10.1038/s41583-022-00583-8>

The first time the "asymmetry score" is mentioned, there is no explanation of that score. I know it's standard in visual neglect, but I think the concept is simple enough to be very briefly explained in one sentence or even a sentence fragment to avoid unfamiliar readers having to look it up. A few sentences later, it is mentioned "(i.e. visuospatial neglect)," but while clarifying a bit, it doesn't say whether a low or high score is good.

We thank the Reviewer for pointing this out. At first mention of the asymmetry score we have now added a brief definition—"*left minus right omissions; higher scores denote more severe visuospatial neglect*" (Results - Retrospective study - 1.2 Neuropsychological assessment lines 150-156).

Results

Am I correct in understanding that "total missed target" becomes "total score" and then "total missed target score" and goes back to "total score"? I think it would be easier to clarify/harmonise that.

Thank you for the suggestion. We harmonised the used terms: after the first definition of "total score" (Results - Retrospective study - 1.2 Neuropsychological assessment: "*total score (total missed targets)*" lines 150-156) we used this definition throughout the whole text.

"parcel-level lesion load analysis" the reported values make it hard to evaluate the results. Are all the parcels the same volume? If so, it would seem that the control network has the most cumulated volume of damage from SVR-LSM. From a quick look at the discussion, the other two networks are the stars of the show, not this one.

The Schaefer atlas parcels, used in our Lesion Quantification Toolkit (LQT) analysis, are not of equal volume, as their definition is based on functional homogeneity rather than spatial extent (Schaefer et al. 2018).

To address the Reviewer's concern and to directly assess the proportional impact on each of the 17 large-scale Yeo networks, we have now calculated and reported the percentage of each entire large-scale network's total volume that is overlapped by our SVR-LSM significant ROI.

This revised analysis reveals the following percentage overlaps between our SVR-LSM significant ROI and the total volume of large-scale networks:

- Dorsal Attention Network B (DAN B): 0.84%
- Control Network B (Cont B): 0.32%
- Salience Ventral Attention Network A (VAN A): 0.18%

The Methods section (Retrospective study - 1.5 Lesion symptom mapping lines 661-673) and the Results (Retrospective study - 1.3 Lesion symptom mapping lines 208-217) section are now updated to include this analysis. Accordingly, we have now revised the Discussion section to reflect these findings and to clarify the interpretation of the network involvement.

"Disconnectome symptom mapping" I'm a bit sceptical about filtering the DSM analysis to only right-hemisphere patients based on the LSM results. The two analyses are independent, so I don't see why LSM findings justify excluding left-hemisphere patients. This approach could miss potential contributions from commissural tracts, like the corpus callosum, which might play a role in visuospatial attention. It also reduces the sample size and might explain why the DSM cluster is surprisingly small and similar to the LSM cluster—something I wouldn't expect given the different inputs. Including left-hemisphere patients could have provided a more complete picture, even if they didn't show significant effects.

We thank the Reviewer for raising this point regarding our analysis of the left hemisphere and the potential role of commissural tracts. We agree that our rationale requires a more detailed explanation.

First, we wish to clarify that our initial SVR-LSM analyses were performed separately for the left and right hemisphere cohorts, a standard approach to maximize statistical power for unilateral lesions (Röhrig et al., 2023; Frenkel-Toledo et al. 2019, 2021). The decision to focus subsequent analyses on the right hemisphere was therefore driven by the initial, hemisphere-specific finding that only right-sided resections were significantly associated with postoperative visuospatial neglect. Please note that this is now better clarified in both the Methods and the Results (Retrospective study - Methods: 1.5 Lesion symptom mapping lines 633-639; Results: 1.3 Lesion-symptom mapping - SVR-LSM lines 184-188).

However, we fully agree with the Reviewer's comment that the absence of a significant resection-based result does not rule out the possibility of a significant disconnection-based one. To directly address this, we performed an exploratory SVR-DSM on the left-hemisphere cohort for this revision. Consistent with our SVR-LSM results, this additional analysis also yielded no significant voxel clusters associated with visuospatial deficits in the left hemisphere (nor using the Δ Bell's Test total score nor the Δ asymmetry score).

Ultimately, we found the SVR-DSM approach to be less informative than anticipated, even in the right hemisphere. The analysis produced a surprisingly focal cluster that was largely redundant with the SVR-LSM result. We believe this is because the widespread and highly similar disconnection patterns across most patients provided insufficient variance for the SVR-DSM model to identify specific pathways. Our analyses (e.g., Fig. 2A of the first submission) indicated that over 90% of patients shared a very similar pattern of widespread disconnection. In such a scenario the input features (disconnection maps) were highly overlapping across all subjects (independently by the behavioral score). Therefore, this low variance prevented the disclosure of significant voxels in other white matter regions.

For this reason, and to achieve greater anatomical specificity (please see also requests of Reviewer 1), we replaced the SVR-DSM analysis entirely with a region-to-region connectome-based methodology aimed at describing the white matter network associated with visuospatial neglect. Briefly, This methodology uses the SVR-LSM cluster to seed connectomes (100 HCP unrelated dataset). We then applied graph theory to identify the most critical network hubs and connections ('edges'). We then validated the clinical relevance of these specific pathways in our patient cohort by correlating their disconnection with postoperative deficits. This approach provided a more granular, network-based model and, crucially, allowed us to directly assess the role of commissural pathways, identifying a key transcallosal connection whose disconnection was significantly associated with neglect. To complement this, a population-average track-density imaging (TDI) map visualized the anatomical core of this network, showing the highest fiber density in the white matter below the superior and middle frontal gyri comprising intra-frontal connections extending towards the corpus callosum and the internal capsule. (See Methods - Retrospective study - 1.6/1.7 lines 674-803).

In summary, our revised approach confirms the right-hemisphere specificity of the deficit while providing a far more granular and informative analysis of the specific pathways involved, including the callosal connections the Reviewer rightly highlighted.

Röhrig L, Rosenzopf H, Wöhrstein S, Karnath HO. The need for hemispheric separation in pairwise structural disconnection studies. *Hum Brain Mapp.* 2023 Nov;44(16):5212-5220. doi: 10.1002/hbm.26445.

Frenkel-Toledo S, Fridberg G, Ofir S, Bartur G, Lowenthal-Raz J, Granot O, Handelzalts S, Soroker N. Lesion location impact on functional recovery of the hemiparetic upper limb. *PLoS One.* 2019 Jul 19;14(7):e0219738. doi: 10.1371/journal.pone.0219738.

Frenkel-Toledo S, Ofir-Geva S, Mansano L, Granot O, Soroker N. Stroke Lesion Impact on Lower Limb Function. *Front Hum Neurosci.* 2021 Feb 1;15:592975. doi: 10.3389/fnhum.2021.592975.

Figures 1, 2 and 6 have inconsistent display conventions and no orientation indications. This might confuse a lot of readers and needs to be fixed.

All figures are now amended using the neurological convention and explicitly labelling the orientations.

Why is the full name of the Bells test mentioned only at this point, after "asymmetry score" has been mentioned multiple times?

We amended the manuscript mentioning the Bells test before the asymmetry score (Results - Retrospective study - 1.2 Neuropsychological Assessment lines 150-156).

I have no idea what I am supposed to see in Figure 3. Why is the Bells Test in the middle? Why is it a correlation matrix between a set of tracts and only one variable? What are the values between the tracts? Correlations between pairs of disconnected tracts?

We apologize for the unclear presentation of the figure. The correlation displayed was between the amount of disconnection of tracts and the Δ Bell's Test asymmetry score. The other values between tracts were the correlations between the amount of disconnection across tracts. Please note that this analysis is no longer present in the paper, as it was substituted by the analysis described in Methods section 1.7/1.7.1 Validation of consensus edge disconnection against postoperative deficits. lines 760-803.

"White matter disconnection: tractography in single patients" this paragraph needs to be rewritten. There is not enough information at this point of the article to understand where these results come from. "tracts correlated with the Δ asymmetry score Bells Test": It is the percentage of disconnections (number of streamlines, which I am not sure is a reliable measure of disconnections or connection for that matter) correlated to the score, not "the tracts". And this is not explained here. Initially, I thought the disconnection measure was a binary variable (disconnected or not).

We thank the Reviewer for this feedback and for highlighting the need for greater clarity in our single-patient tractography analysis. We also agree that quantifying structural connectivity requires careful consideration, particularly as the density of reconstructed streamlines is not necessarily indicative of the underlying axonal connections (Jones et al. 2013; Smith et al. 2015).

To address this, the tractography analysis has been completely revised in the current manuscript, and the methods and results are now introduced more clearly in their respective sections to improve clarity (Methods - Retrospective study - 1.7/1.7.1 lines 760-803) (Results - Retrospective study - 1.5 Clinical validation of consensus edges via patient-specific disconnection modelling lines lines 266-279)

Specifically, we have replaced the previous analysis with a state-of-the-art quantitative approach. We no longer use streamline counts. Instead, connectivity is quantified using the Fibre Bundle Capacity (FBC). This measure is derived by first processing the whole-brain tractograms with the Spherical-deconvolution Informed Filtering of Tractograms (SIFT2) model (Smith et al., 2015). This assigns a cross-sectional area multiplier to each individual streamline, which ensures that the resulting tractogram is quantitatively representative of the underlying fibre densities estimated from the diffusion data.

The FBC for any given edge (connection between two grey-matter parcels) is the sum of the SIFT2 weights of the streamlines connecting the two nodes multiplied by the SIFT2 subject-specific μ coefficient, which provides an estimate of the total intra-axonal cross-sectional area of the pathway in absolute units (e.g., mm^2), a metric with a clear biological interpretation (Smith et al., 2022; Tahedl et al. 2025).

The network connections we investigated in this patient-specific analysis were the 'consensus edges' derived from our normative connectome study (for the normative analysis, see Methods - Section 1.6 674-747 Results 1.4/1.4.1 lines 219-255). These edges represent the most consistently important structural connections of the neglect-associated network identified from our graph theory analysis of 100 healthy HCP participants.

Therefore, the correlation reported is not between a deficit and "the number of streamlines within tracts" themselves, but between the deficit and the percentage of FBC reduction for each of these specific consensus edges. This disconnection value was calculated for each patient by comparing the preoperative FBC of an edge with its FBC in a postoperatively-simulated state (i.e., after removing streamlines and SIFT2 weights passing through the resection cavity).

We are confident that this revised, quantitative methodology directly addresses the Reviewer's concerns and provides a more reliable and biologically meaningful assessment of white matter disconnection.

Jones DK, Knösche TR, Turner R. White matter integrity, fiber count, and other fallacies: the do's and don'ts of diffusion MRI. *Neuroimage*. 2013 Jun;73:239-54. doi: 10.1016/j.neuroimage.2012.06.081.

Smith RE, Tournier JD, Calamante F, Connelly A. SIFT2: Enabling dense quantitative assessment of brain white matter connectivity using streamlines tractography. *Neuroimage*. 2015 Oct 1;119:338-51. doi: 10.1016/j.neuroimage.2015.06.092.

Smith, Robert E., et al. "Quantitative Streamlines Tractography: Methods and Inter-Subject Normalisation." *Aperture Neuro*, vol. 2, Apr. 2022, pp. 1–25, <https://doi.org/10.52294/ApertureNeuro.2022.2.NEOD9565>.

Tahedl, M., Tournier, J.-D., & Smith, R. E. (2025). Structural connectome construction using constrained spherical deconvolution in multi-shell diffusion-weighted magnetic resonance imaging. *Nature Protocols*. <https://doi.org/10.1038/s41596-024-01129-1>

Prospective study:

This part of the results is very confusing. The left and right hemispheres are not always consistently mentioned.

Thank you for the comment. We made more clear the distinction between left and right results.

Discussion

"atlas-based disconnectome" I believe disconnectome is not "atlas-based".

The term "atlas-based disconnectome" was imprecise. We would like to clarify that this methodology is no longer part of the revised manuscript (SVR-DSM).

In response to feedback from the review process requesting greater anatomical granularity, we have replaced the previous SVR-DSM analysis with a region-to-region connectome-based approach. As detailed in the revised manuscript, our new analysis pipeline consists of two main steps:

We first define the network structurally associated with the SVR-LSM cluster by building a normative connectome using data from 100 healthy HCP participants. Using graph theory, we identify the key 'consensus edges' that form the backbone of this neglect-associated network. (Methods - Retrospective study - 1.6-1.6.2 lines 674-747).

We then validate the clinical relevance of these specific pathways in our patient cohort by performing a correlation analysis between the percentage of surgical disconnection for each consensus edge and the postoperative change in visuospatial performance. (Methods - Retrospective study - 1.7 Diffusion tractography in single patients / 1.7.1 Validation of consensus edge disconnection against postoperative deficits lines 760-803).

Filtering DSM to right hemisphere patients: The discussion does not adequately address the decision to exclude left hemisphere patients from the DSM analysis. While the right hemisphere's dominance in visuospatial attention is well-documented, left hemisphere lesions or disconnections (e.g., commissural tracts) could still play a role. For instance, the DES findings suggest that left hemisphere stimulation occasionally induces neglect-like errors. This omission limits the scope of the DSM analysis and deserves critical discussion here.

We thank the Reviewer for this important question. As we have detailed extensively in our replies to Reviewer 1 (points #7 and #12) and to this Reviewer's earlier point on "Disconnectome symptom mapping", our decision to focus on the right hemisphere was driven by null findings in both lesion-based (SVR-LSM) and exploratory disconnection-based (SVR-DSM) analyses in the left-hemisphere cohort.

Beyond those analyses, our prospective DES results provide the most direct causal evidence supporting this lateralization. Regarding the Reviewer's comment on the DES findings, we would like to clarify a key result from our intraoperative mapping (Results-Prospective study-

2.2 Intraoperative stimulation lines 287-319). While DES in the left hemisphere did elicit some responses (i.e., task-unrelated involuntary eye movements), it crucially failed to produce any visuospatial selective attention errors (i.e., target omissions), which was the specific deficit of interest. This contrasts with the right hemisphere, where DES reliably produced target omissions.

Differences between the Bells Test (used for LSM and DSM) and the iVSAT (used during DES) could explain why DES findings suggest bilateral and ventrolateral contributions that are not captured in LSM or DSM. The discussion does not consider how task demands might influence which regions and tracts are identified as critical.

We appreciate the reviewer's insightful comment. We agree with the reviewer that this discrepancy might reflect differences in task demands and sensitivity. We now propose an hypothesis to explain this observation into the Discussion (lines 515-526):

"However, an apparent discrepancy emerged between this last intraoperative finding and postoperative outcome: resection of these same ventrolateral regions was not associated with a specific deficit on the Bells Test. This mismatch does not necessarily imply functional resilience but might reflect the differential sensitivity of the two tasks to network disruption. The iVSAT, which demands rapid detection of infrequent targets among distractors, might engage more the stimulus-driven functions of the VAN compared to the Bells Test, which is untimed, self-paced and barely requires fast target detection. Therefore, the absence of a postoperative deficit on the Bells test likely indicates that our standard assessment tool was not sensitive to the specific VAN-related impairment possibly unmasked by the iVSAT during surgery. Future work should pair disconnection measures with tasks that more explicitly probe VAN operations—such as those measuring vigilance or stimulus-driven reorienting—to reveal the true functional cost of disconnecting these ventrolateral frontal pathways."

Methods

"Only voxels resected in at least 5% of the patients were included." What does that mean? Does that mean the lesion masks were modified before running the disconnectome to only contain the voxels present in at least 5% of the patients? This is hardly conventional. There doesn't seem to be a justification for that choice. Added to the selection of right hemisphere lesions, this is a lot of filtering.

The Reviewer is correct that we only included voxels resected in at least a certain percentage of patients (10%, as stated in our Methods - Retrospective study - 1.5 Lesion-symptom mapping, lines 655-657 please see Reviewer's 3 point 4) in our group-level SVR-LSM analysis. However, we want to clarify that this is not a modification of the individual patient lesion masks. Rather, it is a standard statistical threshold applied at the group analysis stage. The threshold only defines which voxels across the group have sufficient data to be included in the final statistical map.

Excluding voxels that are not damaged in a minimum number of patients is a conventional and recommended procedure to restrict the analysis to regions where there is adequate statistical power to draw meaningful inferences. This restriction to voxels with 'sufficient lesion affection'

prevents results from being biased by regions that are only rarely damaged (Karnath, Sperber and Rorden, 2019).

Karnath HO, Sperber C, Rorden C. Mapping human brain lesions and their functional consequences. *Neuroimage*. 2018 Jan 15;165:180-189. doi: 10.1016/j.neuroimage.2017.10.028.

Figure 6: Why are the colour scales changing between panels D, E and F? They seem to be on the same scales of values.

The figure is now amended using a consistent color bar (Figure 5 in the updated manuscript).

Supplementary Table 2: the caption is unclear and incomplete.

For context, the goal of that analysis was to identify which major white matter tracts were spatially associated with our regions of interest. The table showed the results of an analysis using the Tractotron tool, which calculated the probability of overlap between our key regions (e.g., the SVR-LSM cluster) and the white matter tracts of the Human Connectome Project atlas.

However, as part of our significant revision of the manuscript, we concluded that this atlas-based approach was less specific than a direct connectomic analysis. Therefore, this supplementary table and the corresponding Tractotron analysis have been entirely removed. They are replaced by our new region-to-region connectome analysis detailed in the Methods.

Reviewer #3 (Remarks to the Author):

In the current study, Puglisi and collaborators combined Lesion-Symptom-Mapping (LSM) in 153 brain tumour patients and Direct Electrical Stimulation (DES) in 47 patients during awake neurosurgery to unveil the network causally associated with visuospatial attention. LSM and DES identified different grey and white matter regions in a right dorsomedial frontal network linked to asymmetry score.

I appreciate the effort and complexity in conducting this study, which aims to contribute to our understanding of the neural correlates of visuo-spatial attention through LSM and DES. The topic is relevant and of interest to the field. However, several concerns regarding the methodological and conceptual aspects of the study should be addressed to strengthen its contribution and improve its clarity and validity.

Major Concerns

1. The authors rely on specific test scores to define visuo-spatial attention performance. However, these do not align with established diagnostic criteria for visuo-spatial neglect, limiting the study's ability to accurately infer crucial knowledge from pathological visuo-

spatial attention. Incorporating clinical evidence of visuo-spatial neglect in the tested patients using the standard diagnostic frameworks would significantly enhance the validity of the findings.

We thank the Reviewer for letting us clarify this point. Indeed our visuospatial attention assessment doesn't allow for diagnosis of specific clinical dissociations (i.e personal/egocentric/allocentric neglect). As a retrospective study, our neuropsychological battery comprised a collection of established tests assembled for efficiency under clinical time constraints (see response #5 to Reviewer 1). Within this framework, we employed the Bells Test, since cancellation tasks—particularly the Bells Test—have been shown to exhibit high sensitivity for detecting visuo-spatial neglect in foundational studies (Ferber & Karnath, 2001; Azouvi et al., 2002, Menon, 2004). For this reason, although our assessment of visuospatial neglect is not yet fully complete, we believe it is accurate enough to detect significant changes in the selective / exploratory component of visuospatial attention.

Ferber S, Karnath HO. How to assess spatial neglect—line bisection or cancellation tasks? *J Clin Exp Neuropsychol*. 2001 Oct;23(5):599-607. doi: 10.1076/jcen.23.5.599.1243.

Azouvi P, Samuel C, Louis-Dreyfus A, Bernati T, Bartolomeo P, Beis JM, Chokron S, Leclercq M, Marchal F, Martin Y, De Montety G, Olivier S, Perennou D, Pradat-Diehl P, Prairial C, Rode G, Siérouff E, Wiart L, Rousseaux M; French Collaborative Study Group on Assessment of Unilateral Neglect (GEREN/GRECO). Sensitivity of clinical and behavioural tests of spatial neglect after right hemisphere stroke. *J Neurol Neurosurg Psychiatry*. 2002 Aug;73(2):160-6. doi: 10.1136/jnnp.73.2.160.

Menon A, Korner-Bitensky N. Evaluating unilateral spatial neglect post stroke: working your way through the maze of assessment choices. *Top Stroke Rehabil*. 2004 Summer;11(3):41-66. doi: 10.1310/KQWL-3HQL-4KNM-5F4U.

2. The anatomical correlates identified in the study appear to reflect task-specific processes rather than generalizable visuo-spatial attention mechanisms, providing only a limited theoretical framework. Previous studies (e.g., Verdon et al., 2010; Committeri et al., 2007) have demonstrated neural correlates based on task type variability. The tasks employed in this study visiomotor cancellation (Bells Test) and visual exploration (adapted Diller)—tap into distinct cognitive functions, which should be explicitly acknowledged and discussed. Additionally, the results largely confirm previous findings and their novelty could be better contextualized as it does not emerge in the actual text the authors provide.

We agree with the Reviewer that the two tasks require different cognitive demands. We now explicitly discuss the implications of the difference between our intraoperative and postoperative tasks in the discussion (lines 515-526):

“However, an apparent discrepancy emerged between this last intraoperative finding and postoperative outcome: resection of these same ventrolateral regions was not associated with a specific deficit on the Bells Test. This mismatch does not necessarily imply functional resilience but might reflect the differential sensitivity of the two tasks to network disruption. The iVSAT, which demands rapid detection of infrequent targets among distractors, might engage more the stimulus-driven functions of the VAN compared to the Bells Test, which

is untimed, self-paced and barely requires fast target detection. Therefore, the absence of a postoperative deficit on the Bells test likely indicates that our standard assessment tool was not sensitive to the specific VAN-related impairment possibly unmasked by the iVSAT during surgery. Future work should pair disconnection measures with tasks that more explicitly probe VAN operations—such as those measuring vigilance or stimulus-driven reorienting—to reveal the true functional cost of disconnecting these ventrolateral frontal pathways”.

However, both tasks ultimately rely on a shared core function - exploratory component of visuo-spatial attention—traditionally associated with frontal regions (Verdon et al., 2010), compared with the more perceptual component typically related to parietal areas. In this respect, our data are consistent with the findings of Verdon and colleagues and with more recent evidence on fronto-medial regions (Herbet et al. 2022). While theoretically supportive, we believe, after this revision, that our study provides substantial novelty:

- 1) From a methodological standpoint, it represents the first convergent causal mapping combining lesion mapping with direct electrical stimulation to achieve a higher degree of causal inference than prior studies.
- 2) By providing greater anatomical specificity compared to previous studies (Herbet et al., 2022; Nakajima et al, 2021): we replaced our previous SVR-DSM analysis and atlas-based tract identification with a new multi-stage connectomic pipeline (please see our response to the introductory comments from Reviewer 1 as well as replies to points #7 and #12) . First, instead of identifying predefined tracts, we use the SVR-LSM cluster as a seed to build a quantitative, region-to-region connectome from 100 healthy HCP subjects, with connections weighted by Fibre Bundle Capacity (FBC). Second, we apply graph theory to each HCP connectome to identify its most critical structural 'consensus hubs' and 'consensus edges' (connections between grey-matter parcels) providing a specific, network-based characterization of a visuospatial neglect associated network. Finally, this granular model enabled the clinical validation of these specific pathways in our patients, where we demonstrate that disconnection of the right SCEF-thalamus, right SCEF-brainstem, and the transcallosal SFL-SCEF pathways most strongly associate with the neglect deficit. We have discussed the potential role of these specific pathways in the Discussion (lines 435-476).The new analysis is described in Methods - retrospective study, lines 674-803.

Verdon V, Schwartz S, Lovblad KO, Hauert CA, Vuilleumier P. Neuroanatomy of hemispatial neglect and its functional components: a study using voxel-based lesion-symptom mapping. *Brain*. 2010 Mar;133(Pt 3):880-94. doi: 10.1093/brain/awp305.

Herbet G, Duffau H. Contribution of the medial eye field network to the voluntary deployment of visuospatial attention. *Nat Commun*. 2022 Jan 17;13(1):328. doi: 10.1038/s41467-022-28030-3. PMID: 35039507; PMCID: PMC8763913.

Nakajima, R., Kinoshita, M., & Nakada, M. (2021). Simultaneous Damage of the Cingulate Cortex Zone II and Fronto-Striatal Circuit Causes Prolonged Selective Attentional Deficits. *Frontiers in Human Neuroscience*, 15, 762578.

3. The broad age range of participants (16–77 years) introduces potential variability due to developmental and age-related neurobiological changes, such as ongoing myelination processes in younger participants, especially considering that the sample investigated mainly concerns patients with frontal lobe lesions. Including age as a covariate in the analysis would control for these effects and enhance the robustness of the conclusions.

We agree that the wide age range of our cohort is an important factor to consider, and that developmental and age-related neurobiological changes could potentially influence visuospatial abilities and the effects of frontal lobe lesions.

To address this, we have now incorporated age as a covariate in our SVR-LSM (Methods Retrospective study 1.5 Lesion-symptom mapping lines 633-647). Please note that, as suggested by the other Reviewers, education and tumour grade were also inserted in the model as covariates.

As reported in the updated Results (Retrospective study - 1.3 Lesion-symptom mapping - SVR-LSM lines 184-207), the inclusion of these covariates did not alter the main findings (Figure 1 DG). The same fronto-medial cluster was identified as being significantly associated with postoperative visuospatial neglect, confirming the robustness of our results. This indicates that the observed brain-behavior relationship is primarily driven by the specific lesion location rather than being confounded by age or the other controlled-for variables.

4. The LSM method includes only voxels resected in at least 5% of participants. While there are no strict guidelines for this threshold, a higher value (e.g., 10% of the sample) is generally recommended to enhance reliability and reduce noise. It would be helpful for the authors to either adjust this threshold or provide a clear justification for their choice to ensure the robustness of the findings.

We agree that an appropriate voxel inclusion threshold is critical for robust lesion-symptom mapping.

In our initial analysis, we used a 5% threshold. Following the Reviewer's suggestion, we have now re-run the SVR-LSM analysis using a more conservative 10% threshold to enhance reliability and reduce potential noise from rarely damaged brain regions.

We can confirm that the main findings remained unchanged with this stricter threshold, demonstrating the robustness of our results. We have now updated the Methods section (Retrospective study - 1.5 Lesion-symptom mapping lines 655-657) to reflect the use of this 10% threshold in the final analysis presented in the manuscript.

5. While the paper emphasizes convergent results, the lack of task performance interference during DES applied at the cortical level rather suggests discrepancies in the findings. This evident contradiction should be addressed to provide a more nuanced interpretation of the results and their implications for the hypothesized frontoparietal network.

We thank the Reviewer for highlighting this important point and allowing us to clarify what might have appeared as a discrepancy in our findings concerning intraoperative stimulation.

The statement in our initial manuscript, "*In both right and left hemisphere patients, no interferences in task performance were evoked when DES was applied at cortical level,*" refers specifically to the initial phase of intraoperative mapping. As now clarified in the revised Results section (Prospective study - 2.2 Intraoperative stimulation lines 294-300), this initial mapping is performed on the exposed cortical surface (convexity) before any corticectomy begins.

Our SVR-LSM analysis (Results - Retrospective study - 1.3 Lesion-symptom mapping - SVR-LSM lines 184-207) identified cortical clusters associated with postoperative visuospatial neglect primarily located within the superior frontal sulcus and on the mesial surface in the middle cingulate gyrus. These SVR-LSM-identified cortical regions are, by their anatomical nature, not directly accessible for stimulation during the initial "cortical surface mapping" phase. Therefore, the absence of iVSAT interference during DES of the exposed cortical convexity is fully consistent with, and not contradictory to, the SVR-LSM findings implicating these deeper sulcal and mesial cortical areas.

Crucially, intraoperative mapping did elicit neglect-like errors when DES was applied at the subcortical level, as the surgical resection progressed and provided access to the regions below the SFG and MFG. Our Probability Density Estimation (PDE) of these subcortical eloquent sites (Results - Prospective study - section 2.3 lines 321-329, Figure 5E) shows that the highest probability of evoking neglect-like errors was associated with the white matter below the transition between the SMA-proper and the pre-SMA, and also with the grey matter of the mid-cingulate gyrus itself.

The PDE of sites where DES elicited neglect-like errors shows substantial spatial overlap with the SVR-LSM cluster. This is now clarified in our revised Results (Prospective study - section 2.3, lines 330-338 Figure 5G): "*Regarding the convergence between intraoperative findings and lesion-symptom mapping, the right hemisphere PDE of sites eliciting 'neglect-like' errors (highest probability, >50%) showed substantial spatial overlap with the significant SVR-LSM cluster associated with visuospatial neglect. This overlap was particularly prominent in the grey matter of the mid-cingulate gyrus and in the white matter below the SMA / preSMA transition (Fig. 5G).*"

We have revised the Results section (Prospective study - 2.2 Intraoperative assessment of visuospatial attention lines 294-300) to more clearly delineate the stages of DES mapping (cortical surface vs. subcortical).

6. The association between DES and lesion location with visuo-spatial attention deficits has been demonstrated in prior studies. For example, the involvement of the right-lateralized frontoparietal network in cancellation tasks is well-documented. The manuscript could better highlight how its findings advance existing knowledge or test specific theoretical models of attention.

We thank the Reviewer for this critical point, which gives us the opportunity to clarify the specific and novel contributions of our study. While we agree that the involvement of a right-lateralized frontoparietal network is documented, our study advances the field in two ways: by providing the first systematic, large-scale DES investigation of the frontal lobe's contribution, and by using a task specifically designed to test the selective / exploratory component of visuospatial attention.

1) Novel Anatomical Focus on the Frontal Lobe:

Previous DES studies of neglect have provided important evidence but have predominantly centered on the parietal lobe. For example, the seminal works by Roux et al. (2011) and Vallar et al. (2014) identified critical sites in posterior temporo-parietal regions. The study by Rolland et al. (2018) focused exclusively on the right inferior parietal lobule. Differently, our prospective cohort of 47 patients (27 in the right hemisphere) represents, to our knowledge, the largest and most systematic DES investigation specifically mapping the frontal lobe for visuospatial selective attention.

2) Specificity of the iVSAT:

Prior DES studies have almost exclusively relied on the line bisection task (Roux et al., 2011; Rolland et al., 2018; Vallar et al., 2014). This task is excellent for identifying the perceptual-attentional component of neglect (i.e., a bias in spatial representation). However, it is not designed to probe the selective and exploratory components of attention (Verdon et al. 2010; Herbet et al. 2022). We acknowledge a technical note by Conner et al. (2016) that reported the use of a cancellation task; however, this was a feasibility study in only 7 patients (all posterior temporal or inferior parietal; eloquent sites found in three patients) and did not involve network mapping.

Our intraoperative Visuospatial Attention Test (iVSAT) was specifically designed to test the selective / exploratory component of visuospatial attention by requiring patients to actively search for and identify specific targets among distractors. It is this task specificity that enabled our primary novel finding: a functional double dissociation where DES of the dorsomedial frontal network produces neglect-like errors, while DES of the ventrolateral network produces bilateral errors.

In summary, by shifting the anatomical focus systematically to the frontal lobe and employing a task tailored to probe visuospatial selective attention, our study moves beyond confirming the general involvement of frontoparietal networks. It provides novel, causal evidence that enriches the existing models by demonstrating a functional-anatomical gradient within the frontal lobe itself. Please note that these considerations are now included in the Discussion (lines 389-402, lines 478--490, lines 515-526).

Roux FE, Dufor O, Lauwers-Cances V, Boukhatem L, Brauge D, Draper L, et al. Electrostimulation mapping of spatial neglect. *Neurosurgery*. 2011;69(6):1218-1231. doi:10.1227/NEU.0b013e31822aefd2

Vallar G, Bello L, Bricolo E, Castellano A, Casarotti A, Falini A, et al. Cerebral correlates of visuospatial neglect: a direct cerebral stimulation study. *Hum Brain Mapp*. 2014;35(4):1334-1350. doi:10.1002/hbm.22257

Rolland A, Herbet G, Duffau H. Awake surgery for gliomas within the right inferior parietal lobule: new insights into the functional connectivity gained from stimulation mapping and surgical implications. *World Neurosurg.* 2018;112:e393-e406. doi:10.1016/j.wneu.2018.01.053

Verdon V, Schwartz S, Lovblad KO, Hauert CA, Vuilleumier P. Neuroanatomy of hemispatial neglect and its functional components: a study using voxel-based lesion-symptom mapping. *Brain.* 2010 Mar;133(Pt 3):880-94. doi: 10.1093/brain/awp305. Epub 2009 Dec 22. PMID: 20028714.

Herbet G, Duffau H. Contribution of the medial eye field network to the voluntary deployment of visuospatial attention. *Nat Commun.* 2022 Jan 17;13(1):328. doi: 10.1038/s41467-022-28030-3. PMID: 35039507; PMCID: PMC8763913.

Conner AK, Glenn C, Burks JD, McCoy T, Bonney PA, Chema AA, Case JL, Brunner S, Baker C, Sughrue M. The use of the target cancellation task to identify eloquent visuospatial regions in awake craniotomies: technical note. *Cureus.* 2016 Nov 17;8(11):e883. doi:10.7759/cureus.883

7. I would suggest toning down statements such as " Our results confirm a right-lateralized dorsomedial circuit for contralateral visuospatial attention". A crucial limitation, as noted by the authors, is the lesion distribution bias toward the frontal lobe. This bias restricts the scope of exploration, both in terms of the lesion basis and the effects of electrical stimulation on other brain areas potentially involved in visuo-spatial attention, as already well-known literature on this topic. A more cautious interpretation of the results would better reflect these constraints.

We thank the Reviewer for this comment and fully agree that our conclusions must consider the lesion distribution in our glioma patient cohort. To better reflect this, we have toned down our claims, including removing the specific cited sentence from the abstract, to avoid any implication that our findings define the entire visuospatial attention network.

As the Reviewer notes, our intention was never to rule out the critical involvement of other regions, such as the parietal lobe. We believe the Discussion section of our manuscript already contains the cautious interpretation you recommend. Specifically, we state (see lines 545-551):

"A significant limitation of our study is the relatively small number of patients with parietal lobe involvement due to the lower incidence of gliomas compared to more frequently affected regions, such as the insular and frontal lobes."

We further acknowledge that this *"limitation, in our study, could produce an underestimation of the role of parietal regions."*

We explicitly constrain the interpretation of our findings with the following statement: *"As a consequence, our results should be confined to the role of the frontal lobe connections in visuospatial selective attention."*

We believe these explicit statements ensure that our findings are interpreted within their proper context. We discussed the involvement in visuospatial attention of a specific frontal network, without excluding the role of other key brain regions.

Minor Concerns

1. The introduction provides a narrow view of attention models. Expanding this section to include alternative theoretical frameworks would situate the study within a broader context. Additionally, the authors could elaborate on how their results challenge or support these models.

We thank the Reviewer for this suggestion. We agree that a broader theoretical context strengthens the manuscript. Accordingly, we have revised the Introduction to provide a more comprehensive overview of complementary theoretical models of visuospatial attention, including classical hemispheric specialization, spatial-motor, and arousal/vigilance frameworks (lines 64-84).

In the Discussion, we discuss how our findings align with and expand upon both Mesulam's theory of right-hemisphere dominance and Corbetta and Shulman's dual-network model, by causally implicating right dorsomedial connectivity in contralateral attention deployment and ventrolateral connectivity in non-spatial alerting functions. We chose to focus on these theoretical models as they provide the most direct and comprehensive lens through which to interpret our specific causal findings related to frontal lobe contributions to visuospatial exploration and neglect.

To further address how our results challenge or create a dialogue with existing findings, we also discuss the apparent discrepancy between our results and the classical stroke literature, where neglect is often associated with lesion of lateral white matter.

Mesulam, M. M. (1981). A cortical network for directed attention and unilateral neglect. *Annals of Neurology*, 10(4), 309–325. <https://doi.org/10.1002/ana.410100402>.

Kinsbourne, M. (1987). Mechanisms of Unilateral Neglect. In M. Jeannerod (Ed.), *Advances in Psychology* (Vol. 45, pp. 69–86). North-Holland. [https://doi.org/10.1016/S0166-4115\(08\)61709-4](https://doi.org/10.1016/S0166-4115(08)61709-4).

Rizzolatti G, et al. (1987). *Reorienting attention across the horizontal and vertical meridians: Evidence in favor of a premotor theory of attention*. 25(1A):31.

Corbetta, M., & Shulman, G. L. (2011). Spatial neglect and attention networks. *Annual Review of Neuroscience*, 34, 569–599. <https://doi.org/10.1146/annurev-neuro-061010-113731>.

Heilman KM, et al. (1978, March). Hypoarousal in patients with the neglect syndrome and emotional indifference. 229:32.

2. While the authors critique the limitations of LSM, they also rely on this methodology. Balancing this critique with an acknowledgement of DES limitations, such as diffusion to neighbouring regions, would provide a fairer assessment of both approaches.

We thank the reviewer for encouraging a fair and balanced assessment of all methodologies used.

Our observation in the introduction was specifically aimed at the well-documented limitations of lesion-symptom mapping (LSM) in stroke patients, where the non-uniform distribution of lesions introduces a significant localization bias. We sought to highlight that

a key strength of our study is the use of a glioma patient cohort, which helps to mitigate this specific issue (line 90-99).

More importantly, as the Reviewer notes, we have now revised the introduction to provide the balanced critique that was requested. We now explicitly acknowledge that Direct Electrical Stimulation (DES) also has its own limitations. Our core argument, which we have also clarified, is that the strength of our study lies not in the superiority of any single method, but in their combination. We now state (lines 98-109):

“To address these limitations, we performed a study on patients undergoing awake neurosurgery for brain tumour resection. Distinctly to previous investigations in this area (Herbet & Duffau, 2022; Nakajima et al., 2021), we aimed to conduct a convergent causal mapping by integrating multiple causal approaches, specifically lesion-symptom mapping (LSM) and direct electrical stimulation (DES) (Duffau, 2015; Fornia et al., 2024; Puglisi et al., 2019; Viganò et al., 2022). DES, despite its inherent limitations (e.g. the potential spread of current to neighbouring and/or remote regions, (Borchers et al., 2012; Duffau, 2015)), provides a causal reversible probe to test anatomo-functional relationship with a unique invasive access to the human brain. The strategy of combining these two causal methods allows their respective limitations to be mitigated. By demonstrating concordance between permanent deficits from surgical lesions (LSM) and transient deficits from DES at the same anatomical locations, a high degree of causal inference can be achieved (Siddiqi et al., 2022).”

3. Reducing the use of complex acronyms (e.g., DorsAttnB_FEF_3) and providing consistent explanations for terms would improve readability and accessibility for a broader audience.

In the revised manuscript (Results, 1.3 Lesion-symptom mapping - SVR-LSM 208-217), we added to these acronyms their corresponding anatomical descriptions, thereby clarifying the location of the parcels within their respective large-scale networks. In the Discussion, we avoided the use of complex acronyms and we referred to the general large scale networks definitions.

4. The rationale for using the 17/1000 Schaefer et al. (2018) parcellation should be clarified. Why was this specific framework chosen, and how does it align with the study's aims?

We thank the Reviewer for asking for clarification on our choice of the Schaefer et al. (2018) parcellation for characterizing the network affiliations of our SVR-LSM significant cluster.

Our selection of the Schaefer atlas aligns with the study's aims for several reasons:

1) Alignment with Canonical Functional Networks: A primary reason for choosing the Schaefer atlas is its explicit construction to be functionally aligned with the widely adopted 7/17-network parcellations of Yeo et al. (2011). Each Schaefer parcel is assigned to one of these canonical Yeo networks. This is crucial for our study, as these Yeo networks include clearly delineated Dorsal Attention Network (DAN) and Salience/Ventral Attention

Network (VAN) components—systems that are relevant to the investigation of visuospatial attention. Using the Schaefer atlas thus ensures that our parcel-based lesion load findings can be directly interpreted within the context of these well-characterized and pertinent large-scale functional systems.

2) Widespread Adoption and Comparability: The Schaefer atlases are highly cited and have become a standard in the neuroimaging community. Utilizing this framework enhances the interpretability of our findings and facilitates comparison with a broad range of other studies investigating brain networks.

3) Granularity for Detailed Characterization (see also Reviewer 1 point #6): we specifically utilized the Schaefer 1000-parcel version. Our SVR-LSM cluster comprises different focal components, and this high-resolution parcellation allows for a precise localization of lesion overlap when these components align with finer functional distinctions within broader network regions. While our primary network-level reporting aggregates these findings to the 17 Yeo networks, the finer-grained analysis supports a more detailed anatomical interpretation of which specific sub-regions within these networks are implicated by the SVR-LSM ROI. Schaefer et al. (2018) themselves note that higher-resolution parcellations can be more effective if the effect of interest is highly focal.

We have justified this choice in the Methods section (lines 666-670).

5. The section titled "Impact of iVSAT use on neuropsychological outcomes" could be revised for clarity. Additionally, clearer explanations of terms and concepts throughout the manuscript would enhance understanding.

We appreciate the reviewer's feedback. The section "Impact of iVSAT use on neuropsychological outcomes" (now "2.5 Effect of Intraoperative Visuospatial Selective Attention Task (iVSAT) on Postoperative Visuospatial Neuropsychological Outcomes lines 355-374" has been thoroughly revised for improved clarity and complemented with statistical analysis of postoperative outcome.

6. Another issue is that the results are presented using a neuroradiological convention rather than a neurological convention. I suggest that the authors align the presentation with the neurological convention, ensuring that right hemisphere results are displayed on the right side. This adjustment would improve clarity and consistency for readers in the field.

All results are now presented with a neurological convention.

7. The authors used ANOVA to analyze the behavioural data and included the test asymmetry score in the LSM analysis. Were the data assessed for normality to ensure that the assumptions of ANOVA were met? If not, it would be important to evaluate whether the data distribution is suitable for this analysis or consider alternative statistical approaches. Additionally, the LSM analysis may also be impacted by non-normal data distribution, which could affect the robustness and interpretability of the results. Addressing these issues would help strengthen the study's methodological rigour.

We thank the Reviewer for highlighting the importance of verifying distribution assumptions. In response, we conducted formal normality tests on each behavioral measure (Total and asymmetry scores). Since both variables significantly deviated from a normal distribution (Shapiro–Wilk $p < .001$), in order to ensure the robustness of our findings, we also performed distribution-free analyses using Wilcoxon signed-rank tests for within-subject comparisons. These tests confirmed a significant increase in total omissions from pre- to post-surgery in both left-hemisphere and right-hemisphere patients, consistent with the results from the ANOVA. Wilcoxon signed-rank tests on asymmetry scores confirmed that left-hemisphere patients showed no significant pre–post change, whereas right-hemisphere patients exhibited a robust postoperative increase in asymmetry score (Results - Retrospective study - 1.2 Neuropsychological Assessment lines 150-182).

As regards the potential effect of the distribution on the LSM, we agree that parametric lesion-symptom mapping (LSM) methods can be sensitive to behavioural non-normality. However, SVR-LSM makes no assumptions about the underlying distribution of the dependent or independent variables. Furthermore, as a multivariate technique, SVR-LSM offers greater sensitivity for detecting brain-behavior relationships compared to traditional univariate methods by considering the full spatial pattern of the lesion rather than the status of individual voxels in isolation (Smola & Schölkopf, 2004; Zhang et al., 2014; Demarco and Turkeltaub 2018).

References

Smola, A.J., Schölkopf, B. A tutorial on support vector regression. *Statistics and Computing* 14, 199–222 (2004). <https://doi.org/10.1023/B:STCO.0000035301.49549.88>

Zhang Y, Kimberg DY, Coslett HB, Schwartz MF, Wang Z. Multivariate lesion-symptom mapping using support vector regression. *Hum Brain Mapp.* 2014 Dec;35(12):5861-76. doi: 10.1002/hbm.22590. Epub 2014 Jul 16. PMID: 25044213; PMCID: PMC4213345.

DeMarco AT, Turkeltaub PE. 2018. A multivariate lesion symptom mapping toolbox and examination of lesion-volume biases and correction methods in lesion-symptom mapping. *Human Brain Mapping* 39(11): 4169–4182. <https://doi.org/10.1002/hbm.24289>

8. When performing the SVR-LSM analysis, the authors chose to control for lesion volume using the dTLVC approach described by Zhang et al. (2014). However, the SVR-LSM software (as released by De Marco and collaborators) includes the capability to control lesion volume directly, accounting for both the lesion map and behavioural data. Why did the authors opt not to use this implementation suggested by De Marco et al.? Clarifying this choice would provide valuable insight into the methodological decisions and their potential impact on the results.

We thank the Reviewer for raising this interesting point.

We now implemented the “regress-out-of-both” approach (i.e. regressing lesion volume from both the behavioural scores and the lesion maps) and compared it directly to our dTLVC pipeline. As shown in Revision Figure 1, the regress-both correction yields a substantially larger suprathreshold cluster compared to dTLVC and drives the average prediction accuracy to -0.20 (SD = 0.08), compared with the optimal value obtained with dTLVC (0.30 , SD = 0.09).

At present, there is still no consensus on the gold standard for volume correction (Zhang et al. 2014; DeMarco and Turkeltaub 2018; Wiesen et al. 2019). Similarly to our approach, in a recent paper, Wiesen et al. (2019) directly compared the two methods, finding that dTLVC yielded a better prediction accuracy for their dataset and finally opting for this solution for presenting their main results.

Zhang Y, Kimberg DY, Coslett HB, Schwartz MF, Wang Z. 2014. Multivariate lesion-symptom mapping using support vector regression. *Human Brain Mapping* 35(12): 5861–5876. <https://doi.org/10.1002/hbm.22590>

DeMarco AT, Turkeltaub PE. 2018. A multivariate lesion symptom mapping toolbox and examination of lesion-volume biases and correction methods in lesion-symptom mapping. *Human Brain Mapping* 39(11): 4169–4182. <https://doi.org/10.1002/hbm.24289>

Wiesen D, Sperber C, Yourganov G, Rorden C, Karnath H-O. 2019. Using machine learning-based lesion behavior mapping to identify anatomical networks of cognitive dysfunction: Spatial neglect and attention. *NeuroImage* 201: 116000. <https://doi.org/10.1016/j.neuroimage.2019.07.013>

Supplementary Figure 1: Significant SVR-LSM cluster associated with postoperative Bell's Test asymmetry score changes, corrected for cluster-level family-wise error. The figure compares clusters derived using two different lesion volume control methods: (Red) Controlling for lesion volume via regression in the SVR model (prediction accuracy: -0.20 ± 0.08 SD). (Green) Using direct total lesion volume control (dTLVC; prediction accuracy: 0.30 ± 0.09 SD). The dTLVC method was adopted for the main analyses presented in the manuscript.

Reviewer #1 (Remarks to the Author):

I have now read the revised manuscript in detail. I would like to thank the authors for taking my comments and suggestions very seriously and for their impressive work. I believe the revised anatomical analysis workflow is much more robust, and the subsequent results go well beyond what I was expecting. In particular, I am very pleased to see that the authors now clearly succeed in delineating the relevant white matter systems underlying visuospatial neglect, especially the implication of the projections of the supplementary cingulate eye field. From my perspective, this finding is highly original and carries important implications both fundamentally and clinically.

My final suggestion would be to provide a methodological figure, placed at the end of the introduction, to help readers easily navigate the methodological workflow, given the large number of analyses presented.

Congratulations on this excellent work.

We thank the Reviewer for the positive evaluation of our revised manuscript and for the insightful comments, which have significantly contributed to improving the quality and impact of our work.

We have created a new figure to provide a clear overview of the study's methodological workflow. This figure is now referenced at the end of the Introduction. The following sentence has been added to the end of the introduction:

"The overall workflow of the study, detailing the integration of these distinct methodological stages, is illustrated in Figure 1." (lines 141-142).

Reviewer #2 (Remarks to the Author):

The authors addressed all major points. Either they implemented the changes I requested, or they replaced the criticised analyses with more robust alternatives and explained their rationale.

I would just note that the rebuttal could have been clearer if the authors had quoted their revised text directly. As it is, checking the changes requires going back and forth with the manuscript, which makes the process a bit tedious.

We thank the Reviewer for the positive evaluation and for confirming that our revisions have adequately addressed the initial concerns. We apologize that the rebuttal format made the review process difficult.

Reviewer #3 (Remarks to the Author):

The authors have significantly improved the manuscript, and I acknowledge the imaging methodological innovation and the considerable effort involved in testing such a large cohort of patients. I congratulate the authors on their work. However, several points still require clarification. While the results are of interest and contribute to the field, they substantially overlap with findings from previous classical neuropsychological studies, thus somewhat

modulating the overall novelty of the present work. There are also specific methodological aspects, particularly concerning the neuropsychological testing, that need to be more explicitly addressed.

We thank the Reviewer for the positive assessment of our work and for this comment, which allows us to clarify the specific novel contributions of our work in the context of the existing literature.

This study specifically investigates the selective/exploratory component of visuospatial attention, a function that the neuropsychological literature has associated with the right frontal lobe. Our work builds upon this foundational knowledge by providing a new level of anatomical precision and causal validation.

We believe our novel contribution is threefold:

- Our study provides the first convergent causal mapping of this system by integrating multivariate lesion-symptom mapping (LSM) with direct electrical stimulation (DES). This dual approach allows for a higher degree of causal inference than is possible with either method alone.

- We provide a new level of anatomical detail that surpasses not only classical lesion studies but also recent work in this area performed in neurosurgical patients (Nakajima et al. 2021; Herbet et al. 2022). Our novel connectomic pipeline delivers a quantitative, region-to-region description of this frontal sub-network. This allowed us to identify specific pathways—namely the right SCEF-thalamus, right SCEF-brainstem, and the transcallosal SFL-SCEF connections—whose surgical disconnection is most significantly correlated with postoperative performance decrease in the asymmetry score of the Bells Test.

- Our intraoperative mapping provides direct causal evidence for a functional dissociation within the frontal lobe. We demonstrate that stimulation of the dorsomedial network elicits contralateral, neglect-like errors, whereas stimulation of the ventrolateral network causes bilateral, non-lateralized attentional errors. This causally-defined dissociation adds a new input to the existing attentional models and opens new perspectives for studying visuospatial attention subnetworks within the frontal lobe.

Page 6 lines 158-175

A central issue in the manuscript concerns the use of a continuous "total score" in the analyses rather than a threshold-based pathological score for the Bell's test. While this approach may allow for more nuanced statistical analysis, it compromises the clinical interpretability of the findings. In the absence of a comparison to normative data or cut-off thresholds, it remains unclear whether the observed deficits reflect actual neuropsychological impairments (e.g., pathological score). Clarifying this point would strengthen the conclusions regarding lesion-symptom mapping and its clinical relevance.

Page 7 lines 189-193

A similar consideration applies to the LSM analysis. Although the use of a continuous behavioral score is methodologically sound, it does not allow for a clear distinction between patients with and without attentional impairments. In contrast, classical neuropsychological assessments typically compare a patient's performance to a normative dataset and classify

outcomes dichotomously (i.e., pathological vs. non-pathological). Incorporating a clinical categorical variable, indicating whether each patient shows a clinically significant deficit, could provide additional insights into the neural correlates of attention deficits. At the very least, the manuscript should acknowledge this limitation and justify the methodological choice more explicitly.

We thank the Reviewer for this important comment, as it allows us to clarify our methodological approach. The decision to use continuous behavioral scores rather than a dichotomized, threshold-based variable was a deliberate choice grounded in established statistical best practices to maximize the accuracy and power of the lesion-symptom mapping analysis (De Marco and Turkeltaub 2018; Zhang et al., 2014).

While categorical classifications are valuable in clinical settings, dichotomizing continuous data for statistical analysis introduces significant problems. This process discards a large amount of information about the variance in patients' performance, which substantially reduces statistical power and increases the risk of false-negative results. Furthermore, it relies on arbitrary cut-off points, which can lead to clinically similar patients being classified into different groups, thereby reducing the stability and replicability of the findings (MacCallum et al. 2002; Streiner et al. 2002; Altman et al. 2006).

By using a continuous dependent variable, we preserve the full spectrum of performance. This allows the Support Vector Regression Lesion-Symptom Mapping (SVR-LSM) model to learn a more nuanced relationship between the location of a lesion and the severity of the resulting deficit. This approach is more sensitive for capturing the graded effects of brain damage on function. In the SVR-LSM analysis, the use of continuous variables allows for more robust control of covariates like age, education, and resection volume, as performed in our analysis. The resulting maps are therefore more stable and provide a more precise localization of the neural correlates underlying the deficit in question.

However, to directly address the Reviewer's concern and explicitly link our findings to a traditional clinical classification, we generated a resection overlap map using only the subgroup of patients who presented with a postoperative pathological asymmetry score post-surgery. This map, derived from a purely clinical-categorical grouping, reveals that the area of maximal resection overlap converges substantially with the primary cluster identified by our continuous SVR-LSM analysis. However, by using a continuous dependent variable, the SVR-LSM output goes a step further than a simple overlap map. It allows us to identify the specific brain cluster where the degree of damage is predictive of the severity of the deficit. Furthermore, to ensure full clinical transparency, we explicitly report the number of patients who met the criteria for a pathological deficit within the manuscript and provide detailed individual patient scores in the supplementary tables. This combination ensures that while our main analysis leverages the statistical power of continuous data, the results are aligned with a traditional clinical-categorical classification.

We have added the rationale of using continuous behavioural scores and the novel analysis in the manuscript (Methods - lines 681 - 686 / Results - lines 211 - 215). Supplementary Figure 4 illustrates the results (lines 1433 - 1436).

Page 10 lines 301-315

The same limitation mentioned above applies to the iVSAT scores. It would be informative to

assess whether stimulation induced a clinically meaningful neuropsychological deficit in individual patients. This would ideally require a comparison with normative data, which could clarify whether the observed behavioral changes reflect actual deficits rather than subclinical modulations. Without such a reference point, it is more accurate to interpret the results as evidence of attentional modulation rather than as a demonstration of the neural bases of pathological attentional impairments. For instance, in the discussion, at around line 382, the authors discuss "visuospatial selective attention impairment" due to the stimulation.

We thank the Reviewer for this insightful comment and fully agree that our terminology should be more precise. We have revised the manuscript to consistently refer to these intraoperative DES effects as attentional modulations or task interferences rather than pathological impairments.

To demonstrate the clinical relevance of mapping these intraoperative modulations using the iVSAT, we point to the postoperative outcomes in the prospective cohort where this technique guided the surgery. By preserving the regions where stimulation induced these modulations, patients did not develop significant postoperative deficits, as confirmed by the Bells Test. This successful preservation of function provides evidence that these intraoperative modulations are clinically meaningful markers of eloquent brain networks.

Page 17, line 515

The use of two different tasks to assess attentional functions introduces a methodological limitation that should be explicitly acknowledged. Differences in task demands, sensitivity, and specificity may contribute to discrepancies between the patients' clinical profiles and the areas identified through stimulation (e.g., as discussed around line 515). A more detailed discussion of this point would help contextualize the results and avoid potential overinterpretations. It may also guide future studies toward a more unified and systematic testing approach.

We thank the Reviewer for this comment and agree that this point warrants a more detailed explanation. Accordingly, we have expanded and modified the Discussion section to better contextualize our findings and to stress the need for future work to adopt standardized approaches across assessment settings (lines 534-545).

Retracing the steps of my previous comment, the manuscript refers broadly to "visuospatial attention," but it is important to clarify that the study addresses only a specific component of this multifaceted construct. Although the authors' findings are in line with previous literature, the interpretation should be appropriately restrained, given the methodological differences and the specificity of the tasks employed. I recommend adjusting the tone of the manuscript to reflect that the study investigates only a subcomponent of visuo-spatial attention and does not offer a comprehensive exploration of the neural underpinnings of visuo-spatial attention.

We thank the Reviewer for this comment. We have revised the manuscript—particularly the Abstract and Discussion—to adjust the tone and to state explicitly that our study examines a specific subcomponent of visuospatial attention, namely the visuospatial exploratory/selective component, rather than the broader construct of visuospatial attention. This clarification ensures a more precise and appropriately cautious interpretation of our findings.

References

- MacCallum, R. C., Zhang, S., Preacher, K. J., & Rucker, D. D. (2002). On the practice of dichotomization of quantitative variables. *Psychological Methods*, 7(1), 19–40.
- Streiner, D. L. (2002). Breaking up is hard to do: the heartbreak of dichotomizing continuous data. *The Canadian Journal of Psychiatry*, 47(3), 262–266.
- Altman DG, Royston P. The cost of dichotomising continuous variables. *BMJ*. 2006 May 6;332(7549):1080. doi: 10.1136/bmj.332.7549.1080.
- Herbet, G., & Duffau, H. (2022). Contribution of the medial eye field network to the voluntary deployment of visuospatial attention. *Nature Communications*, 13(1), Articolo 1.
- Nakajima, R., Kinoshita, M., & Nakada, M. (2021). Simultaneous Damage of the Cingulate Cortex Zone II and Fronto- Striatal Circuit Causes Prolonged Selective Attentional Deficits. *Frontiers in Human Neuroscience*, 15, 762578.
- DeMarco, A. T., & Turkeltaub, P. E. (2018). A multivariate lesion symptom mapping toolbox and examination of lesion-volume biases and correction methods in lesion-symptom mapping. *Human Brain Mapping*, 39(11), 4169–4182.
- Zhang, Y., Kimberg, D. Y., Coslett, H. B., Schwartz, M. F., & Wang, Z. (2014). Multivariate lesion-symptom mapping using support vector regression. *Human Brain Mapping*, 35(12), 5861–5876. <https://doi.org/10.1002/hbm.22590>